# GENERALIZATION ANALYSIS OF SGD IN LINEAR REGRESSION UNDER COVARIATE SHIFT: A VIEW FROM PRECONDITIONING

## ABSTRACT

Recent years have witnessed the widespread success of stochastic gradient descent (SGD)-type algorithms across various problem domains, including those involving covariate shift tasks. However, the underlying mechanisms that enable SGD to generalize effectively in covariate shift settings, as well as the specific types of covariate shift problems where SGD demonstrates provable efficiency, remain insufficiently understood. This paper investigates SGD in the context of linear regression under a canonical covariate shift problem. Our analysis is two-fold: First, we derive an upper bound for the target excess risk of SGD, incorporating two critical practical techniques—momentum acceleration and step decay scheduling. Second, we analyze SGD's performance by framing it as a preconditioned estimator, enabling us to identify conditions under which SGD achieves statistical optimality. We demonstrate that SGD attains optimal performance in several commonly studied settings. Additionally, we demonstrate that there exist separations between several commonly used methods.

## 1 INTRODUCTION

Out-of-distribution generalization ability is necessitated by the ubiquitous distributional shift in modern machine learning tasks. Covariate shift, as a critical form of distribution shift, arises when the input distribution diverges across source and target domains, while the conditional distribution of the target given the input remains invariant (Sugiyama and Kawanabe, 2012). This phenomenon is ubiquitous in modern learning tasks. It can be exemplified by clinical tasks' heterogeneity stemming from inter-hospital variations in equipment and treatment (Guan and Liu, 2021), as well as biases in basic financial problems of loan applications where training covariate distributions are skewed toward approved applicants (Marshall et al., 2010). It also has implications for the large language model training, where curated training data diversifies real-world user prompts (Jin et al., 2024; Wang et al., 2020). A broad range of approaches has been revisited and proposed for covariate shift, spanning importance-weighting, distributionally robust optimization, and classical estimators such as maximum likelihood and ridge regression.

In contrast, the prevailing practice remains a straightforward, computationally efficient, source-only method: models are trained with SGD-type algorithms. SGD-type algorithms utilize little knowledge of the target distribution (Bottou and Bousquet, 2007; Kingma and Ba, 2015; Bottou et al., 2018). The training trajectory is determined entirely by the source data, while only a few parameters, such as momentum and step size, remain tunable (Sutskever et al., 2013; Zhang and Mitliagkas, 2019; Zhuang et al., 2020; Xie et al., 2024).

The success of SGD-type algorithms rests on the hope that knowledge distilled from the source distribution can transfer effectively to the target (Shen et al., 2021; Wenzel et al., 2022). Consequently, their empirical effectiveness naturally raises a fundamental question:

*When and why do source-driven SGD procedures remain effective under distribution shift?*

A theoretical characterization of SGD-type algorithms's generalization over covariate shift is naturally motivated and crucial to answering this question, yet remains limited.

Prevailing theoretical analyses of covariate shift use high-dimensional linear regression as a canonical setup (Ma et al., 2023). The focus on linear models is twofold. First, linear models are a cornerstone of statistical and machine-learning theory, with broad implications—including their correspondence to infinitely wide neural networks via the neural tangent kernel (Jacot et al., 2018). Many phenomena are not model-specific but already emerge in these canonical settings (Du et al., 2020; Lee et al., 2021). Second, linear models accommodate structural assumptions originate from kernel regimes (Caponnetto and De Vito, 2007), enabling fine-grained theoretical analyses and thereby yielding deeper insights (Arora et al., 2019).

Concretely, we study SGD-type methods for covariate shift in high-dimensional linear regression through the framework of preconditioned estimator: (1) First, we derive an upper bound on the excess risk of accelerated SGD (ASGD) with the exponentially decaying stepsize schedule, and translate the upper bound as the excess risk of a suitably preconditioned estimator (Pathak et al., 2024); (2) Second, from the preconditioned estimator's viewpoint, we identify minimax optimal regimes for ASGD, which cover widely examined settings. Furthermore, we demonstrate that there exist separations in the optimality regions of several methods.

Problem (1) is the technical challenging part, where we derive an upper bound on the target excess risk for ASGD under an exponentially decaying stepsize schedule in Section 4. Both ASGD and stepsize schedule are standard in linear regression optimization and crucial for achieving near-statistically optimal last-iterate excess risk. Practically, momentum and stepwise learning-rate schedules are defining features of many widely used optimizers (Nesterov, 1983; Kingma and Ba, 2015; He et al., 2016; Loshchilov and Hutter, 2017; Brown et al., 2020). Theoretically, ASGD accelerates convergence of the expected iterate, while an exponential step-size decay reduces the variance. Though both are standard, the excess risk under our framework remains chanllenging and underexplored, even in the in-distribution low-dimensional setting. Furthermore, equipped with the excess risk upper bound, we formulate ASGD as a parallel preconditioning estimator in Section 4.1, thereby clarifying our bound and facilitating the subsequent minimax analyses.

For problem (2), we provide a general condition where ASGD achieves minimax optimal rates in Section 5. The condition is shown to hold across a broad range of commonly-studied problem class. In addition, we demonstrate separations in the optimality region between several commonly used methods. First, under a construction where the target prioritizes the large eigenspace of the source covariance, ASGD achieves the optimal $\tilde{\mathcal{O}}\left(1/\sqrt{n}\right)$ rate whereas ridge regression attains a suboptimal $\tilde{\mathcal{O}}\left(1/n\right)$. Second, despite momentum can increases the noise, it can still broaden the optimality regime of SGD when the initial bias is large.

## 2 RELATED WORK

**Optimality in Covariate Shift.** There is a vast theoretical literature on the covariate shift problem (e.g., Ben-David et al. (2010); Germain et al. (2013); Cortes et al. (2010; 2019) and the review in Sugiyama and Kawanabe (2012); Kouw and Loog (2019)). Confining to the context of optimality, pioneering work includes Shimodaira (2000), which studies the weighted maximum likelihood method in the asymptotic setting, Kpotufe and Martinet (2021), which delves into a local nonparametric setup and considers the minimax optimality of a nearest-neighbor-based method. More recently, a thread of research considers the optimality of the covariate shift problem under linear/kernel regression. This includes minimax optimality under general distribution shifts, which lead to suboptimal or inapplicable results under the covariate shift problem (Zhang et al., 2022; Mousavi Kalan et al., 2020). As for the specific covariate shift problem in linear/kernel regression, seminal works Lei et al. (2021); Pathak et al. (2024) consider the preconditioned linear estimator in the linear/kernel regression setup, and establish their instance-wise minimax optimality framework. In parallel, research has examined the optimality of specific algorithms. Principal component regression has been analyzed in (Cai and Hall, 2006; Tang et al., 2025) under setups like single-point prediction. (Ma et al., 2023; Pathak et al., 2022) consider the optimality region of kernel ridge regression under function classes defined by bounded likelihood discrepancy. Ge et al. (2024) demonstrates that maximum likelihood estimation achieves optimality in low-dimensional settings. Our results delve into the prevalent SGD-type algorithm and establish a general optimality framework covering broad problem settings. And we also demonstrate that there exists a separation between the optimality region of ASGD, vanilla SGD, and ridge regression, despite their seemingly parallel optimality under standard setups.

**Stochastic Gradient Methods in Linear Regression.** Recent theoretical analyses of SGD for linear regression have tightened the link between practice and theory. In particular, exponentially decaying step-size schedules—ubiquitous in implementations—now carry minimax-optimal risk guarantees (Ge et al., 2019; Pan et al., 2022), a result that lay beyond conventional black-box analyses. Acceleration likewise remains effective under substantial gradient noise for appropriate noise models, as demonstrated by Jain et al. (2018); Varre and Flammarion (2022). The important subsequent work establishes provable generalization for stochastic-gradient methods (Zou et al., 2021; Wu et al., 2022a; Li et al., 2024; Zhang et al., 2024) under the over overparameterized problems. In the specific covariate shift problem, Wu et al. (2022b) establishes instance target excess risk upper bounds in linear for vanilla SGD under covariate shift. Back to our setup, the combination of ASGD and stepsize schedules in linear regression analysis is technical-challenging and unprecedented, even in the in-distribution and low-dimensional setting due to complex noise-propagation of fourth momentum and non-commutable matrices. Besides, a related line of work studies SGD-type algorithms for nonparametric regression. Dieuleveut and Bach (2016) analyze stochastic gradient methods in reproducing kernel Hilbert spaces and further establish their optimality. These studies, however, do not concern out-of-distribution or acceleration techniques such as momentum and step-size schedules.

## 3 PROBLEM FORMULATION AND PRELIMINARIES

**Notations.** We denote the spectral norm, Frobenius norm, and nuclear norm of a matrix $\mathbf{A}$ by $\|\mathbf{A}\|$, $\|\mathbf{A}\|_F$, and $\|\mathbf{A}\|_*$, respectively. Define the elliptical norm of vector $\mathbf{x}$ under positive definite matrix $\mathbf{M}$ as $\|\mathbf{x}\|_{\mathbf{M}}^2 = \mathbf{x}^\top \mathbf{M} \mathbf{x}$. We use $\mathbf{O}$ to denote the matrix with all entries equal to zero. For positive integer $n$, let $[n] = \{1, 2, \ldots, n\}$. The diagonal matrix with sequence $\{a_i\}_{i=1}^d$ as its diagonal entries is denoted by $\mathrm{diag}\{a_i\}_{i=1}^d$. For a vector $\mathbf{x} \in \mathbb{R}^d$, denote $\mathbf{x}_{k_1:k_2} \in \mathbb{R}^d$ as the vector where only $k_1 + 1$-th to $k_2$-th entries are kept and others are set to zero. For a matrix $\mathbf{A} \in \mathbb{R}^{d \times d}$, let $\mathbf{A}_{k_1:k_2} \in \mathbb{R}^{d \times d}$ denote the matrix obtained by retaining only the submatrix from the $(k_1 + 1)$-th to the $k_2$-th rows and columns, with all other entries set to zero.

### 3.1 LINEAR REGRESSION UNDER COVARIATE SHIFT

The regression problem using covariate $\mathbf{x} \in \mathbb{R}^d$ to predict the response $y \in \mathbb{R}$. In the covariate shift problem, there are two distinct data domains on the covariate and the response: a source domain $\mathcal{S}$ and a target domain $\mathcal{T}$. Let $P_{\mathbf{x} \times y}$ denote the joint distribution of $(\mathbf{x}, y)$ over domain $\mathcal{S}$ and $Q_{\mathbf{x} \times y}$ denote the joint distribution of $(\mathbf{x}, y)$ over domain $\mathcal{T}$.

We assume access to $n$ i.i.d. samples $\{(\mathbf{x}_i, y_i)\}_{i=1}^n$ drawn from $P_{\mathbf{x} \times y}$, while the predictor's performance is evaluated under the generalization risk on the target distribution $Q_{\mathbf{x} \times y}$. Covariate shift refers to the problems where the marginal distribution $P_{\mathbf{x}}$ may differ from the marginal distribution $Q_{\mathbf{x}}$, while the conditional distribution $y|\mathbf{x}$ remains unchanged in both domains. Denote the covariance of the source and target distributions as $\mathbf{S} = \mathbb{E}_{P_{\mathbf{x}}}[\mathbf{x}\mathbf{x}^\top]$ and $\mathbf{T} = \mathbb{E}_{Q_{\mathbf{x}}}[\mathbf{x}\mathbf{x}^\top]$. The eigenvalue decomposition of $\mathbf{S}$ and $\mathbf{T}$ are given by

$$\mathbf{S} = \mathbf{U}\mathrm{diag}\{\lambda_1, \ldots, \lambda_d\}\mathbf{U}^\top, \quad \mathbf{T} = \mathbf{V}\mathrm{diag}\{\mu_1, \ldots, \mu_d\}\mathbf{V}^\top, \tag{1}$$

where $\lambda_1 \geq \cdots \geq \lambda_d$ are eigenvalues of $\mathbf{S}$ in non-increasing order and $\{\mu_i\}_{i=1}^d$ are eigenvalues of $\mathbf{T}$ in non-increasing order. For simplicity, we assume that $\mathbf{U}$ is the standard orthonormal basis in $\mathbb{R}^d$.

For any estimator $\mathbf{w} \in \mathbb{R}^d$, the source risk $\mathcal{E}_{\mathcal{S}}(\mathbf{w})$ and target risk $\mathcal{E}_{\mathcal{T}}(\mathbf{w})$ are defined as:

$$\mathcal{E}_{\mathcal{S}}(\mathbf{w}) = \frac{1}{2}\mathbb{E}_{P_{\mathbf{x} \times y}}(y - \langle \mathbf{w}, \mathbf{x}\rangle)^2, \quad \mathcal{E}_{\mathcal{T}}(\mathbf{w}) = \frac{1}{2}\mathbb{E}_{Q_{\mathbf{x} \times y}}(y - \langle \mathbf{w}, \mathbf{x}\rangle)^2. \tag{2}$$

We impose the following assumption on the response model in both the source and target distributions.

**Assumption 1.** *For both source and target domains, the response $y$ is generated by $y = (\mathbf{w}^*)^\top \mathbf{x} + \epsilon$, where $\mathbf{w}^* \in \mathbb{R}^d$ denotes the ground truth. The noise $\epsilon$ satisfies $\mathbb{E}[\epsilon|\mathbf{x}] = 0$ and $\mathbb{E}[\epsilon^2|\mathbf{x}] \leq \sigma^2$.*

The performance of estimator $\mathbf{w}$ is evaluated by the excess risk on the target distribution $Q_{\mathbf{x} \times y}$:

$$\mathcal{R}_{\mathcal{T}}(\mathbf{w}) = \frac{1}{2}\left(\mathcal{E}_{\mathcal{T}}(\mathbf{w}) - \min_{\mathbf{w}} \mathcal{E}_{\mathcal{T}}(\mathbf{w})\right) = \frac{1}{2}\|\mathbf{w} - \mathbf{w}^*\|_{\mathbf{T}}^2. \tag{3}$$

## 3.2 ASSUMPTIONS

We adopt several assumptions widely used in kernel linear regression. We focus on the minimax optimality under the elliptical constraint framework proposed by Pathak et al. (2024).

**Assumption 2.** *We assume that the ground truth $\mathbf{w}^*$ lies in the elliptical constraint set: $W_{\mathbf{M}} = \left\{\mathbf{w}^* \in \mathbb{R}^d : \|\mathbf{w}^*\|_{\mathbf{M}}^2 \leq 1\right\}$, where $\mathbf{M} \in \mathbb{R}^{d \times d}$ is a given positive definite matrix.*

**Remark 1.** *We introduce $M$ to involve the interpolation space in the reproducing kernel Hilbert space (RKHS) framework. When $\mathbf{M} = \mathbf{I}$, the set $W_{\mathbf{I}}$ simplifies to the standard Euclidean unit ball $\left\{\mathbf{w}^* : \|\mathbf{w}^*\|_2^2 \leq 1\right\}$, which also corresponds precisely to the unit ball in the RKHS induced by the linear kernel $k(\mathbf{x}, \mathbf{y}) = \mathbf{x}^\top \mathbf{y}$. When $\mathbf{M} = \mathbf{S}^{1-s}$, the set $W_{\mathbf{S}^{1-s}}$ aligns with the unit ball in the interpolation space $[\mathcal{H}_{P_{\mathbf{x}}}]^s$ associated with the RKHS generated by the linear kernel under distributions $P_{\mathbf{x}}$. Such conditions are standard and often referred to as source conditions in the RKHS framework (Caponnetto and De Vito, 2007).*

As formalized in the following assumption, we assume that the $L_{2,P_{\mathbf{x}}}$-norm of $(\mathbf{w}^*)^\top \mathbf{x}$ is finite.

**Assumption 3.** *We assume that there exists $c > 0$ such that $\|\mathbf{w}^*\|_{\mathbf{S}}^2 \leq c$.*

**Remark 2.** *This assumption is mild and implies that for any ground truth parameter $\mathbf{w}^* \in W$, the excess risk of $\mathbf{w}_0 = \mathbf{0}$ under the source distribution $P_{\mathbf{x} \times y}$ is finite. In other words, the $L_{2,P_{\mathbf{x}}}$-norm of $(\mathbf{w}^*)^\top \mathbf{x}$ is bounded by c. Furthermore, this assumption leads to the bound $\left\|\mathbf{M}^{-1/2}\mathbf{S}\mathbf{M}^{-1/2}\right\| \leq c$.*

To derive the target excess risk upper bound for ASGD, we require the following assumption that the fourth moment of the source covariates is bounded.

**Assumption 4.** *There exists a constant $\psi \geq 1$, such that for every PSD matrix A, we have*

$$\mathbb{E}_{P_{\mathbf{x}}}\left[\mathbf{x}\mathbf{x}^\top \mathbf{A}\mathbf{x}\mathbf{x}^\top\right] \leq \psi \operatorname{tr}(\mathbf{S}\mathbf{A})\mathbf{S}. \tag{4}$$

**Remark 3.** *The assumption 4 is standard in the SGD excess risk analysis (Jain et al., 2017; 2018; Zou et al., 2021; Wu et al., 2022a;b). It holds for distributions with bounded kurtosis for the projection of $\mathbf{x}$ onto any $\mathbf{z} \in \mathbb{R}^d$. Specifically, if there exists a constant $c > 0$ such that for any $\mathbf{z} \in \mathbb{R}^d$, the following inequality holds: $\mathbb{E}_{P_{\mathbf{x}}} \langle \mathbf{z}, \mathbf{x} \rangle^4 \leq c \langle \mathbf{z}, \mathbf{S}\mathbf{z} \rangle^2$. For instance, if $\mathbf{S}^{-\frac{1}{2}}\mathbf{x}$ follows a Gaussian distribution, it holds with $\psi = 3$. Indeed, we impose this condition to handle the case where $\|\mathbf{w}^*\|_2 = \infty$. If $\|\mathbf{w}^*\|_2$ is finite, all of our conclusions hold under a weaker assumption $\mathbb{E}_{P_{\mathbf{x}}}\left[\|\mathbf{x}\|^2 \mathbf{x}\mathbf{x}^\top\right] \leq \psi\mathbf{S}$.*

## 3.3 MINIMAX OPTIMALITY

Statistical minimax optimality identifies the estimator that achieves the smallest worst-case excess risk across certain problem class. In this section, we present the minimax optimal estimator and its corresponding excess risk. The considered problem class $\mathcal{P}(W_{\mathbf{M}}, \mathbf{S}, \mathbf{T})$ is defined as below.

**Definition 1** (Problem Class). *The problem class $\mathcal{P}(W_{\mathbf{M}}, \mathbf{S}, \mathbf{T})$ consists of all independent distributions $P \times Q$ satisfy (1) $\mathbf{S} = \mathbb{E}_{P_{\mathbf{x}}}\left[\mathbf{x}\mathbf{x}^\top\right]$, $\mathbf{T} = \mathbb{E}_{Q_{\mathbf{x}}}\left[\mathbf{x}\mathbf{x}^\top\right]$; (2) Assumptions 1, 2, 3, 4 hold.*

The minimax lower bound over $\mathcal{P}(W, \mathbf{S}, \mathbf{T})$ shown by Pathak et al. (2024), is presented in Theorem 5.

**Theorem 5** (Theorem 2 in Pathak et al. (2024)). *Given positive semi-definite matrices $\mathbf{S}$, $\mathbf{T}$, $\mathbf{M}$ and probability $\tilde{P} \in \mathcal{P}(W_{\mathbf{M}}, \mathbf{S}, \mathbf{T})$, samples $\{(\mathbf{x}_i, y_i)\}_{i=1}^n$ are drawn from the source distribution of $\tilde{P}$. For any random estimator $\hat{\mathbf{w}} = \mathsf{A}\left(\{(\mathbf{x}_i, y_i)\}_{i=1}^n, \mathbf{S}, \mathbf{T}, \xi\right)$, where $\mathsf{A} : \mathbb{R}^{2d^2+n(d+1)+1} \to \mathbb{R}^d$ is an arbitrary measurable mapping, and $\xi$ encodes the algorithm's randomness, then we have*

$$\inf_{\hat{\mathbf{w}}} \sup_{\tilde{P} \in \mathcal{P}(W_{\mathbf{M}}, \mathbf{S}, \mathbf{T})} \mathbb{E}_{\tilde{P}^{\otimes n} \times P_\xi} \|\hat{\mathbf{w}} - \mathbf{w}^*\|_{\mathbf{T}}^2 \geq \sup_{\mathbf{F} \succeq \mathbf{O}, \|\mathbf{F}\|_* \leq 1/\pi^2} \left\langle \mathbf{T}', \left(\mathbf{F}^{-1} + n\mathbf{S}'/\sigma^2\right)^{-1} \right\rangle, \tag{5}$$

*where $\mathbf{S}' = \mathbf{M}^{-1/2}\mathbf{S}\mathbf{M}^{-1/2}$ and $\mathbf{T}' = \mathbf{M}^{-1/2}\mathbf{T}\mathbf{M}^{-1/2}$.*

Theorem 5 provides the algorithm-independent, worst-case lower bound over problem class $\mathcal{P}(W_{\mathbf{M}}, \mathbf{S}, \mathbf{T})$ for any instance of $\mathbf{S}$, $\mathbf{T}$ and $\mathbf{M}$, while not yielding an explicit convergence rate.

---

**Algorithm 1** Accelerated Stochastic Gradient Descent (ASGD) with exponentially decaying step size

---

**Require:** Initial weight $\mathbf{w}_0 = \mathbf{v}_0$, initial step size $\delta$, $\gamma$, momentum $\alpha$, $\beta$, total sample size $n$, $i = 1$.
    **for** $\ell = 1, 2, \ldots, \log_2 n$ **do**
        $\delta_{(\ell)} \leftarrow \delta_0/4^{\ell-1}$, $\gamma_{(\ell)} \leftarrow \gamma_0/4^{\ell-1}$
        **for** $t = 1, 2, \ldots, \frac{n}{\log_2 n}$ **do**
            Sample a fresh data $(\mathbf{x}_i, y_i)$
            $\mathbf{u}_{i-1} \leftarrow \alpha \mathbf{w}_{i-1} + (1 - \alpha)\mathbf{v}_{i-1}$
            $\mathbf{g}_i \leftarrow \left(\mathbf{x}_i^\top \mathbf{u}_{i-1} - y_i\right) \mathbf{x}_i$
            $\mathbf{w}_i \leftarrow \mathbf{u}_{i-1} - \delta_{(\ell)}\mathbf{g}_i$
            $\mathbf{v}_i \leftarrow \beta \mathbf{u}_{i-1} + (1 - \beta)\mathbf{v}_{i-1} - \gamma_{(\ell)}\mathbf{g}_i$
            $i \leftarrow i + 1$
        **end for**
    **end for**

---

### 3.3.1 LINEAR PRECONDITIONED ESTIMATOR

The linear preconditioned estimator $\hat{\mathbf{w}}_{\mathbf{A}}$ defined in the following can be viewed as a linear transformation of a generalized form of the ordinary least squares (OLS) estimator $\frac{1}{n}\mathbf{S}^{-1}\sum_{i=1}^n \mathbf{x}_i y_i$:

$$\hat{\mathbf{w}}_{\mathbf{A}} = \frac{1}{n}\mathbf{M}^{-1/2}\mathbf{A}\mathbf{M}^{1/2}\mathbf{S}^{-1}\sum_{i=1}^n \mathbf{x}_i y_i, \tag{6}$$

where $\mathbf{A} \in \mathbb{R}^{d \times d}$ is a preconditioner. The preconditioned estimator $\hat{\mathbf{w}}_{\mathbf{A}_{\mathbf{M},\mathbf{S},\mathbf{T}}^{\mathrm{Opt}}}$ achieves the minimax lower bound by minimizing the excess risk within its class. The optimal preconditioning matrix $\mathbf{A}_{\mathbf{M},\mathbf{S},\mathbf{T}}^{\mathrm{Opt}}$ is given in Lei et al. (2021); Pathak et al. (2024) as

$$\mathbf{A}_{\mathbf{M},\mathbf{S},\mathbf{T}}^{\mathrm{Opt}} = \arg\min_{\mathbf{A} \in \mathbb{R}^{d \times d}} \underbrace{\left\|(\mathbf{I} - \mathbf{A})^\top \mathbf{T}'(\mathbf{I} - \mathbf{A})\right\|}_{\text{The supremum Bias of } \hat{\mathbf{w}}_{\mathbf{A}} \text{ over } \mathcal{P}(W_{\mathbf{M}},\mathbf{S},\mathbf{T})} + \underbrace{\frac{\sigma^2 + \psi\,\|\mathbf{S}'\|}{n}\left\langle \mathbf{T}', \mathbf{A}\left(\mathbf{S}'\right)^{-1}\mathbf{A}^\top \right\rangle}_{\text{Variance of } \hat{\mathbf{w}}_{\mathbf{A}}}. \tag{7}$$

It is worth noting that both the minimax lower bound (5) and the optimal preconditioner (7) require prior knowledge of the covariance matrices $\mathbf{S}$, $\mathbf{T}$, and the constraint matrix $\mathbf{M}$.

## 4 ASGD TARGET EXCESS RISK UPPER BOUND

The empirical success of SGD-type algorithms has made direct application of them the prevalent method for solving large-scale covariate-shift problems. In this section, we establish an upper bound on the target excess risk for SGD-type algorithms within a unified framework.

As presented in Algorithm 1, we analyze the ASGD algorithm (Jain et al., 2018; Li et al., 2024), the standard acceleration method for linear regression, and adopts practical but analytically challenging geometrically decaying step sizes (Ge et al., 2019; Wu et al., 2022a). In Algorithm 1, $\mathbf{g}_i$ denotes the stochastic gradient evaluated at $\mathbf{u}_{i-1}$. The parameters $\alpha$ and $\beta$ are the momentum parameters, while $\delta_{(\ell)}$ and $\gamma_{(\ell)}$ represent step sizes initial from $\delta$ and $\gamma$. These step sizes are piecewise constant within each stage $1 \leq \ell \leq \lfloor \log_2 n \rfloor$, and are divided by 4 after each stage. Besides, when $\gamma = \delta$, Algorithm 1 reduces to the vanilla SGD method with geometrically decaying step size. To align with the subsequent minimax optimality analysis, Theorem 6 establishes a target bound on the excess risk for the class $\mathcal{P}(W_{\mathbf{M}}, \mathbf{S}, \mathbf{T})$. The class in Theorem 6 assumes $\mathbf{M}$ and $\mathbf{S}$ commute, which is a mild requirement since it encompasses the standard source condition in the associated RKHS. For a general risk bound, Appendix A.4 provides an instance-wise upper bound valid for any given $\mathbf{w}_*$.

**Parameter Choice.** The parameters in Algorithm 1 are selected according to the following scheme:

$$\delta \in \left[\frac{(\ln n)^2 \tilde{\kappa}}{c_1 n \sum_{i > \tilde{\kappa}} \lambda_i}, \frac{1}{c_2 \ln n\ \mathrm{tr}(\mathbf{S})}\right], \ \gamma \in \left[\delta, \frac{1}{c_3 \ln n \sum_{i > \tilde{\kappa}} \lambda_i}\right], \ \beta = \frac{\delta}{c_4 \tilde{\kappa}\gamma \ln n}, \alpha = \frac{1}{1 + \beta}, \tag{8}$$

for $\tilde{\kappa} \leq \tilde{\kappa}_{\sup}$, where $\tilde{\kappa}_{\sup} = \sup_{\tilde{\kappa}}\left\{\frac{\tilde{\kappa}\,\mathrm{tr}(\mathbf{S})}{\sum_{i > \tilde{\kappa}} \lambda_i} \leq \frac{c_5 n}{(\ln n)^3}\right\}$ determines the maximal admissible momentum and step size. $c_1, \ldots, c_5$ are constants; specific values are provided in Appendix A.1.3.

**Theorem 6** (Upper Bound of ASGD). *Let $\mathbf{S}$, $\mathbf{T}$, and $\mathbf{M}$ be positive semi-definite matrix such that $\mathbf{M}$ commutes with $\mathbf{S}$. Suppose we get samples $\{(\mathbf{x}_i, y_i)\}_{i=1}^n$ drawn from the source distribution of $\tilde{P} \in \mathcal{P}(W_{\mathbf{M}}, \mathbf{S}, \mathbf{T})$. When $n \geq 16$, we choose the initial step size $\delta$, $\gamma$ and the momentum $\alpha$, $\beta$ according to the parameter choice. Denote the output of Algorithm 1 as $\mathbf{w}_n^{\mathrm{SGD}}$, the target excess risk of $\mathbf{w}_n^{\mathrm{SGD}}$ over problem class $\mathcal{P}(W_{\mathbf{M}}, \mathbf{S}, \mathbf{T})$ can be uniformly bounded from the above by*

$$
\sup_{\tilde{P}} \mathbb{E}_{\tilde{P} \otimes n} \left\| \mathbf{w}_n^{\mathrm{SGD}} - \mathbf{w}^* \right\|_{\mathbf{T}}^2 \lesssim \sigma^2 \underbrace{\left[ \sum_{i=1}^{k^*} \frac{(\ln n)^2 t_{ii}}{n \lambda_i} + n (\gamma + \delta)^2 \sum_{i=k^*+1}^d \lambda_i t_{ii} \right]}_{\textit{Effective Variance}} + \underbrace{\left\| \mathbf{T}'_{k^*:d} \right\|_2}_{\textit{Effective Bias}}, \quad (9)
$$

*where $k^* = \max\left\{ k : \lambda_k > \frac{32(\ln n)^2}{(\gamma+\delta)n\ln 2} \right\}$, often referred to as the effective dimension (Bartlett et al., 2020; Zou et al., 2021), $\mathbf{T}'_{0:k^*} = \mathbf{M}^{-1/2}\mathbf{T}_{0:k^*}\mathbf{M}^{-1/2}$ and $\mathbf{T}'_{k^*:d} = \mathbf{M}^{-1/2}\mathbf{T}_{k^*:d}\mathbf{M}^{-1/2}$. $t_{ii}$ denotes the $i$-th diagonal entry of $\mathbf{T}$, and $\{\lambda_i\}_{i=1}^d$ are eigenvalues of $\mathbf{S}$.*

Theorem 6 provides the uniform target excess risk upper bound for ASGD over problem class $\mathcal{P}(W_{\mathbf{M}}, \mathbf{S}, \mathbf{T})$. The upper bound (9) decomposes into effective bias and effective variance. The effective bias corresponds to the risk of simply performing a deterministic Algorithm 1 without gradient noise, and thus depends on the deviation $\mathbf{w}_0 - \mathbf{w}^*$ between the initialization and ground truth. The effective variance quantifies the additional randomness introduced by both the noise term $\epsilon$ and $\mathbf{x}_t \mathbf{x}_t^\top$ within the stochastic gradient, as well as its complex evolution across the iterations.

Theorem 6 reveals that ASGD proceeds greedily along the eigendirections of $\mathbf{S}$, and exhibits distinct behaviors in two subspaces $\mathbf{S}_{1:k^*}$ and $\mathbf{S}_{k^*:d}$ separated by the effective dimension $k^*$. Specifically, there exists a phase transition in ASGD's excess risk: (1) In the directions corresponding to large eigenvalues (indexed by $k \leq k^*$), ASGD accurately approaches $\mathbf{w}^*$ with negligible bias, whereas the variance term $t_{kk}/(n\lambda_k)$ dominates the risk. (2) Along the directions associated with small eigenvalues ($k > k^*$), the bias remains at the same scale as in initialization, leading to a worst-case bias of $\left\| \mathbf{T}'_{k^*:d} \right\|_2$. The residual variance scales as $n(\gamma + \delta)^2 \lambda_k t_{kk}$. Therefore, the effective dimension $k^*$, as a function of the sample size $n$, the initial step size $\gamma + \delta$, and the spectral structure of $\mathbf{S}$, encapsulates ASGD's bias-reduction capacity.

**Remark 4** (Impact of Momentum). *As shown in (29), increasing the momentum $\beta$ allows for a larger admissible step size $\gamma$, which in turn leads to a larger effective dimension $k^*$ and improves ASGD's ability to reduce bias. However, if the momentum is set too large, the variance induced by $\mathbf{x}_t \mathbf{x}_t^\top$ may diverge. (29) also specifies the maximal admissible momentum and step size $\gamma^{\max}$ and $\delta^{\max} = 1/(\psi \operatorname{tr} \mathbf{S})$ that ensures convergence of the target excess risk, thereby characterizing the maximal admissible effective dimension $k^{\max} = \max\left\{ k : \lambda_k > 32(\ln n)^2 / ((\gamma^{\max} + \delta^{\max})n) \ln 2 \right\}$ and the upper limit of ASGD's bias reduction capacity.*

To bound the target excess risk of ASGD, we use an entrywise analysis of the covariance matrix along the iteration, which presents greater challenges than the eigendirection-wise approach in the in-distribution case. There are two primary challenges in this analysis: (1) Controlling the fourth-moment variance introduced by $\mathbf{x}_t \mathbf{x}_t^\top$ along the complicated propagation; (2) Precisely characterizing the bias contraction rate of the expected dynamics. These challenges arise from the use of momentum combined with decaying step sizes, which render the iteration operators ($\hat{\mathbf{A}}_t$ in defined in (18) in Appendix) piecewise non-commutative and lacking monotonic contraction properties.

When bounding the fourth-moment variance, we show that at each iteration, the covariance matrix of the stochastic update can be controlled by that of its expected counterpart. This allows us to reduce the analysis to the expected gradient descent dynamics, ignoring the fourth-moment variance. We characterize the bias contraction rate along each eigendirection. For directions with large eigenvalues ($k \geq k^*$), we show that the norm of the product of the (piecewise constant) iteration operators is dominated by the first phase, resulting in exponential decay. For directions with small eigenvalues ($k < k^*$), we prove that under a suitable projection matrix $\hat{\mathbf{P}}$ with $|\hat{\mathbf{P}}| \leq 2$, the norm of the operator product is bounded by one, leading to a bias of the same order as the initialization.

### 4.1 ASGD AS A PRECONDITIONER

In this section, we introduce a novel perspective by showing that the behavior of ASGD over the problem class $\mathcal{P}(W_{\mathbf{M}}, \mathbf{S}, \mathbf{T})$ can be effectively approximated by that of a linearly preconditioned estimator $\hat{\mathbf{w}}_{\mathbf{A}_{k^*}}$ with $\mathbf{A}_{k^*} = \operatorname{diag}\{\mathbf{I}_{k^*}, \mathbf{O}_{k^*:d}\}$, where $k^*$ denotes the effective dimension. This perspective allows us to explicitly identify the problem class for which ASGD generalizes effectively.

**Theorem 7.** *Under the conditions of Theorem 6, and for given step sizes $\gamma$ and $\delta$, the uniform target excess risk of ASGD over problem class $\mathcal{P}(W_{\mathbf{M}}, \mathbf{S}, \mathbf{T})$ can be bounded by $R_{\gamma+\delta}$:*

$$
\sup_{\tilde{P} \in \mathcal{P}(W_{\mathbf{M}}, \mathbf{S}, \mathbf{T})} \mathbb{E}_{\tilde{P}^{\otimes n}} \left\| \mathbf{w}_n^{\mathrm{SGD}} - \mathbf{w}^* \right\|_{\mathbf{T}}^2
$$

$$
\lesssim \underbrace{\left\| \left( \mathbf{I} - \begin{bmatrix} \mathbf{I}_{0:k^*} & \mathbf{O} \\ \mathbf{O} & \mathbf{O} \end{bmatrix} \right) \mathbf{T}' \left( \mathbf{I} - \begin{bmatrix} \mathbf{I}_{0:k^*} & \mathbf{O} \\ \mathbf{O} & \mathbf{O} \end{bmatrix} \right) \right\|}_{\textit{Bias of } \hat{\mathbf{w}}_{\mathbf{A}_{k^*}} \textit{ over } \mathcal{P}(W_{\mathbf{M}}, \mathbf{S}, \mathbf{T})} + \underbrace{\frac{1}{n} \left\langle \mathbf{T}', \begin{bmatrix} \mathbf{I}_{0:k^*} & \mathbf{O} \\ \mathbf{O} & \mathbf{O} \end{bmatrix} (\mathbf{S}')^{-1} \begin{bmatrix} \mathbf{I}_{0:k^*} & \mathbf{O} \\ \mathbf{O} & \mathbf{O} \end{bmatrix} \right\rangle}_{\textit{Variance of } \hat{\mathbf{w}}_{\mathbf{A}_{k^*}}}
$$

$$
+ \underbrace{n(\gamma + \delta)^2 \left\langle \mathbf{T}', \begin{bmatrix} \mathbf{O} & \mathbf{O} \\ \mathbf{O} & \mathbf{I}_{k^*:d} \end{bmatrix} (\mathbf{S}') \begin{bmatrix} \mathbf{O} & \mathbf{O} \\ \mathbf{O} & \mathbf{I}_{k^*:d} \end{bmatrix} \right\rangle}_{\textit{Residual Variance}} \equiv R_{\gamma+\delta},
$$

(10)

*where $k^* = \max \left\{ k : \lambda_k > \frac{32(\ln n)^2}{(\gamma+\delta) n \ln 2} \right\}$, $\mathbf{T}' = \mathbf{M}^{-1/2} \mathbf{T} \mathbf{M}^{-1/2}$ and $\mathbf{S}' = \mathbf{M}^{-1/2} \mathbf{S} \mathbf{M}^{-1/2}$.*

Theorem 7 shows that the uniform target excess risk of ASGD can be bounded by that of $\hat{\mathbf{w}}_{\mathbf{A}_{k^*}}$ and a residual variance arising from eigendirections outside the top-$k^*$ eigenspace of $\mathbf{S}$. The bound highlights how the trade-off between bias and variance is governed by the effective dimension $k^*$.

**Remark 5** (Bias and Variance Behavior of $R_{\gamma+\delta}$). *We refer $B_{\gamma+\delta}$ to the bias of $\hat{\mathbf{w}}_{\mathbf{A}_{k^*}}$ over $\mathcal{P}(W_{\mathbf{M}}, \mathbf{S}, \mathbf{T})$ and $B_{\gamma+\delta}$ to the sum of the variance of $\hat{\mathbf{w}}_{\mathbf{A}_{k^*}}$ and the residual variance. Thus, $R_{\gamma+\delta} = B_{\gamma+\delta} + V_{\gamma+\delta}$. For a given problem class $\mathcal{P}(W_{\mathbf{M}}, \mathbf{S}, \mathbf{T})$ and total sample size $n$, when the initial step size $\gamma + \delta$ increases, $V_{\gamma+\delta}$ increases steadily, while $B_{\gamma+\delta}$ remains flat until the effective dimension $k^*$ increases, at which point it drops sharply.*

**Remark 6** (Best Choice of Step Size over Problem Class $\mathcal{P}(W_{\mathbf{M}}, \mathbf{S}, \mathbf{T})$). *The best choice of initial step size for a given problem can be determined by minimizing $R_{\gamma+\delta}$. Let $\lambda_{k_1} > \lambda_{k_2} > \cdots > \lambda_{k_m}$ denote the distinct eigenvalues of $\{\lambda_i\}_{i=1}^d$, arranged in decreasing order. For $i \in [m]$, the index $k_i$ denotes the largest index $j$ such that $\lambda_j = \lambda_{k_i}$. For $i \in [m]$, define $\gamma^{k_i} = 32(\ln n)^2/(n \lambda_{k_i} \ln 2)$, and let $\delta^{k_i} = \min \left\{ \gamma^{k_i}, 1/(\psi \operatorname{tr} \mathbf{S}) \right\}$. Let $R(k_i) = R_{\gamma^{k_i} + \delta^{k_i}}$. Then, the best choice of $\gamma$ and $\delta$ is given by $\gamma^{k_{best}}$ and $\delta^{k_{best}}$, where $k_{best} = \min \left\{ k^\dagger, k^{\max} \right\}$, and $k^\dagger = \arg \min_{k_i} \{R(k_i)\}$ denotes the bias–variance intersection of problem class $\mathcal{P}(W_{\mathbf{M}}, \mathbf{S}, \mathbf{T})$. Moreover, the choice of step size can be practically approximated via widely used hyperparameter tuning in deep learning (Sutskever et al., 2013; Zhang and Mitliagkas, 2019; Zhuang et al., 2020; Xie et al., 2024).*

ASGD is efficient when the bias–variance intersection $k^\dagger$ is less than the maximal effective dimension $k^{\max}$. This corresponds to problem classes where $\mathbf{T}'$ is concentrated within the top-$k^{\max}$ eigenspace of $\mathbf{S}$, and leaves little mass outside it such as $\left\| \mathbf{T}'_{k^{\max}:d} \right\|$ and $\operatorname{tr}\left( \mathbf{T}'_{k^{\max}:d} \right)$ are small.

## 5 OPTIMALITY ANALYSIS

We begin the analyses with the following sufficient condition for optimality of ASGD.

**Theorem 8.** *Recall that the maximal admissible effective dimension $k^{\max}$ defined in Remark 4 and the target excess risk bound $\{R(k_{best})\}_{i=1}^m$ defined in Remark 6. ASGD can reach optimality over $\mathcal{P}(W_{\mathbf{M}}, \mathbf{S}, \mathbf{T})$, if there exists $k_i \leq k^{\max}$ such that*

$$
R(k_i) \approx \max_{\substack{\mathsf{A} \subseteq \{1, \ldots, k_1\}: \\ \sum_{i \in \mathsf{A}} \frac{1}{n \lambda_i} \leq 1}} \left\langle \mathbf{T}', \left( \mathbf{S}'_{\mathsf{A}} \right)^{-1} \right\rangle + \sup_{\substack{\mathbf{F} \succeq \mathbf{O}, \\ \|\mathbf{F}\|_* \leq 1}} \left\langle \mathbf{T}', \left( \mathbf{F}^{-1} + n \mathbf{S}'_{k_1+1:d} \right)^{-1} \right\rangle. \tag{11}
$$

Under the condition in (11), ASGD with step sizes $\gamma^{k_i}$ and $\delta^{k_i}$ defined in Remark 6 can reach optimality. The first term on the right-hand side corresponds to the necessary variance incurred

when accurately estimating the ground truth within eigenspace $\mathbf{S}_{1:k_1}$. The second term captures the unavoidable bias in the tail eigenspace $\mathbf{S}_{k_1:d}$. This aligns with the fact that ASGD proceeds greedily along the eigendirections of $\mathbf{S}$ and achieves the best performance over the problem class $\mathcal{P}(W_{\mathbf{M}}, \mathbf{S}, \mathbf{T})$ when the best step size is properly chosen to strike the bias–variance trade-off.

As the corollaries of Theorem 8, ASGD achieves optimality across a broad common scenarios. We first demonstrate the optimality of ASGD under the case of under-parameterized setup in (Ge et al., 2024), where sample size $n$ is sufficiently large. SGD can attain an optimal target excess risk of order $\tilde{\mathcal{O}}\left(\operatorname{tr}\left(\mathbf{T}\mathbf{S}^{-1}\right)/n\right)$ even in the absence of target data information.

**Corollary 9.** *For any positive semi-definite matrix $\mathbf{S}$, $\mathbf{T}$, and $\mathbf{M}$, we get samples $\{(\mathbf{x}_i, y_i)\}_{i=1}^n$ drawn from the source distribution of $\tilde{P} \in \mathcal{P}(W_{\mathbf{M}}, \mathbf{S}, \mathbf{T})$. When $\frac{n}{\ln^2 n} > \max\left\{\lambda_d^{-1}, \left\|\mathbf{M}^{-\frac{1}{2}}\mathbf{T}\mathbf{M}^{-\frac{1}{2}}\right\|\right\}$, SGD with $\delta = \gamma = \tilde{\Theta}(1)$ can reach optimal rate $\tilde{\mathcal{O}}\left(\operatorname{tr}\left(\mathbf{T}\mathbf{S}^{-1}\right)/n\right)$.*

Theorem 8 also implies that SGD can achieve optimality under the $B$-bounded density ratio class, a widely-adopted problem class in the covariate shift literature (Cortes et al., 2010; Ma et al., 2023; Feng et al., 2023). The class $\mathcal{P}_{B,\mathbf{T}}$ includes all problems such that $\mathbb{E}_{Q_{\mathbf{x}}}\mathbf{x}\mathbf{x}^\top = \mathbf{T}$ and $dQ_{\mathbf{x}}/dP_{\mathbf{x}}^B \leq B$. We need two conditions in this setting: (1) The eigenvalues $\{\mu_i\}_{i=1}^d$ of $\mathbf{T}$ satisfy the standard regularity condition (Yang et al., 2017; Ma et al., 2023; Feng et al., 2023): for any $\delta > 0$, define $d(\delta) = \min\{j \geq 1 | \mu_j \leq \delta^2\}$, and assume that $\sum_{j=d(\delta)+1}^d \mu_j \leq Cd(\delta)\delta^2$ for some universal constant $C > 0$. (2) The bias–variance intersection point is admissible: Denote $d_1 = \max_i\left\{\mu_i \geq \frac{\sigma^2 Bi}{n}\right\}$ and $k_B^* = \max_k\left\{\lambda_k^B \geq \frac{\mu_{d_1}}{B}\right\}$, then $k_B^* \leq k^{\max}$.

**Corollary 10** ($B$-Bounded). *Under the above conditions and $\mathbf{M} = \mathbf{I}$, SGD with $\gamma \asymp (\ln n)^2/(n\lambda_{k_B^*})$ and $\delta \asymp \min\{\gamma, 1/(\operatorname{tr}\mathbf{S}\ln n)\}$ can achieve the optimal rate $\tilde{\mathcal{O}}\left(\inf_{\delta>0}\left\{\delta^2 + \frac{\sigma^2 Bd(\delta)}{n}\right\}\right)$.*

### 5.1 SEPARATIONS

We then establish separations between methods. First, we compare (A)SGD with the standard offline algorithm, ridge regression; second, we demonstrate the effectiveness of momentum by comparing ASGD to SGD. Specifically, we exemplify the learning problems that create the separations.

The separation between the (A)SGD and ridge can be understood through the preconditioning lens: they each correspond to a distinct **diagonal** precondition strategy. (A)SGD applies a sharp truncation via the precondition matrix $\operatorname{diag}\{\mathbf{I}_{k^*}, \mathbf{O}_{k^*:d}\}$ stated in Section 4.1, eliminating bias in the top-$k^*$ eigenspace of $\mathbf{S}$. By contrast, ridge regression with regularizer $\lambda$ corresponds to the smoother preconditioner $\mathbf{K}_\lambda = \frac{\mathbf{S}}{\mathbf{S}+\lambda\mathbf{I}}$, which leaves residual bias in the top eigenspace. While reducing $\lambda$ decreases bias, it simultaneously inflates variance, creating an unavoidable trade-off. The following example quantifies this separation (the ridge regression lower bound is from Tang et al. (2025)).

**Theorem 11.** *When $\mathbf{S} = \operatorname{diag}\left\{\mathbf{I}_k, \frac{1}{\sqrt{n}}\mathbf{I}_{k+1:\lfloor\sqrt{n}\rfloor}, \mathbf{O}_{\lfloor\sqrt{n}\rfloor:d}\right\}$, $\mathbf{T} = \mathbf{I}_d$, $\mathbf{w}^* = [\mathbf{1}_k, \mathbf{0}_{k+1:d}]^\top$, $k = \mathcal{O}(1)$, $\mathbf{M} = \operatorname{diag}\left\{\frac{1}{k}\mathbf{I}_k, \infty\mathbf{I}_{k+1:d}\right\}$, the excess risk of ridge regression for any $\lambda \geq 0$ is lower bounded by $\mathcal{O}(1/\sqrt{n})$, while (A)SGD with $\delta^{k_{best}} = \gamma^{k_{best}} = \frac{\ln^2 n}{n}$ achieves the optimal rate $\mathcal{O}(1/n)$.*

We demonstrate the separation of ASGD and vanilla SGD through single point prediction, one most standard covariate shift setting (Donoho, 1994; Box et al., 2015). We adopt the polynomially decaying spectral structure as considered in the seminal work Cai and Hall (2006).

**Corollary 12** (Single Point Prediction). *Consider $\lambda_i \asymp i^{-a}$, $\mathbf{M} = \mathbf{S}^{1-s}$ and $\mathbf{T} = \mathbf{w}\mathbf{w}^\top$ where $\mathbf{w} \in \mathbb{R}^d$ and $\mathbf{w}_i \asymp i^{-(1+r)a/2}$. We assume $(r+s)a \geq 1$ so that $\left\|\mathbf{M}^{-1/2}\mathbf{T}\mathbf{M}^{-1/2}\right\|$ is bounded. For region $s \geq 1$, vanilla SGD achieves optimality up to logarithmic factors; for region $1 > s > \frac{a}{2a-1}$, ASGD achieves optimality up to logarithmic factors. The optimal rate is*

$$\mathbb{E}_{\tilde{P}^{\otimes n}}\left\|\mathbf{w}_n^{\mathrm{SGD}} - \mathbf{w}^*\right\|_{\mathbf{T}}^2 \leq \begin{cases} \tilde{\mathcal{O}}(1/n), & r \geq 1/a; \\ \tilde{\mathcal{O}}\left((1/n)^{\frac{(r+s)a-1}{sa}}\right), & r < 1/a. \end{cases} \tag{12}$$

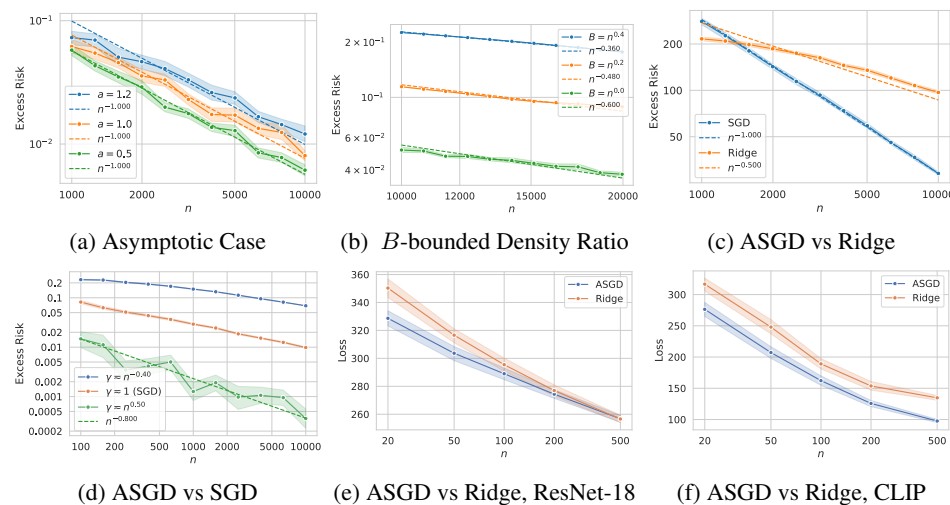

Figure 1: Experimental results: 95% confidence intervals over 100 repeats are shown in the shaded region. Simulation studies (a-d) verify the results in Section 5. Figures (e,f) demonstrate the separation of ASGD over ridge regression on the real-world dataset. Experimental details are in Appendix F.

Corollary 12 indicates adding momentum allows ASGD to achieve optimality over a broader range of smoothness parameters $s$. In particular, ASGD attain optimality for smaller $s$, which correspond to the less smooth ground truth in the interpolation space $[\mathcal{H}]^s$, the problem class with large initial bias.

## 6 EXPERIMENTS

This section presents both the simulated results and the experiments on real-world dataset UTK-Face (Zhang et al., 2017). Dashed lines show the theoretical rate if applicable. The experiment detail is in Appendix F.

Fig 1(a) validates Corollary 9 under the under-parameterized setting ($d = 10$) across different eigenstructures of $\mathbf{S}$. Furthermore, a smaller value of $a$ will yield a larger admissible $\lambda_k$, as defined in Remark 4, thus decreasing the excess risk as illustrated in the figure. Fig 1(b) validates the results of Theorem 10, where escalating scales of $B$ enlarges the discrepancy between two domains and degrades the performance of ASGD. Fig 1(c) illustrates the example in Theorem 11: when the target emphasizes larger eigenvalues, ridge regression with an optimally tuned $\lambda$ still exhibits worse performance across sample sizes $n$, regardless of the $\lambda$ chosen. In Fig 1(d), we examine the single-point prediction problem with large initial bias considered in Corollary 12 with various parameter choices. $\gamma \asymp n^{0.5}$ yields the optimal setting, and the case $\gamma \asymp 1$ reduces to vanilla SGD since we set $\delta \asymp 1$. The results show that ASGD achieves a clear separation from SGD under large initial bias in this setting.

We further evaluate the separation between ASGD and ridge regression on UTKFace dataset (Zhang et al., 2017) and extract features using ResNet-18 (He et al., 2016) and CLIP-ViT-L/14 (Radford et al., 2021). We compare SGD-type algorithms with the ridge regression using the features. We train on the source domain with $n$ data points and perform a grid search on the hyperparameters for all algorithms. As shown in Fig 1 (e, f), SGD methods can consistently outperform ridge regression in this problem, despite the optimally tuned $\lambda$.

## 7 CONCLUSION

This work theoretically characterizes ASGD's OOD generalization under covariate shift in linear regression. We derive excess risk bounds for SGD with momentum and step decay schedule. By viewing ASGD as a preconditioned estimator, we provide a new perspective to identify problems where ASGD is provably optimal and illustrate the separation between several methods.

## 8  ETHICS STATEMENT

Our paper complies with the ICLR Code of Ethics.

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

CONTENTS

# A  PROOFS OF ASGD UPPER BOUND IN SECTION 4 AND SECTION 4.1

In this section, we provide the analysis of ASGD upper bound. The organization of this section is as follows:

- In Section A.1, we present the tools for analyzing ASGD, and provide parameter choice of ASGD hyperparameters in Section A.1.3. Bias-variance decomposition is used to decompose the excess risk into the bias part and variance part. The definition of linear operators on matrices allows us to write the matrix form of the iteration of bias and variance.

- In Section A.2, we summarize the proof. We begin by defining $\tilde{\mathbf{C}}_t$ and $\tilde{\mathbf{B}}_t$, a different version of variance $\mathbf{C}_t$ and bias iteration $\mathbf{B}_t$, and further bounds the difference. Some proofs are deferred to Section A.7 and Section A.8.

- In Section A.3, we prove Theorem 6, and in Section A.4, we present an instance-dependent target excess risk upper bound of ASGD in Theorem 13.

- In Section A.5, we show that ASGD algorithm can be viewed as a preconditioned estimator by proving Theorem 7. We also prove Theorem 11 to show that ASGD is superior to ridge regression.

- In Section A.6, we establish the bounds of the momentum matrix. The bounds are based on the spectral radius of the momentum matrix.

- Section A.7 and Section A.8 provide bounds of semi-stochastic iterations in terms of algorithmic parameters, and the covariance matrix of source and target distributions.

## A.1  PRELIMINARIES

### A.1.1  BIAS-VARIANCE DECOMPOSITION

Given a sequence of data $\{(\mathbf{x}_i, y_i)\}_{i=1}^n$, ASGD starts from initial weight $\mathbf{w}_0 = \mathbf{v}_0$ and recursively calculates

$$\mathbf{u}_{t-1} \leftarrow \alpha \mathbf{w}_{t-1} + (1-\alpha)\mathbf{v}_{t-1}, \tag{13}$$

$$\mathbf{w}_t \leftarrow \mathbf{u}_{t-1} - \delta_t \left(\mathbf{x}_t^\top \mathbf{u}_{t-1} - y_t\right) \mathbf{x}_t, \tag{14}$$

$$\mathbf{v}_t \leftarrow \beta \mathbf{u}_{t-1} + (1-\beta)\mathbf{v}_{t-1} - \gamma_t \left(\mathbf{x}_t^\top \mathbf{u}_{t-1} - y_t\right) \mathbf{x}_t, \tag{15}$$

where $\delta_t$ and $\gamma_t$ are step sizes at iteration $t$. We consider the exponentially decaying step-size schedule

$$\delta_t = \delta/4^{\ell-1}, \gamma_t = \gamma/4^{\ell-1}, \text{if } K(\ell-1)+1 \le t \le K\ell, \tag{16}$$

where $n$ is the number of observations and $K = n/\log_2 n$. For theoretical analysis, we define $\boldsymbol{\eta}_t = \begin{bmatrix} \mathbf{w}_t - \mathbf{w}^* \\ \mathbf{u}_t - \mathbf{w}^* \end{bmatrix}$, where $\mathbf{w}^*$ is the ground-truth weight. Let $c = \alpha(1-\beta)$, $q = \alpha\delta + (1-\alpha)\gamma$, and $q_t = \alpha\delta_t + (1-\alpha)\gamma_t$, by eliminating $\mathbf{v}_t$ in (15), ASGD iteration can be written in the following compact form,

$$\boldsymbol{\eta}_t = \widehat{\mathbf{A}}_t \boldsymbol{\eta}_{t-1} + \boldsymbol{\zeta}_t, \quad \text{where } \widehat{\mathbf{A}}_t = \begin{bmatrix} \mathbf{O} & \mathbf{I} - \delta_t \mathbf{x}_t \mathbf{x}_t^\top \\ -c\mathbf{I} & (1+c)\mathbf{I} - q_t \mathbf{x}_t \mathbf{x}_t^\top \end{bmatrix}, \boldsymbol{\zeta}_t = \begin{bmatrix} \delta_t \epsilon_t \mathbf{x}_t \\ q_t \epsilon_t \mathbf{x}_t \end{bmatrix}, \tag{17}$$

where $\epsilon_t$ is defined in Assumption 1.

Following the standard bias-variance decomposition technique (Jain et al., 2017; Wu et al., 2022a; Li et al., 2024), we decompose the iteration $\boldsymbol{\eta}_t$ into the bias component $\boldsymbol{\eta}_t^{\text{bias}}$ and the variance component $\boldsymbol{\eta}_t^{\text{var}}$,

$$\boldsymbol{\eta}_t^{\text{bias}} = \widehat{\mathbf{A}}_t \boldsymbol{\eta}_{t-1}^{\text{bias}}, \quad \boldsymbol{\eta}_0^{\text{bias}} = \boldsymbol{\eta}_0; \tag{18}$$

$$\boldsymbol{\eta}_t^{\text{var}} = \widehat{\mathbf{A}}_t \boldsymbol{\eta}_{t-1}^{\text{var}} + \boldsymbol{\zeta}_t, \quad \boldsymbol{\eta}_0^{\text{var}} = \mathbf{0}. \tag{19}$$

The decomposition of $\boldsymbol{\eta}_t$ induces the decomposition of excess risk:

$$\mathbb{E} \|\mathbf{w}_n - \mathbf{w}^*\|_{\mathbf{T}}^2 = \left\langle \begin{bmatrix} \mathbf{T} & \mathbf{O} \\ \mathbf{O} & \mathbf{O} \end{bmatrix}, \mathbb{E}\left[\boldsymbol{\eta}_n \boldsymbol{\eta}_n^\top\right] \right\rangle$$

$$\le 2 \cdot \underbrace{\left\langle \begin{bmatrix} \mathbf{T} & \mathbf{O} \\ \mathbf{O} & \mathbf{O} \end{bmatrix}, \mathbb{E}\left[\boldsymbol{\eta}_n^{\text{bias}} \left(\boldsymbol{\eta}_n^{\text{bias}}\right)^\top\right] \right\rangle}_{\text{Bias}} + 2 \cdot \underbrace{\left\langle \begin{bmatrix} \mathbf{T} & \mathbf{O} \\ \mathbf{O} & \mathbf{O} \end{bmatrix}, \mathbb{E}\left[\boldsymbol{\eta}_n^{\text{var}} \left(\boldsymbol{\eta}_n^{\text{var}}\right)^\top\right] \right\rangle}_{\text{Variance}}. \tag{20}$$

### A.1.2 LINEAR OPERATORS

We introduce the following linear operators on matrices to analyze the recursion of $\mathbb{E}\left[\boldsymbol{\eta}_n^{\text{bias}}\left(\boldsymbol{\eta}_n^{\text{bias}}\right)^\top\right]$ and $\mathbb{E}\left[\boldsymbol{\eta}_n^{\text{var}}\left(\boldsymbol{\eta}_n^{\text{var}}\right)^\top\right]$.

$$\mathcal{I} = \mathbf{I} \otimes \mathbf{I}, \quad \mathcal{B}_t = \mathbb{E}\left[\widehat{\mathbf{A}}_t \otimes \widehat{\mathbf{A}}_t\right]. \tag{21}$$

$$\tag{22}$$

Let $\mathbf{A}_t = \mathbb{E}\widehat{\mathbf{A}}_t$ be the deterministic version of $\widehat{\mathbf{A}}_t$, and define

$$\tilde{\mathcal{B}}_t = \mathbf{A}_t \otimes \mathbf{A}_t. \tag{23}$$

We decompose $\widehat{\mathbf{A}}_t$ into two components:

$$\mathbf{V} = \begin{bmatrix} \mathbf{O} & \mathbf{I} \\ -c\mathbf{I} & (1+c)\mathbf{I} \end{bmatrix}, \quad \widehat{\mathbf{G}}_t = \begin{bmatrix} \mathbf{O} & \delta_t \mathbf{x}_t \mathbf{x}_t^\top \\ \mathbf{O} & q_t \mathbf{x}_t \mathbf{x}_t^\top \end{bmatrix}. \tag{24}$$

The deterministic version of $\widehat{\mathbf{G}}_t$ is defined as

$$\mathbf{G}_t = \mathbb{E}\widehat{\mathbf{G}}_t. \tag{25}$$

Therefore, $\widehat{\mathbf{A}}_t = \mathbf{V} - \widehat{\mathbf{G}}_t$ and $\mathbf{A}_t = \mathbf{V} - \mathbf{G}_t$.

The following lemma provides properties of the linear operators.

**Lemma 1.** *The above operators have the following properties:*

*1.* $\mathcal{B}_t \preceq \tilde{\mathcal{B}}_t + \mathbb{E}\left[\widehat{\mathbf{G}}_t \otimes \widehat{\mathbf{G}}_t\right]$.

*2. Suppose Assumption 4 holds. For any PSD matrix $\mathbf{M}$, we have*

$$\mathbb{E}\left[\widehat{\mathbf{G}}_t \otimes \widehat{\mathbf{G}}_t\right] \circ \mathbf{M} \preceq \psi \left\langle \begin{bmatrix} \mathbf{O} & \mathbf{O} \\ \mathbf{O} & \mathbf{S} \end{bmatrix}, \mathbf{M} \right\rangle \begin{bmatrix} \delta_t^2 \mathbf{S} & \delta_t q_t \mathbf{S} \\ \delta_t q_t \mathbf{S} & q_t^2 \mathbf{S} \end{bmatrix}. \tag{26}$$

*Proof.* 1. From the definiton of $\mathcal{B}_t$, we have

$$\begin{aligned} \mathcal{B}_t &= \mathbb{E}\left[\left(\mathbf{V} - \widehat{\mathbf{G}}_t\right) \otimes \left(\mathbf{V} - \widehat{\mathbf{G}}_t\right)\right] \\ &\overset{a}{\preceq} (\mathbf{V} - \mathbf{G}_t) \otimes (\mathbf{V} - \mathbf{G}_t) - \mathbf{G}_t \otimes \mathbf{G}_t + \mathbb{E}\left[\widehat{\mathbf{G}}_t \otimes \widehat{\mathbf{G}}_t\right] \\ &= \tilde{\mathcal{B}}_t + \mathbb{E}\left[\widehat{\mathbf{G}}_t \otimes \widehat{\mathbf{G}}_t\right], \end{aligned} \tag{27}$$

where $\overset{a}{\preceq}$ uses $\mathbb{E}\left[\mathbf{V} \otimes \widehat{\mathbf{G}}_t\right] = \mathbf{V} \otimes \mathbf{G}_t$ and $\overset{a}{\preceq}$ uses $\mathbb{E}\left[\widehat{\mathbf{G}}_t \otimes \mathbf{V}\right] = \mathbf{G}_t \otimes \mathbf{V}$.

2. Apply the partition of $\widehat{\mathbf{G}}_t$ to $\mathbf{M} = \begin{bmatrix} \mathbf{M}_{11} & \mathbf{M}_{12} \\ \mathbf{M}_{21} & \mathbf{M}_{22} \end{bmatrix}$, we have

$$\begin{aligned} \mathbb{E}\left[\widehat{\mathbf{G}}_t \otimes \widehat{\mathbf{G}}_t\right] \circ \mathbf{M} &= \mathbb{E}\begin{bmatrix} \delta_t^2 \mathbf{x}\mathbf{x}^\top \mathbf{M}_{22} \mathbf{x}\mathbf{x}^\top & \delta_t q_t \mathbf{x}\mathbf{x}^\top \mathbf{M}_{22} \mathbf{x}\mathbf{x}^\top \\ \delta_t q_t \mathbf{x}\mathbf{x}^\top \mathbf{M}_{22} \mathbf{x}\mathbf{x}^\top & q_t^2 \mathbf{x}\mathbf{x}^\top \mathbf{M}_{22} \mathbf{x}\mathbf{x}^\top \end{bmatrix} \\ &= \begin{bmatrix} \delta_t^2 & \delta_t q_t \\ \delta_t q_t & q_t^2 \end{bmatrix} \odot \mathbb{E}\left[\mathbf{x}\mathbf{x}^\top \mathbf{M}_{22} \mathbf{x}\mathbf{x}^\top\right] \\ &\overset{a}{\preceq} \begin{bmatrix} \delta_t^2 & \delta_t q_t \\ \delta_t q_t & q_t^2 \end{bmatrix} \odot \left[\psi \left\langle \begin{bmatrix} \mathbf{O} & \mathbf{O} \\ \mathbf{O} & \mathbf{S} \end{bmatrix}, \mathbf{M} \right\rangle \mathbf{S}\right] \\ &\preceq \psi \left\langle \begin{bmatrix} \mathbf{O} & \mathbf{O} \\ \mathbf{O} & \mathbf{S} \end{bmatrix}, \mathbf{M} \right\rangle \begin{bmatrix} \delta_t^2 \mathbf{S} & \delta_t q_t \mathbf{S} \\ \delta_t q_t \mathbf{S} & q_t^2 \mathbf{S} \end{bmatrix}, \end{aligned} \tag{28}$$

where $\odot$ denotes Kronecker product, and $\overset{a}{\preceq}$ holds for Assumption 4 and property of Kronecker product, which is, for any PSD matrices $\mathbf{A}, \mathbf{B} \preceq \mathbf{C}$, we have $\mathbf{A} \odot \mathbf{B} \preceq \mathbf{A} \odot \mathbf{C}$.

$\square$

### A.1.3 PARAMETER CHOICE

This section provides a specific parameter choice procedure. We first choose appropriate positive integer $\tilde{\kappa}$, fix $\alpha = 1/(1 + \beta)$, and choose

$$\delta \in \left[ \frac{64\tilde{\kappa} \ln n \log_2 n}{n \sum_{i > \tilde{\kappa}} \lambda_i}, \frac{1}{2188\psi \operatorname{tr} \mathbf{S} \ln n} \right], \quad \gamma \in \left[ \delta, \frac{1}{2188\psi \sum_{i > \tilde{\kappa}} \lambda_i} \right], \quad \beta = \frac{\delta}{4376\psi\tilde{\kappa}\gamma \ln n}.$$

From the above procedure, we have

$$\frac{n \left[1 - \alpha(1 - \beta)\right]}{\log_2 n \ln n} \geq 16. \tag{29}$$

**Lemma 2.** *Recall that $c = \alpha(1 - \beta)$, $q = \alpha\delta + (1 - \alpha)\gamma$ and $K = n/\log_2 n$. We have the following properties of the parameter choice.*

   *1. We have*

$$\frac{q - \delta}{1 - c} = \frac{\gamma - \delta}{2}, \quad \frac{q - c\delta}{1 - c} = \frac{\gamma + \delta}{2}. \tag{30}$$

   *2. For $i \in [d]$, we have*

$$\delta\lambda_i \leq \frac{1}{2188\psi \ln n} \leq 1, \quad q\lambda_i \leq 1 + c. \tag{31}$$

*Proof.*    1. Note that $1 - c = 2(1 - \alpha)$. Thus, we have

$$\frac{q - \delta}{1 - c} = \frac{(1 - \alpha)(\gamma - \delta)}{1 - c} = \frac{\gamma - \delta}{2}, \quad \frac{q - c\delta}{1 - c} = \frac{q - \delta}{1 - c} + \delta = \frac{\gamma + \delta}{2}. \tag{32}$$

   2. Since $\lambda_i \leq \operatorname{tr} \mathbf{S}$, we have

$$\delta\lambda_i \leq \frac{\lambda_i}{2188\psi \ln n \operatorname{tr} \mathbf{S}} \leq \frac{1}{2188\psi \ln n} \leq 1. \tag{33}$$

Note that $1 - \alpha = \alpha\beta$ and $2\alpha = 1 + c$, we have

$$q = \alpha\delta + (1 - \alpha)\gamma = \alpha\delta + \alpha\beta\gamma \leq 2\alpha\delta = (1 + c)\delta. \tag{34}$$

Therefore, $q\lambda_i \leq (1 + c)\delta\lambda_i \leq 1 + c$.

$\square$

### A.2 PROOF OUTLINE

We express the recursions of $\mathbb{E}\left[\boldsymbol{\eta}_t^{\text{bias}} \left(\boldsymbol{\eta}_t^{\text{bias}}\right)^{\top}\right]$ and $\mathbb{E}\left[\boldsymbol{\eta}_t^{\text{var}} \left(\boldsymbol{\eta}_t^{\text{var}}\right)^{\top}\right]$ using the operators:

$$\mathbb{E}\left[\boldsymbol{\eta}_t^{\text{bias}} \left(\boldsymbol{\eta}_t^{\text{bias}}\right)^{\top}\right] = \mathcal{B}_t \circ \mathbb{E}\left[\boldsymbol{\eta}_{t-1}^{\text{bias}} \left(\boldsymbol{\eta}_{t-1}^{\text{bias}}\right)^{\top}\right], \quad \mathbb{E}\left[\boldsymbol{\eta}_0^{\text{bias}} \left(\boldsymbol{\eta}_0^{\text{bias}}\right)^{\top}\right] = \boldsymbol{\eta}_0 \boldsymbol{\eta}_0^{\top}; \tag{35}$$

$$\mathbb{E}\left[\boldsymbol{\eta}_t^{\text{var}} \left(\boldsymbol{\eta}_t^{\text{var}}\right)^{\top}\right] = \mathcal{B}_t \circ \mathbb{E}\left[\boldsymbol{\eta}_{t-1}^{\text{var}} \left(\boldsymbol{\eta}_{t-1}^{\text{var}}\right)^{\top}\right] + \mathbb{E}\left[\boldsymbol{\zeta}_t \boldsymbol{\zeta}_t^{\top}\right], \quad \mathbb{E}\left[\boldsymbol{\eta}_0^{\text{var}} \left(\boldsymbol{\eta}_0^{\text{var}}\right)^{\top}\right] = \mathbf{O}. \tag{36}$$

Then we construct two recursions similar to the above update rule:

$$\mathbf{B}_t = \mathcal{B} \circ \mathbf{B}_{t-1}, \quad \mathbf{B}_0 = \boldsymbol{\eta}_0 \boldsymbol{\eta}_0^{\top}, \tag{37}$$

$$\mathbf{C}_t = \mathcal{B}_t \circ \mathbf{C}_{t-1} + \sigma^2 \begin{bmatrix} \delta_t^2 \mathbf{S} & \delta_t q_t \mathbf{S} \\ \delta_t q_t \mathbf{S} & q_t^2 \mathbf{S} \end{bmatrix}, \quad \mathbf{C}_0 = \mathbf{O}. \tag{38}$$

The following lemma characterizes $\mathbb{E}\left[\boldsymbol{\eta}_t^{\text{bias}} \left(\boldsymbol{\eta}_t^{\text{bias}}\right)^{\top}\right]$ and $\mathbb{E}\left[\boldsymbol{\eta}_t^{\text{var}} \left(\boldsymbol{\eta}_t^{\text{var}}\right)^{\top}\right]$ by $\mathbf{B}_t$ and $\mathbf{C}_t$, respectively.

**Lemma 3.** *For $0 \leq t \leq n$, $\mathbb{E}\left[\boldsymbol{\eta}_t^{bias} \left(\boldsymbol{\eta}_t^{bias}\right)^{\top}\right] = \mathbf{B}_t$. Furthermore, under Assumption 1, we have $\mathbb{E}\left[\boldsymbol{\eta}_t^{var} \left(\boldsymbol{\eta}_t^{var}\right)^{\top}\right] \preceq \mathbf{C}_t$.*

*Proof.* From (35), the recursion of $\mathbf{B}_t$ is identical to the recursion of $\mathbb{E}\left[\boldsymbol{\eta}_t^{\text{bias}}\left(\boldsymbol{\eta}_t^{\text{bias}}\right)^\top\right]$. This proves the first part of the lemma. For the second part, from (36), we know the conclusion holds for $t = 0$. We assume that $\mathbb{E}\left[\boldsymbol{\eta}_{t-1}^{\text{var}}\left(\boldsymbol{\eta}_{t-1}^{\text{var}}\right)^\top\right] \preceq \mathbf{C}_{t-1}$, then

$$
\begin{aligned}
\mathbb{E}\left[\boldsymbol{\eta}_t^{\text{var}}\left(\boldsymbol{\eta}_t^{\text{var}}\right)^\top\right] &= \mathcal{B} \circ \mathbb{E}\left[\boldsymbol{\eta}_{t-1}^{\text{var}}\left(\boldsymbol{\eta}_{t-1}^{\text{var}}\right)^\top\right] + \mathbb{E}\left[\boldsymbol{\zeta}_t\boldsymbol{\zeta}_t^\top\right] \\
&\preceq \mathcal{B} \circ \mathbf{C}_{t-1} + \mathbb{E}\left[\boldsymbol{\zeta}_t\boldsymbol{\zeta}_t^\top\right] \\
&\overset{a}{\preceq} \mathcal{B} \circ \mathbf{C}_{t-1} + \sigma^2 \begin{bmatrix} \delta_t^2\mathbf{S} & \delta_t q_t\mathbf{S} \\ \delta_t q_t\mathbf{S} & q_t^2\mathbf{S} \end{bmatrix} \\
&= \mathbf{C}_t,
\end{aligned}
\tag{39}
$$

where $\overset{a}{\preceq}$ holds because Assumption 1 implies $\mathbb{E}\left[\epsilon_t^2\mathbf{x}_t\mathbf{x}_t^\top\right] \preceq \sigma^2\mathbf{S}$, and

$$
\begin{aligned}
\mathbb{E}\left[\boldsymbol{\zeta}_t\boldsymbol{\zeta}_t^\top\right] &= \mathbb{E}\left[\begin{bmatrix} \delta_t^2\epsilon_t^2\mathbf{x}_t\mathbf{x}_t^\top & \delta_t q_t\epsilon_t^2\mathbf{x}_t\mathbf{x}_t^\top \\ \delta_t q_t\epsilon_t^2\mathbf{x}_t\mathbf{x}_t^\top & q_t^2\epsilon_t^2\mathbf{x}_t\mathbf{x}_t^\top \end{bmatrix}\right] = \begin{bmatrix} \delta_t^2 & \delta_t q_t \\ \delta_t q_t & q_t^2 \end{bmatrix} \odot \mathbb{E}\left[\epsilon_t^2\mathbf{x}_t\mathbf{x}_t^\top\right] \\
&\overset{a}{\preceq} \begin{bmatrix} \delta_t^2 & \delta_t q_t \\ \delta_t q_t & q_t^2 \end{bmatrix} \odot \sigma^2\mathbf{S} = \sigma^2 \begin{bmatrix} \delta_t^2\mathbf{S} & \delta_t q_t\mathbf{S} \\ \delta_t q_t\mathbf{S} & q_t^2\mathbf{S} \end{bmatrix},
\end{aligned}
\tag{40}
$$

where $\odot$ denotes Kronecker product, and $\overset{a}{\preceq}$ holds because for any PSD matrices $\mathbf{A}$, $\mathbf{B} \preceq \mathbf{C}$, we have $\mathbf{A} \odot \mathbf{B} \preceq \mathbf{A} \odot \mathbf{C}$. $\qquad\square$

With Lemma 3, we have $\mathbb{E}\left[\boldsymbol{\eta}_n^{\text{bias}}\left(\boldsymbol{\eta}_n^{\text{bias}}\right)^\top\right] = \mathbf{B}_n$ and $\mathbb{E}\left[\boldsymbol{\eta}_n^{\text{var}}\left(\boldsymbol{\eta}_n^{\text{var}}\right)^\top\right] \preceq \mathbf{C}_n$. Thus,

$$
\text{Bias} \leq \left\langle \tilde{\mathbf{T}}, \mathbf{B}_n \right\rangle, \quad \text{Variance} \leq \left\langle \tilde{\mathbf{T}}, \mathbf{C}_n \right\rangle,
\tag{41}
$$

where $\tilde{\mathbf{T}} = \begin{bmatrix} \mathbf{T} & \mathbf{O} \\ \mathbf{O} & \mathbf{O} \end{bmatrix}$.

The main technical challenge to directly bound $\mathbf{B}_n$ and $\mathbf{C}_n$ originates from the effect of the fourth moment (*i.e.* $\mathcal{B} \neq \tilde{\mathcal{B}}$), which prevents us from analyzing $\mathbf{B}_t$ in each eigenspace of $\mathbf{S}$. Our proof defines the semi-stochasitc iteration $\tilde{\boldsymbol{\eta}}_t^{\text{bias}}$ and $\tilde{\boldsymbol{\eta}}_t^{\text{var}}$ following Dieuleveut and Bach (2016). We analyzes two new recursions $\tilde{\mathbf{B}}_t$ and $\tilde{\mathbf{C}}_t$ induced by $\tilde{\boldsymbol{\eta}}_t^{\text{bias}}$ and $\tilde{\boldsymbol{\eta}}_t^{\text{var}}$. For the variance component, we establish a uniform bound on $\tilde{\mathbf{C}}_t$ to show that the effect of the fourth moment is actually "self-governed". Specifically, the fourth moment amplifies the excess risk up to a constant. For the bias component, $\mathbf{B}_t$ is decomposed into $\tilde{\mathbf{B}}_t$ and a new term $\mathbf{B}_t^{(1)}$ which resembles $\mathbf{C}_t$. The bound of $\mathbf{B}_t^{(1)}$ is established by applying the bound of $\mathbf{C}_t$.

### A.2.1 Variance Upper Bound

We start with the construction of $\tilde{\boldsymbol{\eta}}_t$ by replacing $\widehat{\mathbf{A}}_t$ by $\mathbf{A}_t$:

$$
\tilde{\boldsymbol{\eta}}_t^{\text{var}} = \mathbf{A}_t\tilde{\boldsymbol{\eta}}_{t-1}^{\text{var}} + \boldsymbol{\zeta}_t, \quad \tilde{\boldsymbol{\eta}}_0^{\text{var}} = \mathbf{0}.
\tag{42}
$$

From this definition, we have $\mathbb{E}\left[\tilde{\boldsymbol{\eta}}_0^{\text{var}}\left(\tilde{\boldsymbol{\eta}}_0^{\text{var}}\right)^\top\right] = \mathbf{O}$ and

$$
\begin{aligned}
\mathbb{E}\left[\tilde{\boldsymbol{\eta}}_t^{\text{var}}\left(\tilde{\boldsymbol{\eta}}_t^{\text{var}}\right)^\top\right] &= \tilde{\mathcal{B}}_t \circ \mathbb{E}\left[\tilde{\boldsymbol{\eta}}_{t-1}^{\text{var}}\left(\tilde{\boldsymbol{\eta}}_{t-1}^{\text{var}}\right)^\top\right] + \mathbb{E}\left[\boldsymbol{\zeta}_t\boldsymbol{\zeta}_t^\top\right] \\
&\preceq \tilde{\mathcal{B}}_t \circ \mathbb{E}\left[\tilde{\boldsymbol{\eta}}_{t-1}^{\text{var}}\left(\tilde{\boldsymbol{\eta}}_{t-1}^{\text{var}}\right)^\top\right] + \sigma^2 \begin{bmatrix} \delta_t^2\mathbf{S} & \delta_t q_t\mathbf{S} \\ \delta_t q_t\mathbf{S} & q_t^2\mathbf{S} \end{bmatrix}.
\end{aligned}
\tag{43}
$$

Therefore, we define $\tilde{\mathbf{C}}_t$ as

$$
\tilde{\mathbf{C}}_t = \tilde{\mathcal{B}}_t \circ \tilde{\mathbf{C}}_{t-1} + \sigma^2 \begin{bmatrix} \delta_t^2\mathbf{S} & \delta_t q_t\mathbf{S} \\ \delta_t q_t\mathbf{S} & q_t^2\mathbf{S} \end{bmatrix}, \quad \tilde{\mathbf{C}}_0 = \mathbf{O}.
\tag{44}
$$

By induction, we have $\mathbb{E}\left[\tilde{\boldsymbol{\eta}}_t^{\text{var}}\left(\tilde{\boldsymbol{\eta}}_t^{\text{var}}\right)^\top\right] \preceq \tilde{\mathbf{C}}_t$.

The following lemma characterizes $\left\langle \tilde{\mathbf{T}}, \tilde{\mathbf{C}}_n \right\rangle$, which is the first step of our proof.

**Lemma 4.** *We have*

$$\left\langle \tilde{\mathbf{T}}, \tilde{\mathbf{C}}_n \right\rangle \leq \sigma^2 \left[ \sum_{i=1}^{k^*} \frac{t_{ii}}{2K\lambda_i} + \frac{128}{15} K \left( \frac{q - c\delta}{1 - c} \right)^2 \sum_{i=k^*+1}^{d} \lambda_i t_{ii} \right]. \tag{45}$$

The second step is to understand the effect of the fourth moment on the variance component. We first construct an auxiliary recursion $\mathbf{C}_t^{(1)}$ as

$$\mathbf{C}_t^{(1)} = \mathcal{B}_t \circ \mathbf{C}_{t-1}^{(1)} + \mathbb{E}\left[ \widehat{\mathbf{G}}_t \otimes \widehat{\mathbf{G}}_t \right] \circ \tilde{\mathbf{C}}_{t-1}, \quad \mathbf{C}_0^{(1)} = \mathbf{O}. \tag{46}$$

The following lemma bounds $\mathbf{C}_t$ from above.

**Lemma 5.** *For $0 \leq t \leq n$, we have $\mathbf{C}_t \preceq \tilde{\mathbf{C}}_t + \mathbf{C}_t^{(1)}$.*

*Proof.* We prove the conclusion by induction. By definition, we have $\mathbf{C}_0 = \tilde{\mathbf{C}}_0 = \mathbf{C}_t^{(1)} = \mathbf{O}$. Therefore, the conclusion holds for $t = 0$. We assume $\mathbf{C}_{t-1} \preceq \tilde{\mathbf{C}}_{t-1} + \mathbf{C}_{t-1}^{(1)}$. Note that

$$
\begin{aligned}
\mathbf{C}_t ={} & \mathcal{B}_t \circ \mathbf{C}_{t-1} + \sigma^2 \begin{bmatrix} \delta_t^2 \mathbf{S} & \delta_t q_t \mathbf{S} \\ \delta_t q_t \mathbf{S} & q_t^2 \mathbf{S} \end{bmatrix} \\
\preceq {} & \mathcal{B}_t \circ \left( \tilde{\mathbf{C}}_{t-1} + \mathbf{C}_{t-1}^{(1)} \right) + \sigma^2 \begin{bmatrix} \delta_t^2 \mathbf{S} & \delta_t q_t \mathbf{S} \\ \delta_t q_t \mathbf{S} & q_t^2 \mathbf{S} \end{bmatrix} \\
={} & \mathcal{B}_t \circ \tilde{\mathbf{C}}_{t-1} + \mathcal{B}_t \circ \mathbf{C}_{t-1}^{(1)} + \sigma^2 \begin{bmatrix} \delta_t^2 \mathbf{S} & \delta_t q_t \mathbf{S} \\ \delta_t q_t \mathbf{S} & q_t^2 \mathbf{S} \end{bmatrix} \\
\overset{a}{\preceq} {} & \tilde{\mathcal{B}}_t \circ \tilde{\mathbf{C}}_{t-1} + \mathbb{E}\left[ \widehat{\mathbf{G}}_t \otimes \widehat{\mathbf{G}}_t \right] \circ \tilde{\mathbf{C}}_{t-1} + \mathcal{B}_t \circ \mathbf{C}_{t-1}^{(1)} + \sigma^2 \begin{bmatrix} \delta_t^2 \mathbf{S} & \delta_t q_t \mathbf{S} \\ \delta_t q_t \mathbf{S} & q_t^2 \mathbf{S} \end{bmatrix} \\
={} & \tilde{\mathcal{B}}_t \circ \tilde{\mathbf{C}}_{t-1} + \sigma^2 \begin{bmatrix} \delta_t^2 \mathbf{S} & \delta_t q_t \mathbf{S} \\ \delta_t q_t \mathbf{S} & q_t^2 \mathbf{S} \end{bmatrix} + \mathcal{B}_t \circ \mathbf{C}_{t-1}^{(1)} + \mathbb{E}\left[ \widehat{\mathbf{G}}_t \otimes \widehat{\mathbf{G}}_t \right] \circ \tilde{\mathbf{C}}_{t-1} \\
={} & \tilde{\mathbf{C}}_t + \mathbf{C}_{t-1}^{(1)},
\end{aligned}
\tag{47}
$$

where $\overset{a}{\preceq}$ uses $\mathcal{B}_t \preceq \tilde{\mathcal{B}}_t + \mathbb{E}\left[ \widehat{\mathbf{G}}_t \otimes \widehat{\mathbf{G}}_t \right]$ in Lemma 1. $\qquad \square$

The following lemma characterizes the noise term $\mathbb{E}\left[ \widehat{\mathbf{G}}_t \otimes \widehat{\mathbf{G}}_t \right] \circ \tilde{\mathbf{C}}_{t-1}$.

**Lemma 6.** *Suppose Assumption 4 holds. Then for $1 \leq t \leq n$ we have*

$$\mathbb{E}\left[ \widehat{\mathbf{G}}_t \otimes \widehat{\mathbf{G}}_t \right] \circ \tilde{\mathbf{C}}_{t-1} \preceq \frac{1}{2} \sigma^2 \begin{bmatrix} \delta_t^2 \mathbf{S} & \delta_t q_t \mathbf{S} \\ \delta_t q_t \mathbf{S} & q_t^2 \mathbf{S} \end{bmatrix}. \tag{48}$$

Lemma 6 shows that the noise term in the recursion of $\mathbf{C}_t^{(1)}$ is uniformly less than that of $\mathbf{C}_t$, Therefore, we can show that $\mathbf{C}_t^{(1)} \preceq \frac{1}{2} \mathbf{C}_t$ for $0 \leq t \leq n$, which is the following lemma.

**Lemma 7.** *Suppose Assumption 4 holds. Then for $1 \leq t \leq n$ we have*

$$\mathbf{C}_t^{(1)} \preceq \frac{1}{2} \mathbf{C}_t. \tag{49}$$

*Proof.* We proceed by induction. For $t = 0$, the conclusion holds by the initial value of $\mathbf{C}_t$ and $\mathbf{C}_t^{(1)}$. We assume that $\mathbf{C}_{t-1}^{(1)} \preceq \frac{1}{2} \mathbf{C}_{t-1}$. By Lemma 6, we have

$$
\begin{aligned}
\mathbf{C}_t^{(1)} ={} & \mathcal{B} \circ \mathbf{C}_{t-1}^{(1)} + \mathbb{E}\left[ \widehat{\mathbf{G}}_t \otimes \widehat{\mathbf{G}}_t \right] \circ \tilde{\mathbf{C}}_{t-1} \\
\preceq {} & \mathcal{B} \circ \mathbf{C}_{t-1}^{(1)} + \frac{1}{2} \sigma^2 \begin{bmatrix} \delta_t^2 \mathbf{S} & \delta_t q_t \mathbf{S} \\ \delta_t q_t \mathbf{S} & q_t^2 \mathbf{S} \end{bmatrix} \\
\preceq {} & \mathcal{B} \circ \left( \frac{1}{2} \mathbf{C}_{t-1} \right) + \frac{1}{2} \sigma^2 \begin{bmatrix} \delta_t^2 \mathbf{S} & \delta_t q_t \mathbf{S} \\ \delta_t q_t \mathbf{S} & q_t^2 \mathbf{S} \end{bmatrix} \\
={} & \frac{1}{2} \left( \mathbf{C}_{t-1} + \sigma^2 \begin{bmatrix} \delta_t^2 \mathbf{S} & \delta_t q_t \mathbf{S} \\ \delta_t q_t \mathbf{S} & q_t^2 \mathbf{S} \end{bmatrix} \right) = \frac{1}{2} \mathbf{C}_t.
\end{aligned}
\tag{50}
$$

This completes the proof. $\qquad \square$

Finally, we show that $\mathbf{C}_t$ is "self-governed" and obtain the upper bound of variance.

**Lemma 8.** *Suppose Assumptions 4 and 1 hold. Then we have $\mathbf{C}_n \preceq 2\tilde{\mathbf{C}}_n$ and*

$$\text{Variance} \leq \sigma^2 \left[ \sum_{i=1}^{k^*} \frac{t_{ii}}{K\lambda_i} + \frac{256}{15} K \left( \frac{q - c\delta}{1 - c} \right)^2 \sum_{i=k^*+1}^{d} \lambda_i t_{ii} \right], \tag{51}$$

*where $k^* = \max \left\{ k : \lambda_k > \frac{16(1-c)\ln n}{(q-c\delta)K} \right\}$.*

*Proof.* We apply Lemma 5 and Lemma 7. For $0 \leq t \leq n$,

$$\mathbf{C}_t \preceq \tilde{\mathbf{C}}_t + \mathbf{C}_t^{(1)} \preceq \tilde{\mathbf{C}}_t + \frac{1}{2}\mathbf{C}_t. \tag{52}$$

Therefore, $\mathbf{C}_n \preceq 2\tilde{\mathbf{C}}_n$. By Lemma 4, taking the inner product with $\tilde{\mathbf{T}}$ yields

$$\text{Variance} \leq \left\langle \tilde{\mathbf{T}}, \mathbf{C}_n \right\rangle \leq 2 \left\langle \tilde{\mathbf{T}}, \tilde{\mathbf{C}}_n \right\rangle$$

$$\leq \sigma^2 \left[ \sum_{i=1}^{k^*} \frac{t_{ii}}{K\lambda_i} + \frac{256}{15} K \left( \frac{q - c\delta}{1 - c} \right)^2 \sum_{i=k^*+1}^{d} \lambda_i t_{ii} \right]. \tag{53}$$

$\square$

### A.2.2 BIAS UPPER BOUND

We follow the similar approach to construct $\tilde{\boldsymbol{\eta}}_t^{\text{bias}}$:

$$\tilde{\boldsymbol{\eta}}_t^{\text{bias}} = \mathbf{A}_t \tilde{\boldsymbol{\eta}}_{t-1}^{\text{bias}}, \quad \tilde{\boldsymbol{\eta}}_0^{\text{bias}} = \boldsymbol{\eta}_0. \tag{54}$$

Then we define $\tilde{\mathbf{B}}_t = \tilde{\mathcal{B}}_t \circ \tilde{\mathbf{B}}_{t-1}$. Therefore,

$$\tilde{\mathbf{B}}_t = \tilde{\mathcal{B}}_t \circ \tilde{\mathbf{B}}_{t-1}, \quad \tilde{\mathbf{B}}_0 = \boldsymbol{\eta}_0 \boldsymbol{\eta}_0^\top, \tag{55}$$

The first step is to characterize $\tilde{\mathbf{B}}_t$. The following lemma bound $\left\langle \tilde{\mathbf{T}}, \tilde{\mathbf{B}}_n \right\rangle$ from above.

**Lemma 9.** *With $\tilde{\mathbf{B}}_t$ defined in (55), we have*

$$\left\langle \tilde{\mathbf{T}}, \tilde{\mathbf{B}}_n \right\rangle \leq \max_{\mathbf{w} \in S(\mathbf{w}_0 - \mathbf{w}^*)} \frac{\|\mathbf{w}\|_{\mathbf{T}_{0:k^*}}^2}{8n^2 (\log_2 n)^4} + 4 \|\mathbf{w}\|_{\mathbf{T}_{k^*:\infty}}^2, \tag{56}$$

*where $k^* = \max \left\{ k : \lambda_k > \frac{16(1-c)\ln n}{(q-c\delta)K} \right\}$ and $S(\mathbf{w}_0 - \mathbf{w}^*) = \left\{ \mathbf{w} \in \mathbb{R}^d : |\mathbf{w}_i| \leq |(\mathbf{w}_0 - \mathbf{w}^*)_i| \right\}$.*

The second step is to bound $\mathbf{B}_t$ by $\tilde{\mathbf{B}}_t$. Define a new recursion $\mathbf{B}_t^{(1)}$ as follows:

$$\mathbf{B}_t^{(1)} = \mathcal{B}_t \circ \mathbf{B}_{t-1}^{(1)} + \mathbb{E}\left[ \widehat{\mathbf{G}}_t \otimes \widehat{\mathbf{G}}_t \right] \circ \tilde{\mathbf{B}}_{t-1}, \quad \mathbf{B}_0^{(1)} = \mathbf{O}. \tag{57}$$

The following lemma bounds $\mathbf{B}_t$ from above.

**Lemma 10.** *For $0 \leq t \leq n$, we have $\mathbf{B}_t \preceq \tilde{\mathbf{B}}_t + \mathbf{B}_t^{(1)}$.*

*Proof.* We prove the conclusion by induction. By definition, we have $\mathbf{B}_0 = \tilde{\mathbf{B}}_0 = \boldsymbol{\eta}_0 \boldsymbol{\eta}_0^\top$ and $\mathbf{B}_t^{(1)} = \mathbf{O}$. Therefore, the conclusion holds for $t = 0$. We assume $\mathbf{B}_{t-1} \preceq \tilde{\mathbf{B}}_{t-1} + \mathbf{B}_{t-1}^{(1)}$. Note that

$$\begin{aligned}
\mathbf{B}_t &= \mathcal{B}_t \circ \mathbf{B}_{t-1} \preceq \mathcal{B}_t \circ \left( \tilde{\mathbf{B}}_{t-1} + \mathbf{B}_{t-1}^{(1)} \right) \\
&\overset{a}{\preceq} \tilde{\mathcal{B}}_t \circ \tilde{\mathbf{B}}_{t-1} + \mathbb{E}\left[ \widehat{\mathbf{G}}_t \otimes \widehat{\mathbf{G}}_t \right] \circ \tilde{\mathbf{B}}_{t-1} + \mathcal{B}_t \circ \mathbf{B}_{t-1}^{(1)} \\
&= \tilde{\mathbf{B}}_t + \mathbf{B}_t^{(1)},
\end{aligned} \tag{58}$$

where $\overset{a}{\preceq}$ uses that $\mathcal{B}_t \preceq \tilde{\mathcal{B}}_t + \mathbb{E}\left[ \widehat{\mathbf{G}}_t \otimes \widehat{\mathbf{G}}_t \right]$ in Lemma 1. $\square$

The following step parallels Appendix A.2.2, if we replace $\mathbf{C}_t$ with $\mathbf{B}_t^{(1)}$. We include detailed proofs for completeness.

**Lemma 11.** *Suppose Assumptions 4 and 1 hold. With $\mathbf{B}_t^{(1)}$ defined in (57), we have*

$$\left\langle \tilde{\mathbf{T}}, \mathbf{B}_n^{(1)} \right\rangle \leq \|\mathbf{w}_0 - \mathbf{w}^*\|_{\mathbf{S}}^2 \cdot \left[ \sum_{i=1}^{k^*} \frac{2t_{ii}}{K\lambda_i} + \frac{512}{15} K \left( \frac{q - c\delta}{1 - c} \right)^2 \sum_{i=k^*+1}^{d} \lambda_i t_{ii} \right]. \tag{59}$$

Finally, we bound $\left\langle \tilde{\mathbf{T}}, \mathbf{B}_n \right\rangle$ and obtain the upper bound of bias.

**Lemma 12.** *Suppose Assumptions 4 and 1 hold. Then we have*

$$\text{Bias} \leq \|\mathbf{w}_0 - \mathbf{w}^*\|_{\mathbf{S}}^2 \cdot \left[ \sum_{i=1}^{k^*} \frac{2t_{ii}}{K\lambda_i} + \frac{512}{15} K \left( \frac{q - c\delta}{1 - c} \right)^2 \sum_{i=k^*+1}^{d} \lambda_i t_{ii} \right]$$

$$+ \max_{\mathbf{w} \in S(\mathbf{w}_0 - \mathbf{w}^*)} \frac{\|\mathbf{w}\|_{\mathbf{T}_{0:k^*}}^2}{8n^2(\log_2 n)^4} + 4 \|\mathbf{w}\|_{\mathbf{T}_{k^*:\infty}}^2, \tag{60}$$

*where $k^* = \max \left\{ k : \lambda_k > \frac{16(1-c)\ln n}{(q-c\delta)K} \right\}$ and $S(\mathbf{w}_0 - \mathbf{w}^*) = \left\{ \mathbf{w} \in \mathbb{R}^d : |\mathbf{w}_i| \leq |(\mathbf{w}_0 - \mathbf{w}^*)_i| \right\}.$*

*Proof.* From Lemma 10, we have $\mathbf{B}_n \preceq \tilde{\mathbf{B}}_n + \mathbf{B}_n^{(1)}$. Taking the inner product with $\tilde{\mathbf{T}}$, we get

$$\text{Bias} \leq \left\langle \tilde{\mathbf{T}}, \mathbf{B}_n \right\rangle \leq \left\langle \tilde{\mathbf{T}}, \tilde{\mathbf{B}}_n \right\rangle + \left\langle \tilde{\mathbf{T}}, \mathbf{B}_n^{(1)} \right\rangle. \tag{61}$$

Recall the definition of $\tilde{\mathbf{B}}_n$ and $\tilde{\mathcal{B}}_t$, we have

$$\tilde{\mathbf{B}}_n = \tilde{\mathcal{B}}_n \circ \tilde{\mathcal{B}}_{n-1} \circ \cdots \circ \tilde{\mathcal{B}}_1 \circ \mathbf{B}_0 = \left( \prod_{t=1}^{n} \mathbf{A}_t \begin{bmatrix} \mathbf{w}_0 - \mathbf{w}^* \\ \mathbf{w}_0 - \mathbf{w}^* \end{bmatrix} \right) \left( \prod_{t=1}^{n} \mathbf{A}_t \begin{bmatrix} \mathbf{w}_0 - \mathbf{w}^* \\ \mathbf{w}_0 - \mathbf{w}^* \end{bmatrix} \right)^{\top}. \tag{62}$$

We apply Lemma 11 and Lemma 9 to obtain

$$\text{Bias} \leq \left\langle \tilde{\mathbf{T}}, \tilde{\mathbf{B}}_n \right\rangle + \left\langle \tilde{\mathbf{T}}, \mathbf{B}_n^{(1)} \right\rangle$$

$$\leq \|\mathbf{w}_0 - \mathbf{w}^*\| \cdot \left[ \sum_{i=1}^{k^*} \frac{2t_{ii}}{K\lambda_i} + \frac{512}{15} K \left( \frac{q - c\delta}{1 - c} \right)^2 \sum_{i=k^*+1}^{d} \lambda_i t_{ii} \right] \tag{63}$$

$$+ \max_{\mathbf{w} \in S(\mathbf{w}_0 - \mathbf{w}^*)} \frac{\|\mathbf{w}\|_{\mathbf{T}_{0:k^*}}^2}{8n^2(\log_2 n)^4} + 4 \|\mathbf{w}\|_{\mathbf{T}_{k^*:\infty}}^2.$$

This completes the proof. □

### A.3 PROOF OF THEOREM 6

*Proof of Theorem 6.* Following the bias-variance decomposition, (20) shows that

$$\mathbb{E} \|\mathbf{w}_n - \mathbf{w}^*\|_{\mathbf{T}}^2 \leq 2 \cdot \text{Bias} + 2 \cdot \text{Variance}. \tag{64}$$

Lemma 12 provides the following upper bound on the bias term:

$$\text{Bias} \leq \|\mathbf{w}_0 - \mathbf{w}^*\|_{\mathbf{S}}^2 \cdot \left[ \sum_{i=1}^{k^*} \frac{2t_{ii}}{K\lambda_i} + \frac{512}{15} K \left( \frac{q - c\delta}{1 - c} \right)^2 \sum_{i=k^*+1}^{d} \lambda_i t_{ii} \right]$$

$$+ \max_{\mathbf{w} \in S(\mathbf{w}_0 - \mathbf{w}^*)} \frac{\|\mathbf{w}\|_{\mathbf{T}_{0:k^*}}^2}{8n^2(\log_2 n)^4} + 4 \|\mathbf{w}\|_{\mathbf{T}_{k^*:\infty}}^2, \tag{65}$$

where $S(\mathbf{w}_0 - \mathbf{w}^*) = \left\{ \mathbf{w} \in \mathbb{R}^d : |\mathbf{w}_i| \leq |(\mathbf{w}_0 - \mathbf{w}^*)_i| \right\}$. Recall that we set $\mathbf{w}_0 = \mathbf{0}$. Since $\mathbf{M}$ and $\mathbf{S}$ commute, so $S(\mathbf{w}_0 - \mathbf{w}^*) \subset W$. Therefore, we have

$$\max_{\mathbf{w} \in S(\mathbf{w}_0 - \mathbf{w}^*)} \|\mathbf{w}\|_{\mathbf{T}_{0:k^*}}^2 \leq \max_{\mathbf{w}^* \in W} \|\mathbf{w}^*\|_{\mathbf{T}_{0:k^*}}^2 = \max_{\mathbf{w}^* \in W} \left\| \mathbf{M}^{1/2}\mathbf{w}^* \right\|_{\mathbf{T}'_{0:k^*}}^2 = \left\| \mathbf{T}'_{0:k^*} \right\|, \tag{66}$$

Similarly, we have $\max_{\mathbf{w} \in S(\mathbf{w}_0 - \mathbf{w}^*)} \|\mathbf{w}\|^2_{\mathbf{T}_{k^*:\infty}} \leq \|\mathbf{T}'_{k^*:\infty}\|$ and $\|\mathbf{w}_0 - \mathbf{w}^*\|^2_{\mathbf{S}} \leq \|\mathbf{S}'\|$. Furthermore, we apply Lemma 8,

$$\mathbb{E} \|\mathbf{w}_n - \mathbf{w}^*\|^2_{\mathbf{T}} \leq \left(\sigma^2 + 2\|\mathbf{S}'\|\right) \cdot \left[\sum_{i=1}^{k^*} \frac{2t_{ii}}{K\lambda_i} + \frac{512}{15}K\left(\frac{q-c\delta}{1-c}\right)^2 \sum_{i=k^*+1}^{d} \lambda_i t_{ii}\right]$$

$$+ \frac{\|\mathbf{T}'_{0:k^*}\|}{8n^2(\log_2 n)^4} + 4\|\mathbf{T}'_{k^*:\infty}\|. \tag{67}$$

where $k^* = \max\left\{k : \lambda_k > \frac{16(1-c)\ln n}{(q-c\delta)K}\right\}$.

We bound the first term of the bias by the first term of variance. Note that

$$\frac{\|\mathbf{T}'_{0:k^*}\|}{8n^2(\log_2 n)^4} \leq \frac{\mathrm{tr}\left(\mathbf{T}'_{0:k^*}\right)}{8n^2(\log_2 n)^4} = \frac{1}{8n^2(\log_2 n)^4} \sum_{i=1}^{k^*} \frac{t_{ii}}{m_i}$$

$$= \frac{1}{8n^2(\log_2 n)^4} \sum_{i=1}^{k^*} \frac{t_{ii}}{K\lambda_i} \cdot \frac{K\lambda_i}{m_i} \leq \left(\max_{i \leq k^*} \frac{\lambda_i}{16n(\log_2 n)^5 m_i}\right) \sum_{i=1}^{k^*} \frac{t_{ii}}{m_i} \tag{68}$$

$$\leq \frac{\|\mathbf{S}'\|}{16n(\log_2 n)^5} \sum_{i=1}^{k^*} \frac{t_{ii}}{m_i}.$$

Therefore, we have

$$\mathbb{E} \|\mathbf{w}_n - \mathbf{w}^*\|^2_{\mathbf{T}} \leq \left(\sigma^2 + 2\|\mathbf{S}'\| + \frac{\|\mathbf{S}'\|}{16n(\log_2 n)^5}\right) \cdot \left[\sum_{i=1}^{k^*} \frac{2t_{ii}}{K\lambda_i} + \frac{512}{15}K\left(\frac{q-c\delta}{1-c}\right)^2 \sum_{i=k^*+1}^{d} \lambda_i t_{ii}\right]$$

$$+ 4\|\mathbf{T}'_{k^*:\infty}\|.$$

This completes the proof. $\qquad\square$

### A.4 INSTANCE UPPER BOUND

In this section, we provide an instance-dependent ASGD target excess risk upper bound.

**Theorem 13.** *Let $\mathbf{S}$, $\mathbf{T}$ be positive semi-definite matrices. Suppose we get samples $\{(\mathbf{x}_i, y_i)\}_{i=1}^n$ drawn from the source distribution of $\tilde{P} \in \mathcal{P}(W_\mathbf{M}, \mathbf{S}, \mathbf{T})$. When $n \geq 16$, we choose the initial step size $\delta$, $\gamma$ and the momentum $\alpha$, $\beta$ according to the parameter choice. Denote the output of Algorithm 1 as $\mathbf{w}_n^{\mathrm{SGD}}$, the target excess risk of $\mathbf{w}_n^{\mathrm{SGD}}$ can be bounded from the above by*

$$\frac{1}{2}\mathbb{E}_{\tilde{P}^{\otimes n}} \left\|\mathbf{w}_n^{\mathrm{SGD}} - \mathbf{w}^*\right\|^2_{\mathbf{T}} \leq \underbrace{\left(\sigma^2 + 2\|\mathbf{w}_0 - \mathbf{w}^*\|^2_{\mathbf{S}}\right) \cdot \left[\sum_{i=1}^{k^*} \frac{2t_{ii}}{K\lambda_i} + \frac{128}{15}K(\gamma+\delta)^2 \sum_{i=k^*+1}^{d} \lambda_i t_{ii}\right]}_{\textit{Effective Variance}}$$

$$+ \max_{\mathbf{w} \in S(\mathbf{w}_0 - \mathbf{w}^*)} \frac{\|\mathbf{w}\|^2_{\mathbf{T}_{0:k^*}}}{8n^2(\log_2 n)^4} + 4\|\mathbf{w}\|^2_{\mathbf{T}_{k^*:\infty}},$$

*where* $S(\mathbf{w}_0 - \mathbf{w}^*) = \left\{\mathbf{w} \in \mathbb{R}^d : |\mathbf{w}_i| \leq |(\mathbf{w}_0 - \mathbf{w}^*)_i|\right\}$, $k^* = \max\left\{k : \lambda_k > \frac{32(\ln n)^2}{(\gamma+\delta)n\ln 2}\right\}$, $\mathbf{T}'_{0:k^*} = \mathbf{M}^{-1/2}\mathbf{T}_{0:k^*}\mathbf{M}^{-1/2}$, *and* $\mathbf{T}'_{k^*:d} = \mathbf{M}^{-1/2}\mathbf{T}_{k^*:d}\mathbf{M}^{-1/2}$. $t_{ii}$ *denotes the $i$-th diagonal entry of $\mathbf{T}$, and $\{\lambda_i\}_{i=1}^d$ are eigenvalues of $\mathbf{S}$.*

*Proof.* Following the bias-variance decomposition, (20) shows that

$$\mathbb{E} \|\mathbf{w}_n - \mathbf{w}^*\|^2_{\mathbf{T}} \leq 2 \cdot \mathrm{Bias} + 2 \cdot \mathrm{Variance}. \tag{69}$$

Lemma 12 provides the following upper bound on the bias term:

$$\mathrm{Bias} \leq \|\mathbf{w}_0 - \mathbf{w}^*\|^2_{\mathbf{S}} \cdot \left[\sum_{i=1}^{k^*} \frac{2t_{ii}}{K\lambda_i} + \frac{512}{15}K\left(\frac{q-c\delta}{1-c}\right)^2 \sum_{i=k^*+1}^{d} \lambda_i t_{ii}\right]$$

$$+ \max_{\mathbf{w} \in S(\mathbf{w}_0 - \mathbf{w}^*)} \frac{\|\mathbf{w}\|^2_{\mathbf{T}_{0:k^*}}}{8n^2(\log_2 n)^4} + 4\|\mathbf{w}\|^2_{\mathbf{T}_{k^*:\infty}}, \tag{70}$$

Lemma 8 provides the following upper bound on the variance term:

$$\text{Variance} \le \sigma^2 \left[ \sum_{i=1}^{k^*} \frac{t_{ii}}{K\lambda_i} + \frac{256}{15} K \left( \frac{q - c\delta}{1 - c} \right)^2 \sum_{i=k^*+1}^{d} \lambda_i t_{ii} \right], \tag{71}$$

We complete the proof by combining the above two results. □

### A.5  PROOFS OF SGD AS A SPECIAL PRECONDITIONER IN SECTION 4.1

In this section, we provide proofs in Section 4.1.

*Proof of Theorem 7.* Theorem 7 is a direct implication of Theorem 6 by noting that

$$\left\| \left( \mathbf{I} - \begin{bmatrix} \mathbf{I}_{0:k^*} & \mathbf{O} \\ \mathbf{O} & \mathbf{O} \end{bmatrix} \right) \mathbf{T}' \left( \mathbf{I} - \begin{bmatrix} \mathbf{I}_{0:k^*} & \mathbf{O} \\ \mathbf{O} & \mathbf{O} \end{bmatrix} \right) \right\| = \left\| \mathbf{T}'_{k^*:d} \right\|, \tag{72}$$

$$\frac{1}{n} \left\langle \mathbf{T}', \begin{bmatrix} \mathbf{I}_{0:k^*} & \mathbf{O} \\ \mathbf{O} & \mathbf{O} \end{bmatrix} (\mathbf{S}')^{-1} \begin{bmatrix} \mathbf{I}_{0:k^*} & \mathbf{O} \\ \mathbf{O} & \mathbf{O} \end{bmatrix} \right\rangle = \sum_{i=1}^{k^*} \frac{t_{ii}}{n\lambda_i}, \tag{73}$$

$$n(\gamma + \delta)^2 \left\langle \mathbf{T}', \begin{bmatrix} \mathbf{O} & \mathbf{O} \\ \mathbf{O} & \mathbf{I}_{k^*:d} \end{bmatrix} (\mathbf{S}') \begin{bmatrix} \mathbf{O} & \mathbf{O} \\ \mathbf{O} & \mathbf{I}_{k^*:d} \end{bmatrix} \right\rangle = n(\gamma + \delta)^2 \sum_{i=k^*+1}^{d} \lambda_i t_{ii}. \tag{74}$$

□

*Proof of Theorem 11.* By Theorem 6, we have $k^* = k$, and

$$\sup_{\tilde{P}} \mathbb{E}_{\tilde{P}^{\otimes n}} \left\| \mathbf{w}_n^{\text{SGD}} - \mathbf{w}^* \right\|_{\mathbf{T}}^2 \lesssim \sum_{i=1}^{k} \frac{(\ln n)^2}{n} + \frac{\ln^4}{n} \sum_{i=k+1}^{\lfloor \sqrt{n} \rfloor} \frac{1}{\sqrt{n}} + \left\| \mathbf{T}'_{k^*:d} \right\| \tag{75}$$

$$\overset{a}{\le} \tilde{\mathcal{O}}(1/n),$$

where $\overset{a}{\le}$ uses $\left\| \mathbf{T}'_{k^*:d} \right\| = 0$ since $\mathbf{M} = \text{diag} \left\{ \frac{1}{k} \mathbf{I}_k, \infty \mathbf{I}_{k+1:d} \right\}$. □

### A.6  PROPERTIES OF MOMENTUM MATRIX

#### A.6.1  BOUND OF SPECTRAL RADIUS

Recall that the definition of $\mathbf{A}_t$ is

$$\mathbf{A}_t = \mathbb{E}\widehat{\mathbf{A}}_t = \begin{bmatrix} \mathbf{O} & \mathbf{I} - \delta_t \mathbf{S} \\ -c\mathbf{I} & (1 + c)\mathbf{I} - q_t \mathbf{S} \end{bmatrix}. \tag{76}$$

Note that $\mathbf{S}$ is diagonal and $\mathbf{A}_t$ is block-diagonal in the eigenspace of $\mathbf{S}$. Let $\mathbf{A}_{t,i}$ denotes the $i$-th block corresponding to $\lambda_i$, the $i$-th largest eigenvalue. Therefore,

$$\mathbf{A}_{t,i} = \begin{bmatrix} 0 & 1 - \delta_t \lambda_i \\ -c & 1 + c - q_t \lambda_i \end{bmatrix}. \tag{77}$$

For convenience, we also define $\ell$-th stage version

$$\mathbf{A}_{(\ell)} = \begin{bmatrix} \mathbf{O} & \mathbf{I} - \delta_{(\ell)} \mathbf{S} \\ -c\mathbf{I} & (1 + c)\mathbf{I} - q_{(\ell)} \mathbf{S} \end{bmatrix}, \quad \mathbf{A}_{(\ell),i} = \begin{bmatrix} 0 & 1 - \delta_{(\ell)} \lambda_i \\ -c & 1 + c - q_{(\ell)} \lambda_i \end{bmatrix}. \tag{78}$$

Note that only the product of step size and eigenvalue appears in $\mathbf{A}_{t,i}$, we further define

$$\mathbf{A}(\lambda) = \begin{bmatrix} 0 & 1 - \delta\lambda \\ -c & 1 + c - q\lambda \end{bmatrix}. \tag{79}$$

Recall the exponential decaying step size schedule (16), we have

$$\mathbf{A}_{t,i} = \mathbf{A}_{(\ell),i} = \mathbf{A} \left( \frac{\lambda_i}{4^{\ell-1}} \right), \quad \text{if } K(\ell - 1) + 1 \le t \le K\ell. \tag{80}$$

The eigenvalues of $\mathbf{A}(\lambda)$ are

$$x_1 = \frac{1 + c - q\lambda}{2} - \frac{\sqrt{(1 + c - q\lambda)^2 - 4c(1 - \delta\lambda)}}{2}, \tag{81}$$

$$x_2 = \frac{1 + c - q\lambda}{2} + \frac{\sqrt{(1 + c - q\lambda)^2 - 4c(1 - \delta\lambda)}}{2}. \tag{82}$$

Solving $(1 + c - q\lambda)^2 - 4c(1 - \delta\lambda) \leq 0$ yields

$$\underbrace{\frac{(1 - c)^2}{\left(\sqrt{q - c\delta} + \sqrt{c(q - \delta)}\right)^2}}_{\lambda^\dagger} < \lambda < \underbrace{\frac{(1 - c)^2}{\left(\sqrt{q - c\delta} - \sqrt{c(q - \delta)}\right)^2}}_{\lambda^\ddagger}. \tag{83}$$

We define three intervals

$$I_1 = \left[0, \lambda^\dagger\right], \quad I_2 = \left(\lambda^\dagger, \lambda^\ddagger\right), \quad I_3 = \left[\lambda^\ddagger, +\infty\right). \tag{84}$$

Note that the spectral radius $\rho(\mathbf{A}(\lambda)) = |x_2|$. We adopt Lemma E.2 from Li et al. (2024), which characterizes $x_1$ and $x_2$.

**Lemma 13.** *Let $\lambda \geq 0$.*

- *If $\lambda \in I_1$, then $x_1$ and $x_2$ are real, and*

$$x_1 \leq x_2 \leq 1 - \frac{q - c\delta}{1 - c}\lambda; \tag{85}$$

- *If $\lambda \in I_2$, then $x_1$ and $x_2$ are complex, and*

$$|x_1| = |x_2| = \sqrt{c(1 - \delta\lambda)}; \tag{86}$$

- *If $\lambda \in I_3$, then $x_1$ and $x_2$ are real, and*

$$x_1 \leq x_2 \leq \frac{c\delta}{q}. \tag{87}$$

### A.6.2 BOUND OF PRODUCT OF MOMENTUM MATRIX

In this section, we provide bounds of $\mathbf{A}^k(\lambda)$. The following lemma provides upper bound of $\left\|\mathbf{A}^k(\lambda)\right\|$.

**Lemma 14.** *Given $\mathbf{A}(\lambda)$ that are defined in (79), we have*

$$\left\|\mathbf{A}^k(\lambda)\right\| \leq \left\|\mathbf{A}^k(\lambda)\right\|_F \leq \sqrt{6}k\left[\rho(\mathbf{A}(\lambda))\right]^{k-1}. \tag{88}$$

*Proof.* Define

$$a_k = \frac{x_2^k - x_1^k}{x_2 - x_1}, \tag{89}$$

we have $a_k \in \mathbb{R}$ and

$$\mathbf{A}^k(\lambda) = \begin{bmatrix} -c(1 - \delta\lambda)a_{k-1} & (1 - \delta\lambda)a_k \\ -ca_k & a_{k+1} \end{bmatrix}. \tag{90}$$

Note that for any $\lambda \geq 0$, we have $|x_1| \leq |x_2|$, and

$$\begin{aligned} |a_k| = \left|\frac{x_2^k - x_1^k}{x_2 - x_1}\right| &= \left|\sum_{i=0}^{k-1} x_1^i x_2^{k-1-i}\right| \\ &\overset{a}{\leq} \sum_{i=0}^{k-1} |x_1|^i |x_2|^{k-1-i} \overset{b}{\leq} \sum_{i=0}^{k-1} |x_2|^{k-1} \\ &= k|x_2|^{k-1}, \end{aligned} \tag{91}$$

where $\overset{a}{\leq}$ uses the triangular inequality for complex number, and $\overset{b}{\leq}$ uses $|x_1| \leq |x_2|$. Direct calculation yields $x_1 x_2 = c\,(1 - \delta\lambda)$, so $|c\,(1 - \delta\lambda)| \leq |x_2|^2$. We bound $\left\| \mathbf{A}^k(\lambda) \right\|_F^2$ by

$$
\begin{aligned}
\left\| \mathbf{A}^k(\lambda) \right\|_F^2 &= [-c\,(1 - \delta\lambda)\,a_{k-1}]^2 + [(1 - \delta\lambda)\,a_k]^2 + (-ca_k)^2 + a_{k+1}^2 \\
&\leq (k-1)^2 |x_2|^{2k} + k^2 |x_2|^{2(k-1)} + k^2 |x_2|^{2(k-1)} + (k+1)^2 |x_2|^{2k} \\
&\leq \left[ (k-1)^2 + k^2 + k^2 + (k+1)^2 \right] |x_2|^{2(k-1)} \\
&= \left( 4k^2 + 2 \right) |x_2|^{2(k-1)}. \\
&\leq 6k^2 |x_2|^{2(k-1)}.
\end{aligned}
\tag{92}
$$

Therefore, $\left\| \mathbf{A}^k(\lambda) \right\| \leq \left\| \mathbf{A}^k(\lambda) \right\|_F \leq \sqrt{6}k |x_2|^{2(k-1)} = \sqrt{6}k \left[ \rho(\mathbf{A}(\lambda)) \right]^{k-1}$. □

For $k \leq K$, the following lemma bounds $\left\| \mathbf{A}^k(\lambda) \right\|$ from above uniformly.

**Lemma 15.** *For $\lambda \leq \lambda_1$, we have*

$$
\left\| \mathbf{A}^k(\lambda) \right\| \leq \sqrt{6}K.
\tag{93}
$$

*Proof.* For $k = 0$, the conclusion is obvious. If $k \geq 1$, for $\lambda \leq \lambda_1$, we have $\rho(\mathbf{A}(\lambda) \leq 1$. Thus, by Lemma 14,

$$
\left\| \mathbf{A}^k(\lambda) \right\| \leq \sqrt{6}K \left[ \rho(\mathbf{A}(\lambda) \right]^{k-1}.
\tag{94}
$$

□

The following lemma bounds $\left\| \mathbf{A}^K(\lambda) \right\|$ from above.

**Lemma 16.** *For $\lambda \geq \frac{4(1-c)\ln n}{(q - c\delta)K}$ and $n \geq 16$, we have*

$$
\left\| \mathbf{A}^K(\lambda) \right\| \leq \frac{\sqrt{6}}{n^2 \log_2 n} \leq 1.
\tag{95}
$$

*Proof.* We bound $\left\| \mathbf{A}^K(\lambda) \right\|$ for $\lambda \in I_1, I_2, I_3$, respectively.

1. If $\lambda \in I_1$, by Lemma 13 and assumption,

$$
\rho(\mathbf{A}(\lambda)) = |x_2| \leq 1 - \frac{q - c\delta}{1 - c}\lambda \leq 1 - \frac{4\ln n}{4K}.
\tag{96}
$$

Thus, by Lemma 14,

$$
\begin{aligned}
\left\| \mathbf{A}^K(\lambda) \right\| &\leq \sqrt{6}K \left[ \rho(\mathbf{A}(\lambda)) \right]^{K-1} \leq \sqrt{6}K \left( 1 - 4\ln n \right)^{K-1} \\
&= \sqrt{6}K \exp\left[ (K-1)\ln\left( 1 - 4\ln n \right) \right] \\
&\overset{a}{\leq} \sqrt{6}K \exp\left[ -\frac{4(K-1)\ln n}{K} \right] \overset{b}{\leq} \sqrt{6}K \exp\left( -3\ln n \right) \\
&= \frac{\sqrt{6}}{n^2 \log_2 n},
\end{aligned}
\tag{97}
$$

where $\overset{a}{\leq}$ uses $\ln x \leq x - 1, \forall x \in \mathbb{R}$, and $\overset{b}{\leq}$ holds for $n \geq 16 \implies K \geq 4 \implies \frac{K-1}{K} \geq \frac{3}{4}$.

2. If $\lambda \in I_2$, by Lemma 13 and assumption,

$$
\rho(\mathbf{A}(\lambda)) = |x_2| = \sqrt{c(1 - \delta\lambda)} \leq \sqrt{c}.
\tag{98}
$$

Thus, by Lemma 14,

$$
\begin{aligned}
\left\| \mathbf{A}^K(\lambda) \right\| \leq & \sqrt{6} K \left[ \rho(\mathbf{A}(\lambda)) \right]^{K-1} \leq \sqrt{6} K \left( \sqrt{c} \right)^{K-1} \\
= & \sqrt{6} K \exp \left[ -\frac{(K-1)\ln c}{2} \right] \\
\overset{a}{\leq} & \sqrt{6} K \exp \left[ -\frac{(K-1)(1-c)}{2} \right] \overset{b}{\leq} \sqrt{6} K \exp \left[ -\frac{8(K-1)\ln n}{K} \right] \\
\leq & \sqrt{6} K \exp\left( -6 \ln n \right) = \frac{\sqrt{6}}{n^5 \log_2 n} \leq \frac{\sqrt{6}}{n^2 \log_2 n},
\end{aligned}
\tag{99}
$$

where $\overset{a}{\leq}$ uses $\ln x \leq x-1, \forall x \in \mathbb{R}$, and $\overset{b}{\leq}$ holds for $K(1-c) \geq 16 \ln n$ in (29) and $\frac{K-1}{K} \geq \frac{3}{4}$.

3. If $\lambda \in I_2$, by Lemma 13 and assumption,

$$
\rho(\mathbf{A}(\lambda)) = |x_2| \leq \frac{c\delta}{q} \leq c.
\tag{100}
$$

Thus, by Lemma 14,

$$
\begin{aligned}
\left\| \mathbf{A}^K(\lambda) \right\| \leq & \sqrt{6} K \left[ \rho(\mathbf{A}(\lambda)) \right]^{K-1} \leq \sqrt{6} K c^{K-1} \\
= & \sqrt{6} K \exp \left[ -(K-1)\ln c \right] \\
\overset{a}{\leq} & \sqrt{6} K \exp \left[ -(K-1)(1-c) \right] \overset{b}{\leq} \sqrt{6} K \exp \left[ -\frac{16(K-1)\ln n}{K} \right] \\
\leq & \sqrt{6} K \exp\left( -12 \ln n \right) = \frac{\sqrt{6}}{n^{11} \log_2 n} \leq \frac{\sqrt{6}}{n^2 \log_2 n},
\end{aligned}
\tag{101}
$$

where $\overset{a}{\leq}$ uses $\ln x \leq x-1, \forall x \in \mathbb{R}$, and $\overset{b}{\leq}$ holds for $\frac{K-1}{K} \geq \frac{3}{4}$.

$\square$

For $k \in \mathbb{N}$, we have a uniform bound of $\left| \left( \mathbf{A}^k(\lambda) \begin{bmatrix} 1 \\ 1 \end{bmatrix} \right)_2 \right|$, which is tighter than Lemma 15.

**Lemma 17.** *For $\lambda \leq \lambda_1$ and $k \in \mathbb{N}$, we have*

$$
\left| \left( \mathbf{A}^k(\lambda) \begin{bmatrix} 1 \\ 1 \end{bmatrix} \right)_2 \right| \leq \begin{cases} 1, & \lambda \in I_1, I_3; \\ 2, & \lambda \in I_2. \end{cases}
\tag{102}
$$

*Proof.* From (90), we have

$$
\left| \left( \mathbf{A}^k(\lambda) \begin{bmatrix} 1 \\ 1 \end{bmatrix} \right)_2 \right| = |a_{k+1} - c a_k|.
\tag{103}
$$

We bound $|a_{k+1} - c a_k| \leq 2$ for $\lambda \in I_1, I_2, I_3$, respectively.

1. If $\lambda \in I_1$, by Lemma 13, and $\delta \leq q$, we have $a_k \geq 0$, and

$$
x_1 \leq x_2 \leq 1 - \frac{q - c\delta}{1 - c}\lambda \leq 1 - \delta \lambda.
\tag{104}
$$

Since $x_1 x_2 = c(1 - \delta\lambda)$, we have $c \leq x_1 \leq x_2$. Therefore,

$$
\begin{aligned}
a_{k+1} - c a_k \geq & a_{k+1} - x_1 a_k = x_2^k > 0, \\
a_{k+1} - c a_k \leq & a_{k+1} - x_1 x_2 a_k = \sum_{i=0}^{k} x_1^i x_2^{k-i} - x_1 x_2 \sum_{i=0}^{k-1} x_1^i x_2^{k-1-i} \\
= & \sum_{i=0}^{k} x_1^i x_2^{k-i} - x_2 \sum_{i=1}^{k} x_1^i x_2^{k-i} = x_2^k + (1 - x_2) \sum_{i=1}^{k} x_1^i x_2^{k-i} \\
\leq & x_2^k + k(1 - x_2)x_2^k = x_2^k \left[ 1 + k(1 - x_2) \right] \overset{a}{\leq} 1,
\end{aligned}
\tag{105}
$$

where $\overset{a}{\leq}$ applies Lemma 26.

2. If $\lambda \in I_2$, by Lemma 13, $x_1$ and $x_2$ are complex and conjugate. Let $x_{1,2} = r(\cos\theta \pm \mathrm{i}\sin\theta)$, we have $r = \sqrt{c(1-\delta\lambda)} \leq 1$ and $0 \leq \theta \leq \pi/2$ where $2r\cos\theta = x_1 + x_2 = 1 + c - q\lambda \geq 0$ from Lemma 2. Thus

$$
\begin{aligned}
a_{k+1} - ca_k &= \frac{r^k \sin((k+1)\theta)}{\sin\theta} - \frac{r^{k-1}\sin(k\theta)}{\sin\theta} \\
&\overset{a}{=} r^{k-1}\left( r\cos k\theta + \frac{r\cos\theta - c}{\sin\theta}\sin k\theta \right) \\
&= r^{k-1}\left( r\cos k\theta + \frac{r-c}{\sin\theta}\sin k\theta - \frac{r(1-\cos\theta)}{\sin\theta}\sin k\theta \right) \\
&\overset{b}{=} r^{k-1}\left( r\cos k\theta + \frac{r-c}{\sin\theta}\sin k\theta - r\tan\frac{\theta}{2}\sin k\theta \right),
\end{aligned}
\tag{106}
$$

where $\overset{a}{=}$ is from $\sin((k+1)\theta) = \sin k\theta\cos\theta + \cos k\theta\sin\theta$, and $\overset{b}{=}$ is from

$$
\frac{1-\cos\theta}{\sin\theta} = \frac{2\sin^2\frac{\theta}{2}}{2\sin\frac{\theta}{2}\cos\frac{\theta}{2}} = \tan\frac{\theta}{2}.
\tag{107}
$$

By triangular inequality, and $|\sin k\theta| \leq 1$, $|\cos k\theta| \leq 1$, $\left|\tan\frac{\theta}{2}\right| \leq 1$

$$
\begin{aligned}
|a_{k+1} - ca_k| &\leq r^{k-1}\left( r|\cos k\theta| + |r-c|\left|\frac{\sin k\theta}{\sin\theta}\right| \right) + r^k\left|\tan\frac{\theta}{2}\right||\sin k\theta| \\
&\overset{a}{\leq} r^{k-1}(r + k(1-r)) + r^k = r^{k-1}(1 + (k-1)(1-r)) + r^k \\
&\overset{b}{\leq} 2,
\end{aligned}
\tag{108}
$$

where $\overset{a}{\leq}$ holds for $r^2 \leq c \leq 1 \implies |r-c| \leq \max\left\{\left|r-r^2\right|, |r-1|\right\} = 1-r$ and $\left|\frac{\sin k\theta}{\sin\theta}\right| \leq k$ in Lemma 27, $\overset{b}{\leq}$ is from Lemma 26 and $0 \leq r \leq 1$.

3. If $\lambda \in I_3$, by Lemma 13, and $\delta \leq q$, we have $a_k \geq 0$, and

$$
x_1 \leq x_2 \leq \frac{c\delta}{q} \leq c.
\tag{109}
$$

Therefore,

$$
\begin{aligned}
a_{k+1} - ca_k &\geq a_{k+1} - a_k = \sum_{i=0}^{k} x_1^i x_2^{k-i} - \sum_{i=0}^{k-1} x_1^i x_2^{k-1-i} \\
&= x_1^k - (1-x_2)\sum_{i=0}^{k-1} x_1^i x_2^{k-1-i} \\
&\geq x_1^k - k(1-x_2)x_2^{k-1} \\
&\geq -x_2^{k-1}\left(1 + (k-1)x_2^k\right) \overset{a}{\geq} -1, \\
a_{k+1} - ca_k &\leq a_{k+1} - x_2 a_k = x_1^k \leq 1,
\end{aligned}
\tag{110}
$$

where $\overset{a}{\geq}$ holds for Lemma 26.

$\square$

For $\lambda \leq \frac{(1-c)^2}{q-c\delta}$, we define $\mathbf{P}$, which diagonalizes $\mathbf{V}$:

$$
\mathbf{P} = \begin{bmatrix} 1 & -1 \\ 1 & -c \end{bmatrix}, \quad \mathbf{P}^{-1} = \frac{1}{1-c}\begin{bmatrix} -c & 1 \\ -1 & 1 \end{bmatrix}.
\tag{111}
$$

The following lemma provides bound of $\mathbf{P}^{-1}\mathbf{A}(\lambda)\mathbf{P}$.

**Lemma 18.** *Let $\mathbf{P}$ and $\mathbf{P}^{-1}$ defined in (111). Suppose $\lambda \leq \frac{(1-c)^2}{q-c\delta}$, we have*

$$\left\| \mathbf{P}^{-1} \mathbf{A}(\lambda) \mathbf{P} \right\| \leq 1. \tag{112}$$

*Proof.* Let

$$\mathbf{M} = \mathbf{P}^{-1} \mathbf{A}(\lambda) \mathbf{P} = \begin{bmatrix} 1 - \xi\lambda & c\xi\lambda \\ -\eta\lambda & c + c\eta\lambda \end{bmatrix}, \tag{113}$$

we will show that $\mathbf{I} - \mathbf{M}^{\top}\mathbf{M}$ is a PSD matrix. Let $\xi = \frac{q-c\delta}{1-c}$ and $\eta = \frac{q-\delta}{1-c}$, so $\xi\lambda < 1-c$. Direct calculation shows that

$$\mathbf{M}^{\top}\mathbf{M} = \begin{bmatrix} (1-\xi\lambda)^2 + \eta^2\lambda^2 & c\lambda\left(\xi - \eta - \left(\xi^2 + \eta^2\right)\lambda\right) \\ c\lambda\left(\xi - \eta - \left(\xi^2 + \eta^2\right)\lambda\right) & c^2\xi^2\lambda^2 + c^2(1+\eta\lambda)^2 \end{bmatrix}. \tag{114}$$

Furthermore,

$$\begin{aligned}
\left(\mathbf{I} - \mathbf{M}^{\top}\mathbf{M}\right)_{11} &= 1 - \left[(1-\xi\lambda)^2 + \eta^2\lambda^2\right] = 2\xi\lambda - \xi^2\lambda^2 - \eta^2\lambda^2 \\
&\overset{a}{\geq} 2\xi\lambda - 2\xi^2\lambda^2 = 2\xi\lambda(1-\xi\lambda) \\
&\overset{b}{\geq} 0, \\
\det\left(\mathbf{I} - \mathbf{M}^{\top}\mathbf{M}\right) &= \lambda\left[2(1-c^2)\xi - (\xi^2 + \eta^2 + 2c^2\xi\eta)\lambda\right] \\
&\overset{c}{\geq} \lambda\left[2(1-c^2)\xi - 2(1+c^2)\xi^2\lambda\right] \\
&\overset{d}{\geq} \lambda\left[2(1-c^2)\xi - 2(1+c^2)(1-c)\xi\right] \\
&= \left[2(1-c^2) - 2(1+c^2)(1-c)\right]\xi\lambda \\
&= 2c(1-c)^2\xi\lambda \geq 0
\end{aligned} \tag{115}$$

where $\overset{a}{\geq}$ and $\overset{c}{\geq}$ uses $\eta \leq \xi$, $\overset{b}{\geq}$ and $\overset{d}{\geq}$ a uses $\xi\lambda \leq 1 - c \leq 1$. Therefore, by Sylvester's criterion, $\mathbf{I} - \mathbf{M}^{\top}\mathbf{M}$ is a PSD matrix. From the definition of $\mathbf{M}$, we have

$$\left\| \mathbf{P}^{-1} \mathbf{A}(\lambda) \mathbf{P} \right\| = \|\mathbf{M}\| = \sup_{\mathbf{x}} \frac{\|\mathbf{M}\mathbf{x}\|}{\|\mathbf{x}\|} = \sup_{\mathbf{x}} \frac{\mathbf{x}^{\top}\mathbf{M}\mathbf{M}\mathbf{x}}{\mathbf{x}^{\top}\mathbf{x}} \leq 1. \tag{116}$$

$\square$

The following lemma provides upper bound of the product of momentum matrices.

**Lemma 19.** *For $\mu_1, \mu_2, \ldots, \mu_k \leq \frac{(1-c)^2}{q-c\delta}$, we have*

$$\left\| \prod_{i=1}^{k} \mathbf{A}(\mu_i) \right\| \leq \frac{4}{1-c}. \tag{117}$$

*Proof.* Note that

$$\begin{aligned}
\left\| \prod_{i=1}^{k} \mathbf{A}(\mu_i) \right\| &= \left\| \mathbf{P} \left( \prod_{i=1}^{k} \mathbf{P}^{-1} \mathbf{A}(\mu_i) \mathbf{P} \right) \mathbf{P}^{-1} \right\| \\
&\leq \|\mathbf{P}\| \prod_{i=1}^{k} \left\| \mathbf{P}^{-1} \mathbf{A}(\mu_i) \mathbf{P} \right\| \left\| \mathbf{P}^{-1} \right\| \\
&\overset{a}{\leq} 2 \cdot 1 \cdot \frac{2}{1-c} = \frac{4}{1-c},
\end{aligned} \tag{118}$$

where $\overset{a}{\leq}$ applies $\|\mathbf{P}\| \leq \|\mathbf{P}\|_F \leq 2$, $\left\| \mathbf{P}^{-1} \right\| \leq \left\| \mathbf{P}^{-1} \right\|_F \leq \frac{2}{1-c}$ and Lemma 18. $\square$

The following lemma provides an upper bound of the product of momentum matrices applied to noise vector $\begin{bmatrix} \delta & q \end{bmatrix}^{\top}$.

**Lemma 20.** *For $\mu_1, \mu_2, \ldots, \mu_k \leq \frac{(1-c)^2}{q-c\delta}$, we have*

$$\left\| \prod_{i=1}^{k} \mathbf{A}(\mu_i) \begin{bmatrix} \delta \\ q \end{bmatrix} \right\| \leq \frac{2\sqrt{2}(q-c\delta)}{1-c}. \tag{119}$$

*Proof.* Note that

$$
\begin{aligned}
\left\| \prod_{i=1}^{k} \mathbf{A}(\mu_i) \right\| &= \left\| \mathbf{P} \left( \prod_{i=1}^{k} \mathbf{P}^{-1}\mathbf{A}(\mu_i)\mathbf{P} \right) \mathbf{P}^{-1} \begin{bmatrix} \delta \\ q \end{bmatrix} \right\| \\
&= \left\| \mathbf{P} \left( \prod_{i=1}^{k} \mathbf{P}^{-1}\mathbf{A}(\mu_i)\mathbf{P} \right) \begin{bmatrix} \frac{q-c\delta}{1-c} \\ \frac{q-\delta}{1-c} \end{bmatrix} \right\| \\
&\leq \|\mathbf{P}\| \prod_{i=1}^{k} \left\| \mathbf{P}^{-1}\mathbf{A}(\mu_i)\mathbf{P} \right\| \left\| \begin{bmatrix} \frac{q-c\delta}{1-c} \\ \frac{q-\delta}{1-c} \end{bmatrix} \right\| \\
&\overset{a}{\leq} 2 \cdot 1 \cdot \frac{\sqrt{2}(q-c\delta)}{1-c} = \frac{2\sqrt{2}(q-c\delta)}{1-c},
\end{aligned} \tag{120}
$$

where $\overset{a}{\leq}$ applies $\|\mathbf{P}\| \leq 2$, $\frac{q-\delta}{1-c} \leq \frac{q-c\delta}{1-c}$ and Lemma 18. □

The following lemma provides an upper bound of the product of momentum matrices applied to bias vector $[1 \quad 1]^{\top}$.

**Lemma 21.** *For $\mu_1, \mu_2, \ldots, \mu_k \leq \frac{(1-c)^2}{q-c\delta}$, we have*

$$\left\| \prod_{i=1}^{k} \mathbf{A}(\mu_i) \begin{bmatrix} 1 \\ 1 \end{bmatrix} \right\| \leq 2. \tag{121}$$

*Proof.* Note that

$$
\begin{aligned}
\left\| \prod_{i=1}^{k} \mathbf{A}(\mu_i) \right\| &= \left\| \mathbf{P} \left( \prod_{i=1}^{k} \mathbf{P}^{-1}\mathbf{A}(\mu_i)\mathbf{P} \right) \mathbf{P}^{-1} \begin{bmatrix} 1 \\ 1 \end{bmatrix} \right\| \\
&= \left\| \mathbf{P} \left( \prod_{i=1}^{k} \mathbf{P}^{-1}\mathbf{A}(\mu_i)\mathbf{P} \right) \begin{bmatrix} 1 \\ 0 \end{bmatrix} \right\| \\
&\leq \|\mathbf{P}\| \prod_{i=1}^{k} \left\| \mathbf{P}^{-1}\mathbf{A}(\mu_i)\mathbf{P} \right\| \left\| \begin{bmatrix} 1 \\ 0 \end{bmatrix} \right\| \\
&\overset{a}{\leq} 2 \cdot 1 \cdot 1 = 2,
\end{aligned} \tag{122}
$$

where $\overset{a}{\leq}$ applies $\|\mathbf{P}\| \leq 2$ and Lemma 18. □

### A.7 VARIANCE UPPER BOUND

This section analyzes $\tilde{\mathbf{C}}_t$ which defined in (44). We first provide a characterization of the stationary state, and then prove Lemma 4 and 6.

#### A.7.1 ANALYSIS OF STATIONARY STATE

We introduce the stationary state matrix at $\ell$-th stage:

$$\tilde{\mathbf{Q}}_{(\ell)} = \sum_{k=1}^{\infty} \tilde{\mathcal{B}}_{(\ell)}^{k} \circ \begin{bmatrix} \delta_{(\ell)}^2 \mathbf{S} & \delta_{(\ell)} q_{(\ell)} \mathbf{S} \\ \delta_{(\ell)} q_{(\ell)} \mathbf{S} & q_{(\ell)}^2 \mathbf{S} \end{bmatrix}. \tag{123}$$

Lemma F.4 in Li et al. (2024) shows $\tilde{\mathbf{Q}}_{(\ell)}$ exists and finite. Note that since $\tilde{\mathcal{B}}_t = \mathbf{A}_{(\ell)} \otimes \mathbf{A}_{(\ell)}$ and $\mathbf{A}_{(\ell)}$ is block-diagonal, each $\tilde{\mathcal{B}}_{(\ell)}^k \circ \begin{bmatrix} \delta_{(\ell)}^2 \mathbf{S} & \delta_{(\ell)} q_{(\ell)} \mathbf{S} \\ \delta_{(\ell)} q_{(\ell)} \mathbf{S} & q_{(\ell)}^2 \mathbf{S} \end{bmatrix}$ is also block-diagonal. Thus, $\tilde{\mathbf{Q}}_{(\ell)}$ is block-diagonal, and we denote the $i$-th block as $\tilde{\mathbf{Q}}_{(\ell),i} \in \mathbb{R}^{2\times 2}$. Furthermore, we define

$$\tilde{\mathcal{B}}_{t,i} = \mathbf{A}_{t,i} \otimes \mathbf{A}_{t,i}, \quad \tilde{\mathcal{B}}_{(\ell),i} = \mathbf{A}_{(\ell),i} \otimes \mathbf{A}_{(\ell),i}, \tag{124}$$

Then $\tilde{\mathbf{Q}}_{(\ell),i}$ can be represented as

$$\tilde{\mathbf{Q}}_{(\ell),i} = \sum_{k=1}^{\infty} \tilde{\mathcal{B}}_{(\ell),i}^k \circ \begin{bmatrix} \delta_{(\ell)}^2 \lambda_i & \delta_{(\ell)} q_{(\ell)} \lambda_i \\ \delta_{(\ell)} q_{(\ell)} \lambda_i & q_{(\ell)}^2 \lambda_i \end{bmatrix}. \tag{125}$$

Define an operator $\mathcal{T}_{(\ell)} = \mathcal{I} - \tilde{\mathcal{B}}_{(\ell)} + \mathbf{G}_{(\ell)} \otimes \mathbf{G}_{(\ell)} = \mathcal{I} - \mathbf{V} \otimes \mathbf{V} + \mathbf{V} \otimes \mathbf{G}_{(\ell)} + \mathbf{G}_{(\ell)} \otimes \mathbf{V}$, and

$$\mathbf{U}_{(\ell)} = \mathcal{T}_{(\ell)}^{-1} \circ \begin{bmatrix} \delta_{(\ell)}^2 \mathbf{S} & \delta_{(\ell)} q_{(\ell)} \mathbf{S} \\ \delta_{(\ell)} q_{(\ell)} \mathbf{S} & q_{(\ell)}^2 \mathbf{S} \end{bmatrix}. \tag{126}$$

The same argument holds for $\mathbf{U}_{(\ell)}$ to be block-diagonal, and $i$-th block of $\mathbf{U}_{(\ell)}$ is denoted as $\mathbf{U}_{(\ell),i} \in \mathbb{R}^{2\times 2}$. The following lemma characterize $\tilde{\mathbf{Q}}_{(\ell)}$ using $\mathbf{U}_{(\ell),i}$.

**Lemma 22.** *Let $\tilde{\mathbf{Q}}_{(\ell)}$ defined in* (123). *Then we have*

$$\tilde{\mathbf{Q}}_{(\ell),i} = \frac{1}{1 - \left(\mathbf{U}_{(\ell),i}\right)_{22} \lambda_i} \mathbf{U}_{(\ell),i}. \tag{127}$$

*Proof.* Note that

$$\begin{aligned}
\sum_{k=0}^{\infty} \tilde{\mathcal{B}}_{(\ell)}^k &= \left(\mathcal{I} - \tilde{\mathcal{B}}_{(\ell)}\right)^{-1} = \left(\mathcal{T}_{(\ell)} - \mathbf{G}_{(\ell)} \otimes \mathbf{G}_{(\ell)}\right)^{-1} \\
&= \left[\mathcal{T}_{(\ell)} \circ \left(\mathcal{I} - \mathcal{T}_{(\ell)}^{-1} \circ \left(\mathbf{G}_{(\ell)} \otimes \mathbf{G}_{(\ell)}\right)\right)\right]^{-1} \\
&= \left(\mathcal{I} - \mathcal{T}_{(\ell)}^{-1} \circ \left(\mathbf{G}_{(\ell)} \otimes \mathbf{G}_{(\ell)}\right)\right)^{-1} \circ \mathcal{T}_{(\ell)}^{-1} \\
&= \sum_{k=0}^{\infty} \left(\mathcal{T}_{(\ell)}^{-1} \circ \left(\mathbf{G}_{(\ell)} \otimes \mathbf{G}_{(\ell)}\right)\right)^k \circ \mathcal{T}_{(\ell)}^{-1}.
\end{aligned} \tag{128}$$

We introduce $\mathcal{T}_{(\ell),i} = \mathcal{I} - \mathbf{V}_i \otimes \mathbf{V}_i + \mathbf{V}_i \otimes \mathbf{G}_{(\ell),i} + \mathbf{G}_{(\ell),i} \otimes \mathbf{V}_i$, which operates on $\mathbb{R}^{2\times 2}$ matrix. Therefore, we can calculate the $i$-th block of $\tilde{\mathbf{Q}}_{(\ell)}$ as follows:

$$\begin{aligned}
\tilde{\mathbf{Q}}_{(\ell),i} &= \sum_{k=0}^{\infty} \left(\mathcal{T}_{(\ell),i}^{-1} \circ \left(\mathbf{G}_{(\ell),i} \otimes \mathbf{G}_{(\ell),i}\right)\right)^k \circ \mathcal{T}_{(\ell),i}^{-1} \circ \begin{bmatrix} \delta_{(\ell)}^2 \lambda_i & \delta_{(\ell)} q_{(\ell)} \lambda_i \\ \delta_{(\ell)} q_{(\ell)} \lambda_i & q_{(\ell)}^2 \lambda_i \end{bmatrix} \\
&= \sum_{k=0}^{\infty} \left(\mathcal{T}_{(\ell),i}^{-1} \circ \left(\mathbf{G}_{(\ell),i} \otimes \mathbf{G}_{(\ell),i}\right)\right)^k \circ \mathbf{U}_{(\ell),i} \\
&= \mathbf{U}_{(\ell),i} + \sum_{k=1}^{\infty} \left(\mathcal{T}_{(\ell),i}^{-1} \circ \left(\mathbf{G}_{(\ell),i} \otimes \mathbf{G}_{(\ell),i}\right)\right)^k \circ \mathbf{U}_{(\ell),i} \\
&= \mathbf{U}_{(\ell),i} + \sum_{k=0}^{\infty} \left(\mathcal{T}_{(\ell),i}^{-1} \circ \left(\mathbf{G}_{(\ell),i} \otimes \mathbf{G}_{(\ell),i}\right)\right)^k \circ \mathcal{T}_{(\ell),i}^{-1} \circ \left(\mathbf{G}_{(\ell),i} \otimes \mathbf{G}_{(\ell),i}\right) \circ \mathbf{U}_{(\ell),i} \\
&\overset{a}{=} \mathbf{U}_{(\ell),i} + \left(\mathbf{U}_{(\ell),i}\right)_{22} \lambda_i \sum_{k=0}^{\infty} \left(\mathcal{T}_{(\ell),i}^{-1} \circ \left(\mathbf{G}_{(\ell),i} \otimes \mathbf{G}_{(\ell),i}\right)\right)^k \circ \mathbf{U}_{(\ell),i} \\
&= \mathbf{U}_{(\ell),i} + \left(\mathbf{U}_{(\ell),i}\right)_{22} \lambda_i \tilde{\mathbf{Q}}_{(\ell),i},
\end{aligned} \tag{129}$$

where $\overset{a}{=}$ uses

$$
\begin{aligned}
\mathcal{T}_{(\ell),i}^{-1} \circ \left(\mathbf{G}_{(\ell),i} \otimes \mathbf{G}_{(\ell),i}\right) \circ \mathbf{U}_{(\ell),i} &= \mathcal{T}_{(\ell),i}^{-1} \circ \left(\mathbf{G}_{(\ell),i} \mathbf{U}_{(\ell),i} \mathbf{G}_{(\ell),i}^{\top}\right) \\
&= \mathcal{T}_{(\ell),i}^{-1} \circ \left(\left(\mathbf{U}_{(\ell),i}\right)_{22} \lambda_i \begin{bmatrix} \delta_{(\ell)}^2 \lambda_i & \delta_{(\ell)} q_{(\ell)} \lambda_i \\ \delta_{(\ell)} q_{(\ell)} \lambda_i & q_{(\ell)}^2 \lambda_i \end{bmatrix}\right) \\
&= \left(\mathbf{U}_{(\ell),i}\right)_{22} \lambda_i \cdot \mathcal{T}_{(\ell),i}^{-1} \circ \begin{bmatrix} \delta_{(\ell)}^2 \lambda_i & \delta_{(\ell)} q_{(\ell)} \lambda_i \\ \delta_{(\ell)} q_{(\ell)} \lambda_i & q_{(\ell)}^2 \lambda_i \end{bmatrix} \\
&= \left(\mathbf{U}_{(\ell),i}\right)_{22} \lambda_i \mathbf{U}_{(\ell),i}.
\end{aligned}
\tag{130}
$$

Solving the recursion (129) yields the desired result. $\qquad\square$

The following lemma characterizes $\mathbf{U}_{(\ell),i}$ and $\mathbf{Q}_{(\ell),i}$.

**Lemma 23.** *With $\mathbf{U}_{(\ell),i}$ defined in (126), we have*

1. *By Equation (F.9) of Li et al. (2024), we have*

$$
\left(\mathbf{U}_{(\ell),i}\right)_{22} = \frac{\delta_{(\ell)}}{2} + \frac{(1+c)(q_{(\ell)} - \delta_{(\ell)})}{2\left(1 - c^2 + c\lambda_i(q_{(\ell)} + c\delta_{(\ell)})\right)};
\tag{131}
$$

2. *We have*

$$
\left(\mathbf{U}_{(\ell),i}\right)_{22} \leq \frac{\delta}{2} + \frac{1}{8752\psi\tilde{\kappa}\lambda_i \ln n}.
\tag{132}
$$

3. *By Equation (44) of Jain et al. (2018), we have $\left(\mathbf{U}_{(\ell),i}\right)_{11} = \left(1 - 2\delta_{(\ell)}\lambda_i\right)\left(\mathbf{U}_{(\ell),i}\right)_{22} + \delta_{(\ell)}^2 \lambda_i$;*

4. *We have $\left(\mathbf{U}_{(\ell),i}\right)_{11} \leq \left(\mathbf{U}_{(\ell),i}\right)_{22}$, and $\mathbf{U}_{(\ell),i} \preceq 2\left(\mathbf{U}_{(\ell),i}\right)_{22} \mathbf{I}$.*

5. $\mathbf{U}_{(\ell),i} \preceq \mathbf{Q}_{(\ell),i} \preceq \frac{4}{3}\mathbf{U}_{(\ell),i}$

6. *By Equation (56), (61) and (63) of Jain et al. (2018), we have*

$$
\begin{aligned}
\left(\mathbf{U}_{(\ell),i}\right)_{11} &= \frac{(1 + c - c\delta_i\lambda_i)(q_{(\ell)} - c\delta_{(\ell)}) - 2\delta_{(\ell)}\lambda_i(q_{(\ell)} - c\delta_{(\ell)}) + 2\delta_{(\ell)}^2 \lambda_i}{2(1 - c^2 + c\lambda_i(q_{(\ell)} + c\delta_{(\ell)}))}, \\
\left(\mathbf{U}_{(\ell),i}\right)_{12} &= \frac{\left(1 + c - \lambda_i(q_{(\ell)} + c\delta_{(\ell)})\right)(q_{(\ell)} - c\delta_{(\ell)}) + \delta_{(\ell)}\lambda_i(q_{(\ell)} + c\delta_{(\ell)})}{2(1 - c^2 + c\lambda_i(q_{(\ell)} + c\delta_{(\ell)}))}, \\
\left(\mathbf{U}_{(\ell),i}\right)_{22} &= \frac{(1 + c - c\delta_i\lambda_i)(q_{(\ell)} - c\delta_{(\ell)}) + 2cq_{(\ell)}\delta_{(\ell)}\lambda_i}{2(1 - c^2 + c\lambda_i(q_{(\ell)} + c\delta_{(\ell)}))}.
\end{aligned}
\tag{133}
$$

7. *We have $\mathbf{U}_{(\ell),i} \preceq 16\mathbf{U}_{(\ell+1),i}$.*

8. *We have $\mathbf{Q}_{(\ell),i} \preceq 20\mathbf{Q}_{(\ell+1),i}$.*

*Proof.* For Item 2, following the proof of Lemma F.5 in Li et al. (2024), we have

$$
\left(\mathbf{U}_{(\ell),i}\right)_{22} \leq \frac{\delta}{2} + \frac{\gamma\beta}{2\delta\lambda_i}.
\tag{134}
$$

Recall that $\beta = \frac{\delta}{4376\psi\tilde{\kappa}\gamma \ln n}$ by the parameter choice in Appendix A.1.3, we have

$$
\left(\mathbf{U}_{(\ell),i}\right)_{22} \leq \frac{\delta}{2} + \frac{1}{8752\psi\tilde{\kappa}\lambda_i \ln n}.
\tag{135}
$$

For Item 4, from Item 1, we know $\left(\mathbf{U}_{(\ell),i}\right)_{22} \geq \delta/2$. And from Item 3,

$$
\begin{aligned}
\left(\mathbf{U}_{(\ell),i}\right)_{11} &= \left(\mathbf{U}_{(\ell),i}\right)_{22} - 2\delta_{(\ell)}\lambda_i \left(\mathbf{U}_{(\ell),i}\right)_{22} + \delta_{(\ell)}^2 \lambda_i \\
&\leq \left(\mathbf{U}_{(\ell),i}\right)_{22} - 2\delta_{(\ell)}\lambda_i \cdot \frac{\delta_{(\ell)}}{2} + \delta_{(\ell)}^2 \lambda_i = \left(\mathbf{U}_{(\ell),i}\right)_{22}.
\end{aligned}
\tag{136}
$$

Thus, we have

$$\mathbf{U}_{(\ell),i} \preceq \left(\operatorname{tr} \mathbf{U}_{(\ell),i}\right) \mathbf{I} \leq 2 \left(\mathbf{U}_{(\ell),i}\right)_{22} \mathbf{I}. \tag{137}$$

For Item 5, since parameter choice procedure implies that $\left(\mathbf{U}_{(\ell),i}\right)_{22} \lambda_i \leq \frac{1}{4}$, we have

$$1 \leq \frac{1}{1 - \left(\mathbf{U}_{(\ell),i}\right)_{22} \lambda_i} \leq \frac{4}{3}. \tag{138}$$

Plugging this into (127) completes the proof.

For Item 7, from Item 6, we split the numerator of $\mathbf{U}_{(\ell),i}$ into two parts, based on whether the term contains $\lambda_i$,

$$\operatorname{num}\left(\mathbf{U}_{(\ell),i}\right)_{11} = \underbrace{(1+c)(q_{(\ell)} - c\delta_{(\ell)})}_{\mathbf{M}_{11}} + \underbrace{\left[-c\delta_i(q_{(\ell)} - c\delta_{(\ell)}) - 2\delta_{(\ell)}(q_{(\ell)} - c\delta_{(\ell)}) + 2\delta_{(\ell)}^2 \lambda_i\right] \lambda_i}_{\mathbf{N}_{11}},$$

$$\operatorname{num}\left(\mathbf{U}_{(\ell),i}\right)_{12} = \underbrace{(1+c)(q_{(\ell)} - c\delta_{(\ell)})}_{\mathbf{M}_{12}} + \underbrace{\left[-(q_{(\ell)} + c\delta_{(\ell)})(q_{(\ell)} - c\delta_{(\ell)}) + \delta_{(\ell)}(q_{(\ell)} + c\delta_{(\ell)})\right] \lambda_i}_{\mathbf{N}_{12}},$$

$$\operatorname{num}\left(\mathbf{U}_{(\ell),i}\right)_{22} = \underbrace{(1+c)(q_{(\ell)} - c\delta_{(\ell)})}_{\mathbf{M}_{22}} + \underbrace{\left[-c\delta_i(q_{(\ell)} - c\delta_{(\ell)}) + 2cq\delta_{(\ell)}\right] \lambda_i}_{\mathbf{N}_{22}}, \tag{139}$$

where $\operatorname{num}$ represents the numerator. Note that $\mathbf{M} = (1+c)(q_{(\ell)} - c\delta_{(\ell)}) \begin{bmatrix} 1 & 1 \\ 1 & 1 \end{bmatrix} \succeq \mathbf{O}$. Therefore,

$$\mathbf{U}_{(\ell+1),i} = \frac{\mathbf{M}/4 + \mathbf{N}/16}{2(1 - c^2 + c\lambda_i(q_{(\ell)}/4 + c\delta_{(\ell)}/4))}$$

$$\succeq \frac{\mathbf{M}/16 + \mathbf{N}/16}{2(1 - c^2 + c\lambda_i(q_{(\ell)} + c\delta_{(\ell)}))} = \frac{1}{16} \mathbf{U}_{(\ell),i}. \tag{140}$$

Thus, $\mathbf{U}_{(\ell),i} \preceq 16 \mathbf{U}_{(\ell+1),i}$.

For Item 8, parameter choice procedure implies that $\left(\mathbf{U}_{(\ell),i}\right)_{22} \lambda_i \leq \frac{1}{4}$. Thus, from Lemma 22 and Item 7, we have

$$\tilde{\mathbf{Q}}_{(\ell),i} = \frac{1}{1 - \left(\mathbf{U}_{(\ell),i}\right)_{22} \lambda_i} \mathbf{U}_{(\ell),i}$$

$$\preceq \frac{16}{1 - \left(\mathbf{U}_{(\ell),i}\right)_{22} \lambda_i} \mathbf{U}_{(\ell),i} = \frac{16(1 - \left(\mathbf{U}_{(\ell+1),i}\right)_{22} \lambda_i)}{1 - \left(\mathbf{U}_{(\ell),i}\right)_{22} \lambda_i} \mathbf{Q}_{(\ell+1),i} \tag{141}$$

$$\preceq \frac{16(1 - \left(\mathbf{U}_{(\ell),i}\right)_{22} \lambda_i/4)}{1 - \left(\mathbf{U}_{(\ell),i}\right)_{22} \lambda_i} \mathbf{Q}_{(\ell+1),i} \preceq 20 \mathbf{Q}_{(\ell+1),i},$$

where we uses that $\mathbf{U}_{(\ell),i}$ is PSD matrix and $\left(\mathbf{U}_{(\ell+1),i}\right)_{22} \geq \left(\mathbf{U}_{(\ell),i}\right)_{22} /4$. $\qquad\square$

### A.7.2 PROOF OF LEMMA 4

*Proof of Lemma 4.* We aim to bound $\left\langle \tilde{\mathbf{T}}, \tilde{\mathbf{C}}_n \right\rangle$ from above. By unrolling recursive definition of $\tilde{\mathbf{C}}_{t-1}$ in (44), we obtain

$$\tilde{\mathbf{C}}_n = \tilde{\mathcal{B}}_n \circ \tilde{\mathbf{C}}_{n-1} + \sigma^2 \begin{bmatrix} \delta_n^2 \mathbf{S} & \delta_n q_n \mathbf{S} \\ \delta_n q_n \mathbf{S} & q_n^2 \mathbf{S} \end{bmatrix}$$

$$= \sigma^2 \sum_{s=1}^{n} \tilde{\mathcal{B}}_n \circ \cdots \circ \tilde{\mathcal{B}}_{s+1} \circ \begin{bmatrix} \delta_s^2 \mathbf{S} & \delta_s q_s \mathbf{S} \\ \delta_s q_s \mathbf{S} & q_s^2 \mathbf{S} \end{bmatrix}. \tag{142}$$

Therefore, taking the inner product with $\tilde{\mathbf{T}}$ and using that $\tilde{\mathcal{B}}_{s,i} = \mathbf{A}_{s,i} \otimes \mathbf{A}_{s,i}$, we get

$$\left\langle \tilde{\mathbf{T}}, \tilde{\mathbf{C}}_n \right\rangle = \sigma^2 \sum_{i=1}^{d} t_{ii} \sum_{s=1}^{n} \left( \tilde{\mathcal{B}}_{n,i} \circ \cdots \circ \tilde{\mathcal{B}}_{s+1,i} \circ \begin{bmatrix} \delta_s^2 \lambda_i & \delta_s q_s \lambda_i \\ \delta_s q_s \lambda_i & q_s^2 \lambda_i \end{bmatrix} \right)_{11}, \tag{143}$$

where $t_{ii}$ denotes the $i$-th diagonal element of $\mathbf{T}$. In the following, we will bound each term of the sum $\sum_{i=1}^{d}$ separately.

Let $k^* = \max\left\{k : \lambda_k > \frac{16(1-c)\ln n}{(q-c\delta)K}\right\}$. For each $i$, define $\ell_i^* = \max\left\{\ell : \frac{\lambda_i}{4^{\ell-1}} > \frac{16(1-c)\ln n}{(q-c\delta)K}\right\}$. Note that $i \leq k^*$ implies $\ell_i^* \geq 1$.

If $i \leq k^*$, we bound $\sum_{s=1}^{n} = \sum_{s=1}^{K\ell_i^*} + \sum_{s=K\ell_i^*+1}^{n}$, respectively.

$$
\sum_{s=1}^{K\ell_i^*} \left(\tilde{\mathcal{B}}_{n,i} \circ \cdots \circ \tilde{\mathcal{B}}_{s+1,i} \circ \begin{bmatrix} \delta_s^2 \lambda_i & \delta_s q_s \lambda_i \\ \delta_s q_s \lambda_i & q_s^2 \lambda_i \end{bmatrix}\right)_{11}
$$

$$
= \sum_{m=1}^{\ell_i^*} \left(\tilde{\mathcal{B}}_{n,i} \circ \cdots \circ \tilde{\mathcal{B}}_{K(\ell_i^*+1)+1,i} \circ \tilde{\mathcal{B}}_{(\ell_i^*+1),i}^K \circ \cdots \circ \tilde{\mathcal{B}}_{(m+1),i}^K \circ \sum_{s=1}^{K} \tilde{\mathcal{B}}_{(m),i}^{K-s} \circ \begin{bmatrix} \delta_{(m)}^2 \lambda_i & \delta_{(m)} q_{(m)} \lambda_i \\ \delta_{(m)} q_{(m)} \lambda_i & q_{(m)}^2 \lambda_i \end{bmatrix}\right)_{11}
$$

$$
\overset{a}{\leq} \sigma^2 \sum_{m=1}^{\ell_i^*} \left(\tilde{\mathcal{B}}_{n,i} \circ \cdots \circ \tilde{\mathcal{B}}_{K(\ell_i^*+1)+1,i} \circ \tilde{\mathcal{B}}_{(\ell_i^*+1),i}^K \circ \cdots \circ \tilde{\mathcal{B}}_{(m+1),i}^K \circ \mathbf{Q}_{(m),i}\right)_{11}
$$

$$
\overset{b}{\leq} \sigma^2 \sum_{m=1}^{\ell_i^*} \left(\tilde{\mathcal{B}}_{n,i} \circ \cdots \circ \tilde{\mathcal{B}}_{K(\ell_i^*+1)+1,i} \circ \tilde{\mathcal{B}}_{(\ell_i^*+1),i}^K \circ \cdots \circ \tilde{\mathcal{B}}_{(m+1),i}^K \circ \left[\frac{8}{3}\left(\mathbf{U}_{(m),i}\right)_{22} \mathbf{I}\right]\right)_{11}
$$

$$
\leq \frac{8\left(\mathbf{U}_{(1),i}\right)_{22}}{3} \sum_{m=1}^{\ell-1} \left[\mathbf{A}_{n,i} \cdots \mathbf{A}_{K(\ell_i^*+1)+1,i} \mathbf{A}_{\ell_i^*,i}^K \cdots \mathbf{A}_{(m+1),i}^K \right.
$$

$$
\left. \left(\mathbf{A}_{(m+1),i}^K\right)^\top \cdots \left(\mathbf{A}_{(\ell-1),i}^K\right)^\top \mathbf{A}_{K(\ell_i^*+1)+1,i}^\top \cdots \mathbf{A}_{n,i}^\top\right]_{22}
$$

$$
\leq \frac{8\left(\mathbf{U}_{(1),i}\right)_{22}}{3} \sum_{m=1}^{\ell-1} \underbrace{\left\|\mathbf{A}_{n,i} \cdots \mathbf{A}_{K(\ell_i^*+1)+1,i}\right\|^2}_{\text{Lemma 19}} \underbrace{\left\|\mathbf{A}_{(\ell_i^*),i}^K\right\|^2}_{\text{Lemma 16}} \cdots \underbrace{\left\|\mathbf{A}_{(m+1),i}^K\right\|^2}_{\text{Lemma 16}}
$$

$$
\leq \frac{8\left(\mathbf{U}_{(1),i}\right)_{22}}{3} \cdot \frac{16}{(1-c)^2} \cdot \frac{6}{n^4(\log_2 n)^2} \cdot \log_2 n
$$

$$
\overset{c}{\leq} \frac{\left(\mathbf{U}_{(1),i}\right)_{22}}{256n^2},
$$

$$(144)$$

where $\overset{a}{\leq}$ uses the definition of $\mathbf{Q}_{(m),i}$, $\overset{b}{\leq}$ uses Lemma 23, and $\overset{c}{\leq}$ uses $n \geq 16$. For the second term, we have $\lambda_i/4^{\ell_i^*} \leq \frac{16(1-c)\ln n}{(q-c\delta)K} \leq \frac{(1-c)^2}{q-c\delta}$. Thus, we apply Lemma 20:

$$
\sum_{s=K\ell_i^*+1}^{n} \left(\tilde{\mathcal{B}}_{n,i} \circ \cdots \circ \tilde{\mathcal{B}}_{s+1,i} \circ \begin{bmatrix} \delta_s^2 \lambda_i & \delta_s q_s \lambda_i \\ \delta_s q_s \lambda_i & q_s^2 \lambda_i \end{bmatrix}\right)_{22}
$$

$$
\leq \sum_{s=K\ell_i^*+1}^{n} \lambda_i \underbrace{\left\|\mathbf{A}_{n,i} \cdots \mathbf{A}_{s+1,i} \begin{bmatrix} \delta_s \\ q_s \end{bmatrix}\right\|^2}_{\text{Lemma 20}} \leq 8\sigma^2 \sum_{s=K\ell_i^*+1}^{n} \lambda_i \left(\frac{q_s - c\delta_s}{1-c}\right)^2
$$

$$
= \frac{128\sigma^2}{15} \lambda_i K \left(\frac{q_{(\ell_i^*+1)} - c\delta_{(\ell_i^*+1)}}{1-c}\right)^2
$$

$$(145)$$

$$
= \frac{128\sigma^2}{15} \left(\frac{K\lambda_i}{4^{\ell_i^*}} \cdot \frac{q-c\delta}{1-c}\right) \left(\frac{q_{(\ell_i^*+1)} - c\delta_{(\ell_i^*+1)}}{1-c}\right)
$$

$$
\overset{a}{\leq} \frac{128\sigma^2}{15} \cdot \frac{16\ln n}{K} \cdot 4\left(\mathbf{U}_{(\ell_i^*+1),i}\right)_{22} \leq \frac{8192\sigma^2 \ln n}{15K}\left(\mathbf{U}_{(1),i}\right)_{22},
$$

where $\overset{a}{\leq}$ uses $\lambda_i/4^{\ell_i^*} \leq \frac{16(1-c)\ln n}{(q-c\delta)K}$ and from Lemma 23,

$$
\begin{aligned}
\left(\mathbf{U}_{(\ell_i^*+1),i}\right)_{22} &= \frac{\delta_{(\ell_i^*+1)}}{2} + \frac{(1+c)(q_{(\ell_i^*+1)} - \delta_{(\ell_i^*+1)})}{2\left(1 - c^2 + c\lambda_i(q_{(\ell_i^*+1)} + c\delta_{(\ell_i^*+1)})\right)} \\
&\geq \frac{\delta_{(\ell_i^*+1)}}{2} + \frac{(1+c)(q_{(\ell_i^*+1)} - \delta_{(\ell_i^*+1)})}{2\left(1 - c^2 + \frac{c\lambda_i}{4^{\ell_i^*}}(q + c\delta)\right)} \\
&\geq \frac{\delta_{(\ell_i^*+1)}}{2} + \frac{(1+c)(q_{(\ell_i^*+1)} - \delta_{(\ell_i^*+1)})}{2\left(1 - c^2 + \frac{c(1-c)^2(q+c\delta)}{(q-c\delta)}\right)} \\
&\geq \frac{\delta_{(\ell_i^*+1)}}{2} + \frac{(1+c)(q_{(\ell_i^*+1)} - \delta_{(\ell_i^*+1)})}{2\left(1 - c^2 + \frac{c(1-c)^2(q+cq)}{(q-cq)}\right)} \\
&= \frac{\delta_{(\ell_i^*+1)}}{2} + \frac{q_{(\ell_i^*+1)} - \delta_{(\ell_i^*+1)}}{2(1+c)(1-c)} \\
&\geq \frac{\delta_{(\ell_i^*+1)}}{4} + \frac{q_{(\ell_i^*+1)} - \delta_{(\ell_i^*+1)}}{4(1-c)} = \frac{q_{(\ell_i^*+1)} - c\delta_{(\ell_i^*+1)}}{4(1-c)}.
\end{aligned}
\tag{146}
$$

If $i > k^*$, we have

$$
\begin{aligned}
&\sum_{s=1}^{n} \left(\tilde{\mathcal{B}}_{n,i} \circ \cdots \circ \tilde{\mathcal{B}}_{s+1,i} \circ \begin{bmatrix} \delta_s^2 \lambda_i & \delta_s q_s \lambda_i \\ \delta_s q_s \lambda_i & q_s^2 \lambda_i \end{bmatrix}\right)_{22} \\
&\overset{a}{\leq} \sum_{s=1}^{n} \lambda_i \underbrace{\left\| \mathbf{A}_{n,i} \cdots \mathbf{A}_{s+1,i} \begin{bmatrix} \delta_s \\ q_s \end{bmatrix} \right\|^2}_{\text{Lemma 20}} \leq 8\sigma^2 \sum_{s=1}^{n} \lambda_i \left(\frac{q_s - c\delta_s}{1-c}\right)^2 \\
&= \frac{128}{15} \lambda_i K \left(\frac{q - c\delta}{1-c}\right)^2.
\end{aligned}
\tag{147}
$$

Finally, we have

$$
\begin{aligned}
\left\langle \tilde{\mathbf{T}}, \tilde{\mathbf{C}}_n \right\rangle &= \sigma^2 \sum_{i=1}^{k^*} t_{ii} \left(\frac{(\mathbf{U}_{(1),i})_{22}}{256N^2} + \frac{8192\ln n}{15K}(\mathbf{U}_{(1),i})_{22}\right) + \sigma^2 \sum_{i=k^*+1}^{d} t_{ii} \cdot \frac{128}{15} \lambda_i K \left(\frac{q - c\delta}{1-c}\right)^2 \\
&\leq \sigma^2 \left[\sum_{i=1}^{k^*} \frac{547 t_{ii} \ln n}{K\lambda_i}(\mathbf{U}_{(1),i})_{22} \lambda_i + \frac{128}{15} K \left(\frac{q - c\delta}{1-c}\right)^2 \sum_{i=k^*+1}^{d} \lambda_i t_{ii}\right] \\
&\leq \sigma^2 \left[\left(\sum_{i=1}^{k^*} \frac{547 t_{ii} \ln n}{K\lambda_i}\right)\left(\sum_{j=1}^{k^*} (\mathbf{U}_{(1),j})_{22} \lambda_j\right) + \frac{128}{15} K \left(\frac{q - c\delta}{1-c}\right)^2 \sum_{i=k^*+1}^{d} \lambda_i t_{ii}\right] \\
&\overset{a}{\leq} \sigma^2 \left[\sum_{i=1}^{k^*} \frac{t_{ii}}{2K\lambda_i} + \frac{128}{15} K \left(\frac{q - c\delta}{1-c}\right)^2 \sum_{i=k^*+1}^{d} \lambda_i t_{ii}\right],
\end{aligned}
\tag{148}
$$

where $\overset{a}{\leq}$ uses $\sum_i x_i y_i \leq \sum x_i \sum_j y_j$ if $x_i, y_i \geq 0$, and from the parameter choice procedure, we have $\sum_{j=1}^{k^*} (\mathbf{U}_{(1),j})_{22} \lambda_j \leq \frac{1}{1094\ln n}$. $\qquad\square$

### A.7.3 PROOF OF LEMMA 6

We bound the noise of $\tilde{\mathbf{C}}_t$ of two consecutive stages.

**Lemma 24.** *Let $\ell \geq 2$. If $K(\ell-1) + 1 \leq t \leq K(\ell+1)$, we have*

$$
\sum_{s=K(\ell-1)+1}^{t} \tilde{\mathcal{B}}_{t-1,i} \circ \cdots \circ \tilde{\mathcal{B}}_{s+1,i} \circ \begin{bmatrix} \delta_s^2 \lambda_i & \delta_s q_s \lambda_i \\ \delta_s q_s \lambda_i & q_s^2 \lambda_i \end{bmatrix} \preceq 20\mathbf{Q}_{(\ell+1),i}.
\tag{149}
$$

*Proof.* For $K(\ell-1)+1 \le t \le K\ell+1$, we have $t$ belongs to the $\ell-1$-th stage. From the definition of $\mathbf{Q}_{(\ell)}$, we have

$$
\sum_{s=K(\ell-1)+1}^{t} \tilde{\mathcal{B}}_{t-1,i} \circ \cdots \circ \tilde{\mathcal{B}}_{s+1,i} \circ \begin{bmatrix} \delta_s^2 \lambda_i & \delta_s q_s \lambda_i \\ \delta_s q_s \lambda_i & q_s^2 \lambda_i \end{bmatrix}
$$

$$
= \sum_{s=K(\ell-1)+1}^{t} \tilde{\mathcal{B}}_{(\ell)),i}^{t-K(\ell-1)} \circ \begin{bmatrix} \delta_s^2 \lambda_i & \delta_s q_s \lambda_i \\ \delta_s q_s \lambda_i & q_s^2 \lambda_i \end{bmatrix}
$$

$$
\preceq \sum_{s=K(\ell-1)+1}^{\infty} \tilde{\mathcal{B}}_{(\ell)),i}^{t-K(\ell-1)} \circ \begin{bmatrix} \delta_s^2 \lambda_i & \delta_s q_s \lambda_i \\ \delta_s q_s \lambda_i & q_s^2 \lambda_i \end{bmatrix} \stackrel{a}{=} \mathbf{Q}_{(\ell-1),i} \tag{150}
$$

$$
\stackrel{b}{\preceq} 20\mathbf{Q}_{(\ell),i},
$$

where $\stackrel{a}{=}$ uses the definition of $\mathbf{Q}_{(\ell)}$ and $\stackrel{b}{\preceq}$ uses Lemma 23.

For $K\ell+1 \le t \le K(\ell+1)$, we prove by induction. The case where $t = K(\ell+1)$ has been proven. We suppose (149) holds. Note that by the definition of $\mathbf{Q}_{(\ell),i}$, we have

$$
\mathbf{Q}_{(\ell),i} = (\mathcal{I}-\tilde{\mathcal{B}}_{(\ell),i})^{-1} \circ \begin{bmatrix} \delta_{(\ell)}^2 \lambda_i & \delta_{(\ell)} q_{(\ell)} \lambda_i \\ \delta_{(\ell)} q_{(\ell)} \lambda_i & q_{(\ell)}^2 \lambda_i \end{bmatrix} \implies \tilde{\mathcal{B}}_{(\ell),i} \circ \mathbf{Q}_{(\ell),i} = \mathbf{Q}_{(\ell),i} - \begin{bmatrix} \delta_{(\ell)}^2 \lambda_i & \delta_{(\ell)} q_{(\ell)} \lambda_i \\ \delta_{(\ell)} q_{(\ell)} \lambda_i & q_{(\ell)}^2 \lambda_i \end{bmatrix}. \tag{151}
$$

Therefore, for $t+1$, we have

$$
\sum_{s=K(\ell-1)+1}^{t+1} \tilde{\mathcal{B}}_{t,i} \circ \cdots \circ \tilde{\mathcal{B}}_{s+1,i} \circ \begin{bmatrix} \delta_s^2 \lambda_i & \delta_s q_s \lambda_i \\ \delta_s q_s \lambda_i & q_s^2 \lambda_i \end{bmatrix}
$$

$$
= \tilde{\mathcal{B}}_{(\ell),i} \circ \sum_{s=K(\ell-1)+1}^{t} \tilde{\mathcal{B}}_{t,i} \circ \cdots \circ \tilde{\mathcal{B}}_{s+1,i} \circ \begin{bmatrix} \delta_s^2 \lambda_i & \delta_s q_s \lambda_i \\ \delta_s q_s \lambda_i & q_s^2 \lambda_i \end{bmatrix} + \begin{bmatrix} \delta_s^2 \lambda_i & \delta_s q_s \lambda_i \\ \delta_s q_s \lambda_i & q_s^2 \lambda_i \end{bmatrix} \tag{152}
$$

$$
\stackrel{a}{\preceq} \tilde{\mathcal{B}}_{(\ell),i} \circ \left( 20\mathbf{Q}_{(\ell),i} \right) + \begin{bmatrix} \delta_s^2 \lambda_i & \delta_s q_s \lambda_i \\ \delta_s q_s \lambda_i & q_s^2 \lambda_i \end{bmatrix}
$$

$$
= 20\mathbf{Q}_{(\ell),i} - 19 \begin{bmatrix} \delta_s^2 \lambda_i & \delta_s q_s \lambda_i \\ \delta_s q_s \lambda_i & q_s^2 \lambda_i \end{bmatrix} \preceq 20\mathbf{Q}_{(\ell),i}.
$$

By induction, the lemma holds. $\qquad\square$

Now, we are ready for the proof.

*Proof of Lemma 6.* Our goal is to show that for $1 \le t \le n$, we have

$$
\mathbb{E}\left[ \widehat{\mathbf{G}}_t \otimes \widehat{\mathbf{G}}_t \right] \circ \tilde{\mathbf{C}}_{t-1} \preceq \frac{1}{2}\sigma^2 \begin{bmatrix} \delta_t^2 \mathbf{S} & \delta_t q_t \mathbf{S} \\ \delta_t q_t \mathbf{S} & q_t^2 \mathbf{S} \end{bmatrix}. \tag{153}
$$

Note that by Lemma 1, we have $\mathbb{E}\left[ \widehat{\mathbf{G}}_t \otimes \widehat{\mathbf{G}}_t \right] \circ \tilde{\mathbf{C}}_{t-1} \preceq \psi \left\langle \begin{bmatrix} \mathbf{O} & \mathbf{O} \\ \mathbf{O} & \mathbf{S} \end{bmatrix}, \tilde{\mathbf{C}}_{t-1} \right\rangle \begin{bmatrix} \delta_t^2 \mathbf{S} & \delta_t q_t \mathbf{S} \\ \delta_t q_t \mathbf{S} & q_t^2 \mathbf{S} \end{bmatrix}$.

Therefore, we only have to show that for all $1 \le i \le d$,

$$
\psi \left\langle \begin{bmatrix} \mathbf{O} & \mathbf{O} \\ \mathbf{O} & \mathbf{S} \end{bmatrix}, \tilde{\mathbf{C}}_{t-1} \right\rangle \le \frac{1}{2}\sigma^2. \tag{154}
$$

From the recursive definition of $\tilde{\mathbf{C}}_{t-1}$ in (44), we have:

$$
\tilde{\mathbf{C}}_{t-1} = \tilde{\mathcal{B}}_{t-1} \circ \tilde{\mathbf{C}}_{t-2} + \sigma^2 \begin{bmatrix} \delta_{t-1}^2 \mathbf{S} & \delta_{t-1} q_{t-1} \mathbf{S} \\ \delta_{t-1} q_{t-1} \mathbf{S} & q_{t-1}^2 \mathbf{S} \end{bmatrix}
$$

$$
= \sigma^2 \sum_{s=1}^{t-1} \tilde{\mathcal{B}}_{t-1} \circ \cdots \circ \tilde{\mathcal{B}}_{s+1} \circ \begin{bmatrix} \delta_s^2 \mathbf{S} & \delta_s q_s \mathbf{S} \\ \delta_s q_s \mathbf{S} & q_s^2 \mathbf{S} \end{bmatrix}. \tag{155}
$$

Therefore, taking the inner product with $\begin{bmatrix} \mathbf{O} & \mathbf{O} \\ \mathbf{O} & \mathbf{S} \end{bmatrix}$ and using that $\tilde{\mathcal{B}}_{s,i} = \mathbf{A}_{s,i} \otimes \mathbf{A}_{s,i}$, we get

$$
\left\langle \begin{bmatrix} \mathbf{O} & \mathbf{O} \\ \mathbf{O} & \mathbf{S} \end{bmatrix}, \tilde{\mathbf{C}}_{t-1} \right\rangle = \sigma^2 \sum_{i=1}^{d} \lambda_i \sum_{s=1}^{t-1} \left( \tilde{\mathcal{B}}_{t-1,i} \circ \cdots \circ \tilde{\mathcal{B}}_{s+1,i} \circ \begin{bmatrix} \delta_s^2 \lambda_i & \delta_s q_s \lambda_i \\ \delta_s q_s \lambda_i & q_s^2 \lambda_i \end{bmatrix} \right)_{22}. \tag{156}
$$

Suppose $t-1$ belongs to the $\ell$-th stage, namely, $K(\ell-1)+1 \le t-1 \le K\ell$. For each $i$, define $\ell_i^* = \max\left\{ \ell : \frac{\lambda_i}{4^{\ell-1}} > \frac{16(1-c)\ln n}{(q-c\delta)K} \right\}$.

If $\ell \le \ell_i^* + 1$, we bound $\sum_{s=1}^{t-1} = \sum_{s=1}^{K(\ell-1)} + \sum_{s=K(\ell-1)+1}^{t-1}$, respectively. For the first term, we have

$$
\begin{aligned}
&\sum_{s=1}^{K(\ell-1)} \left( \tilde{\mathcal{B}}_{t-1,i} \circ \cdots \circ \tilde{\mathcal{B}}_{s+1,i} \circ \begin{bmatrix} \delta_s^2 \lambda_i & \delta_s q_s \lambda_i \\ \delta_s q_s \lambda_i & q_s^2 \lambda_i \end{bmatrix} \right)_{22} \\
&= \sum_{m=1}^{\ell-1} \left( \tilde{\mathcal{B}}_{(\ell),i}^{t-1-K(\ell-1)} \circ \tilde{\mathcal{B}}_{(\ell-1),i}^{K} \circ \cdots \circ \tilde{\mathcal{B}}_{(m+1),i}^{K} \circ \sum_{s=1}^{K} \tilde{\mathcal{B}}_{(m),i}^{K-s} \circ \begin{bmatrix} \delta_{(m)}^2 \lambda_i & \delta_{(m)} q_{(m)} \lambda_i \\ \delta_{(m)} q_{(m)} \lambda_i & q_{(m)}^2 \lambda_i \end{bmatrix} \right)_{22} \\
&\overset{a}{\preceq} \sum_{m=1}^{\ell-1} \left( \tilde{\mathcal{B}}_{(\ell),i}^{t-1-K(\ell-1)} \circ \tilde{\mathcal{B}}_{(\ell-1),i}^{K} \circ \cdots \circ \tilde{\mathcal{B}}_{(m+1),i} \circ \mathbf{Q}_{(m),i} \right)_{22} \\
&\overset{b}{\preceq} \sum_{m=1}^{\ell-1} \left( \tilde{\mathcal{B}}_{(\ell),i}^{t-1-K(\ell-1)} \circ \tilde{\mathcal{B}}_{(\ell-1),i}^{K} \circ \cdots \circ \tilde{\mathcal{B}}_{(m+1),i} \circ \left[ \frac{8}{3} \left( \mathbf{U}_{(m),i} \right)_{22} \mathbf{I} \right] \right)_{22} \\
&\le \frac{8 \left( \mathbf{U}_{(1),i} \right)_{22}}{3} \sum_{m=1}^{\ell-1} \left( \mathbf{A}_{(\ell),i}^{t-1-K(\ell-1)} \mathbf{A}_{(\ell-1),i}^{K} \cdots \mathbf{A}_{(m+1),i}^{K} \left( \mathbf{A}_{(m+1),i}^{K} \right)^{\top} \cdots \left( \mathbf{A}_{(\ell-1),i}^{K} \right)^{\top} \left( \mathbf{A}_{(\ell),i}^{t-1-K(\ell-1)} \right)^{\top} \right)_{22} \\
&\le \frac{8 \left( \mathbf{U}_{(1),i} \right)_{22}}{3} \sum_{m=1}^{\ell-1} \underbrace{\left\| \mathbf{A}_{(\ell),i}^{t-1-K(\ell-1)} \right\|^2}_{\text{Lemma 15}} \underbrace{\left\| \mathbf{A}_{(\ell-1),i}^{K} \right\|^2}_{\text{Lemma 16}} \cdots \underbrace{\left\| \mathbf{A}_{(m+1),i}^{K} \right\|^2}_{\text{Lemma 16}} \\
&\le \frac{8 \left( \mathbf{U}_{(1),i} \right)_{22}}{3} \cdot 6K^2 \cdot \frac{6}{n^4 (\log_2 n)^2} \cdot \log_2 n \\
&\overset{c}{\le} \frac{3 \left( \mathbf{U}_{(1),i} \right)_{22}}{2N^2},
\end{aligned}
\tag{157}
$$

where $\overset{a}{\preceq}$ uses the definition of $\mathbf{Q}_{(m),i}$, $\overset{b}{\preceq}$ uses $\mathbf{Q}_{(m),i} \preceq \frac{4}{3} \mathbf{U}_{(m),i} \preceq \frac{8}{3} \left( \mathbf{U}_{(m),i} \right)_{22} \mathbf{I}$ from Lemma 23, and $\overset{c}{\le}$ uses $n \ge 16$. For the second term, we apply Lemma 24,

$$
\begin{aligned}
&\sum_{s=K(\ell-1)+1}^{t-1} \left( \tilde{\mathcal{B}}_{t-1,i} \circ \cdots \circ \tilde{\mathcal{B}}_{s+1,i} \circ \begin{bmatrix} \delta_s^2 \lambda_i & \delta_s q_s \lambda_i \\ \delta_s q_s \lambda_i & q_s^2 \lambda_i \end{bmatrix} \right)_{22} \\
&\le \left( 20 \mathbf{Q}_{(\ell),i} \right)_{22} \le \frac{80}{3} \left( \mathbf{U}_{(\ell),i} \right)_{22} \le \frac{80}{3} \left( \mathbf{U}_{(1),i} \right)_{22}.
\end{aligned}
\tag{158}
$$

Thus, we have

$$
\left\langle \begin{bmatrix} \mathbf{O} & \mathbf{O} \\ \mathbf{O} & \mathbf{S} \end{bmatrix}, \tilde{\mathbf{C}}_{t-1} \right\rangle \le \sigma^2 \sum_{i=1}^{d} \left( \frac{3}{2N^2} + \frac{80}{3} \right) \left( \mathbf{U}_{(1),i} \right)_{22} \lambda_i \le \frac{1}{2} \sigma^2. \tag{159}
$$

If $\ell > \ell_i^* + 1$, we have $\lambda_i/4^{\ell-1} \in I_1$. We bound $\sum_{s=1}^{t-1} = \sum_{s=1}^{K\ell_i^*} + \sum_{s=K\ell_i^*+1}^{t-1}$, respectively. The bound of the first term parallels (144):

$$\sigma^2 \sum_{s=1}^{K\ell_i^*} \left( \tilde{\mathcal{B}}_{t-1,i} \circ \cdots \circ \tilde{\mathcal{B}}_{s+1,i} \circ \begin{bmatrix} \delta_s^2 \lambda_i & \delta_s q_s \lambda_i \\ \delta_s q_s \lambda_i & q_s^2 \lambda_i \end{bmatrix} \right)_{22}$$

$$= \sigma^2 \sum_{m=1}^{\ell_i^*} \left( \tilde{\mathcal{B}}_{t-1,i} \circ \cdots \circ \tilde{\mathcal{B}}_{K(\ell_i^*+1)+1,i} \circ \tilde{\mathcal{B}}_{(\ell_i^*+1),i}^K \circ \cdots \circ \tilde{\mathcal{B}}_{(m+1),i}^K \circ \sum_{s=1}^{K} \tilde{\mathcal{B}}_{(m),i}^{K-s} \circ \begin{bmatrix} \delta_{(m)}^2 \lambda_i & \delta_{(m)} q_{(m)} \lambda_i \\ \delta_{(m)} q_{(m)} \lambda_i & q_{(m)}^2 \lambda_i \end{bmatrix} \right)_{22}$$

$$\overset{a}{\preceq} \sigma^2 \sum_{m=1}^{\ell_i^*} \left( \tilde{\mathcal{B}}_{t-1,i} \circ \cdots \circ \tilde{\mathcal{B}}_{K(\ell_i^*+1)+1,i} \circ \tilde{\mathcal{B}}_{(\ell_i^*+1),i}^K \circ \cdots \circ \tilde{\mathcal{B}}_{(m+1),i}^K \circ \mathbf{Q}_{(m),i} \right)_{22}$$

$$\overset{b}{\preceq} \sigma^2 \sum_{m=1}^{\ell_i^*} \left( \tilde{\mathcal{B}}_{t-1,i} \circ \cdots \circ \tilde{\mathcal{B}}_{K(\ell_i^*+1)+1,i} \circ \tilde{\mathcal{B}}_{(\ell_i^*+1),i}^K \circ \cdots \circ \tilde{\mathcal{B}}_{(m+1),i}^K \circ \left[ \frac{8}{3} \left( \mathbf{U}_{(m),i} \right)_{22} \mathbf{I} \right] \right)_{22}$$

$$\leq \frac{8\sigma^2 \left( \mathbf{U}_{(1),i} \right)_{22}}{3} \sum_{m=1}^{\ell-1} \Big[ \mathbf{A}_{t-1,i} \cdots \mathbf{A}_{K(\ell_i^*+1)+1,i} \mathbf{A}_{\ell_i^*,i}^K \cdots \mathbf{A}_{(m+1),i}^K$$

$$\left( \mathbf{A}_{(m+1),i}^K \right)^\top \cdots \left( \mathbf{A}_{(\ell-1),i}^K \right)^\top \mathbf{A}_{K(\ell_i^*+1)+1,i}^\top \cdots \mathbf{A}_{t-1,i}^\top \Big]_{22}$$

$$\leq \frac{8\sigma^2 \left( \mathbf{U}_{(1),i} \right)_{22}}{3} \sum_{m=1}^{\ell-1} \underbrace{\left\| \mathbf{A}_{t-1,i} \cdots \mathbf{A}_{K(\ell_i^*+1)+1,i} \right\|^2}_{\text{Lemma 19}} \underbrace{\left\| \mathbf{A}_{(\ell_i^*),i}^K \right\|^2}_{\text{Lemma 16}} \cdots \underbrace{\left\| \mathbf{A}_{(m+1),i}^K \right\|^2}_{\text{Lemma 16}}$$

$$\leq \frac{8\sigma^2 \left( \mathbf{U}_{(1),i} \right)_{22}}{3} \cdot \frac{16}{(1-c)^2} \cdot \frac{6}{n^4 (\log_2 n)^2} \cdot \log_2 n$$

$$\overset{c}{\leq} \frac{\sigma^2 \left( \mathbf{U}_{(1),i} \right)_{22}}{256 n^2},$$

(160)

where $\overset{a}{\preceq}$ uses the definition of $\mathbf{Q}_{(m),i}$, $\overset{b}{\preceq}$ uses Lemma 23, and $\overset{c}{\leq}$ uses $n \geq 16$. For the second term, we have $\lambda_i/4^{\ell_i^*} \leq \frac{16(1-c)\ln n}{(q-c\delta)K} \leq \frac{(1-c)^2}{q-c\delta}$. Thus, we apply Lemma 20:

$$\sigma^2 \sum_{s=K\ell_i^*+1}^{t-1} \left( \tilde{\mathcal{B}}_{t-1,i} \circ \cdots \circ \tilde{\mathcal{B}}_{s+1,i} \circ \begin{bmatrix} \delta_s^2 \lambda_i & \delta_s q_s \lambda_i \\ \delta_s q_s \lambda_i & q_s^2 \lambda_i \end{bmatrix} \right)_{22}$$

$$\leq \sigma^2 \sum_{s=K\ell_i^*+1}^{t-1} \lambda_i \underbrace{\left\| \mathbf{A}_{t-1,i} \cdots \mathbf{A}_{s+1,i} \begin{bmatrix} \delta_s \\ q_s \end{bmatrix} \right\|^2}_{\text{Lemma 20}} \leq 8\sigma^2 \sum_{s=K\ell_i^*+1}^{t-1} \lambda_i \left( \frac{q_s - c\delta_s}{1-c} \right)^2$$

(161)

$$= \frac{128\sigma^2}{15} \lambda_i K \left( \frac{q_{(\ell_i^*+1)} - c\delta_{(\ell_i^*+1)}}{1-c} \right)^2$$

$$= \frac{128\sigma^2}{15} \left( \frac{K\lambda_i}{4^{\ell_i^*}} \cdot \frac{q-c\delta}{1-c} \right) \left( \frac{q_{(\ell_i^*+1)} - c\delta_{(\ell_i^*+1)}}{1-c} \right)$$

$$\overset{a}{\leq} \frac{128\sigma^2}{15} \cdot 16\ln n \cdot 4 \left( \mathbf{U}_{(\ell_i^*+1),i} \right)_{22} \leq \frac{8192\sigma^2 \ln n}{15} \left( \mathbf{U}_{(1),i} \right)_{22},$$

where $\overset{a}{\leq}$ uses $\lambda_i/4^{\ell_i^*} \leq \frac{16(1-c)\ln n}{(q-c\delta)K}$ and from Lemma 23,

$$
\begin{aligned}
\left(\mathbf{U}_{(\ell_i^*+1),i}\right)_{22} &= \frac{\delta_{(\ell_i^*+1)}}{2} + \frac{(1+c)(q_{(\ell_i^*+1)} - \delta_{(\ell_i^*+1)})}{2\left(1-c^2 + c\lambda_i(q_{(\ell_i^*+1)} + c\delta_{(\ell_i^*+1)})\right)} \\
&\geq \frac{\delta_{(\ell_i^*+1)}}{2} + \frac{(1+c)(q_{(\ell_i^*+1)} - \delta_{(\ell_i^*+1)})}{2\left(1-c^2 + \frac{c\lambda_i}{4^{\ell_i^*}}(q + c\delta)\right)} \\
&\geq \frac{\delta_{(\ell_i^*+1)}}{2} + \frac{(1+c)(q_{(\ell_i^*+1)} - \delta_{(\ell_i^*+1)})}{2\left(1-c^2 + \frac{c(1-c)^2(q+c\delta)}{(q-c\delta)}\right)} \\
&\geq \frac{\delta_{(\ell_i^*+1)}}{2} + \frac{(1+c)(q_{(\ell_i^*+1)} - \delta_{(\ell_i^*+1)})}{2\left(1-c^2 + \frac{c(1-c)^2(q+cq)}{(q-cq)}\right)} \\
&= \frac{\delta_{(\ell_i^*+1)}}{2} + \frac{q_{(\ell_i^*+1)} - \delta_{(\ell_i^*+1)}}{2(1+c)(1-c)} \\
&\geq \frac{\delta_{(\ell_i^*+1)}}{4} + \frac{q_{(\ell_i^*+1)} - \delta_{(\ell_i^*+1)}}{4(1-c)} = \frac{q_{(\ell_i^*+1)} - c\delta_{(\ell_i^*+1)}}{4(1-c)}.
\end{aligned}
\tag{162}
$$

Thus, we have

$$
\begin{aligned}
\left\langle \begin{bmatrix} \mathbf{O} & \mathbf{O} \\ \mathbf{O} & \mathbf{S} \end{bmatrix}, \tilde{\mathbf{C}}_{t-1} \right\rangle &\leq \sigma^2 \sum_{i=1}^d \left( \frac{1}{256N^2} + \frac{8192\ln n}{15} \right) \left(\mathbf{U}_{(1),i}\right)_{22} \lambda_i \\
&\leq 547\sigma^2 \ln n \sum_{i=1}^d \left(\mathbf{U}_{(1),i}\right)_{22} \lambda_i \leq \frac{1}{2}\sigma^2.
\end{aligned}
\tag{163}
$$

$\square$

## A.8 Bias Upper Bound

### A.8.1 Proof of Lemma 9

*Proof of Lemma 9.* Recall the definition of $\tilde{\mathbf{B}}_n$ and $\tilde{\mathcal{B}}_t$, we have

$$
\tilde{\mathbf{B}}_n = \tilde{\mathcal{B}}_n \circ \tilde{\mathcal{B}}_{n-1} \circ \cdots \circ \tilde{\mathcal{B}}_1 \circ \mathbf{B}_0 = \left( \prod_{t=1}^n \mathbf{A}_t \begin{bmatrix} \mathbf{w}_0 - \mathbf{w}^* \\ \mathbf{w}_0 - \mathbf{w}^* \end{bmatrix} \right) \left( \prod_{t=1}^n \mathbf{A}_t \begin{bmatrix} \mathbf{w}_0 - \mathbf{w}^* \\ \mathbf{w}_0 - \mathbf{w}^* \end{bmatrix} \right)^\top.
\tag{164}
$$

Note that $\mathbf{A}_t$ is block-diagonal, we have

$$
\left( \prod_{t=1}^n \mathbf{A}_t \begin{bmatrix} \mathbf{w}_0 - \mathbf{w}^* \\ \mathbf{w}_0 - \mathbf{w}^* \end{bmatrix} \right)_i = (\mathbf{w}_{0,i} - \mathbf{w}_i^*) \prod_{t=1}^n \mathbf{A}_{t,i} \begin{bmatrix} 1 \\ 1 \end{bmatrix}.
\tag{165}
$$

For $i \leq k^*$, we have $\lambda_i \in I_1$. Let $\ell^* = \max\left\{ \ell : \frac{\lambda_i}{4^{\ell-1}} > \frac{16(1-c)\ln n}{(q-c\delta)K} \right\}$, and note that for $\ell \geq \ell^*$, $\lambda_i/4^{\ell-1} \leq \frac{(1-c)^2}{q-c\delta}$. Therefore, we have

$$
\begin{aligned}
\left( \prod_{t=1}^n \mathbf{A}_{t,i} \begin{bmatrix} 1 \\ 1 \end{bmatrix} \right)_1^2 &\leq \underbrace{\left\| \prod_{t=K\ell^*+1}^n \mathbf{A}_t \right\|^2}_{\text{Lemma 19}} \underbrace{\left\| \mathbf{A}_{(\ell-1)}^K \right\|^2 \cdots \left\| \mathbf{A}_{(1)}^K \right\|^2}_{\text{Lemma 16}} \left\| \begin{bmatrix} 1 \\ 1 \end{bmatrix} \right\|^2 \\
&\leq \frac{16}{(1-c)^2} \cdot \frac{6}{n^4(\log_2 n)^2} \cdot 2 \\
&\overset{a}{\leq} \frac{1}{8n^2(\log_2 n)^4}.
\end{aligned}
\tag{166}
$$

where $\overset{a}{\leq}$ uses $K(1-c) \geq 16 \ln n$. For $i > k^*$, from Lemma 21 we have

$$\left(\prod_{t=1}^{n} \mathbf{A}_{t,i} \begin{bmatrix} 1 \\ 1 \end{bmatrix}\right)_1^2 \leq \left\|\prod_{t=1}^{n} \mathbf{A}_{t,i} \begin{bmatrix} 1 \\ 1 \end{bmatrix}\right\|^2 \leq 4. \tag{167}$$

Consider the following decomposition:

$$\prod_{t=1}^{n} \mathbf{A}_t \begin{bmatrix} \mathbf{w}_0 - \mathbf{w}^* \\ \mathbf{w}_0 - \mathbf{w}^* \end{bmatrix} = \begin{bmatrix} \boldsymbol{\xi}_1 \\ \mathbf{O} \end{bmatrix} + \begin{bmatrix} \mathbf{O} \\ \boldsymbol{\xi}_2 \end{bmatrix}, \tag{168}$$

where $\boldsymbol{\xi}_1 \in \mathbb{R}^{k^*}$ and $\boldsymbol{\xi}_2 \in \mathbb{R}^{d-k^*}$. Then (166) and (166) implies that

$$\left(\begin{bmatrix} \boldsymbol{\xi}_1 \\ \mathbf{O} \end{bmatrix}\right)_i^2 \leq \frac{(\mathbf{w}_0 - \mathbf{w}^*)_i^2}{8n^2(\log_2 n)^4}, \quad \left(\begin{bmatrix} \mathbf{O} \\ \boldsymbol{\xi}_2 \end{bmatrix}\right)_i^2 \leq 4(\mathbf{w}_0 - \mathbf{w}^*)_i^2. \tag{169}$$

Note that $\mathbf{T} \preceq 2\mathbf{T}_{0:k^*} + 2\mathbf{T}_{k^*:\infty}$. Then we have

$$\begin{aligned} \left\langle \tilde{\mathbf{T}}, \tilde{\mathbf{B}}_n \right\rangle &\leq 2 \left\langle \mathbf{T}_{0:k^*}, \tilde{\mathbf{B}}_n \right\rangle + 2 \left\langle \mathbf{T}_{k^*:\infty}, \tilde{\mathbf{B}}_n \right\rangle \\ &= 2 \left\| \begin{bmatrix} \boldsymbol{\xi}_1 \\ \mathbf{O} \end{bmatrix} \right\|_{\mathbf{T}_{0:k^*}}^2 + 2 \left\| \begin{bmatrix} \mathbf{O} \\ \boldsymbol{\xi}_2 \end{bmatrix} \right\|_{\mathbf{T}_{k^*:\infty}}^2 \\ &\leq \max_{\mathbf{w} \in S(\mathbf{w}_0 - \mathbf{w}^*)} \frac{\|\mathbf{w}\|_{\mathbf{T}_{0:k^*}}^2}{8n^2(\log_2 n)^4} + 4 \|\mathbf{w}\|_{\mathbf{T}_{k^*:\infty}}^2 . \end{aligned} \tag{170}$$

This completes the proof. $\qquad\square$

### A.8.2 PROOF OF LEMMA 11

We first analyze $\left\langle \begin{bmatrix} \mathbf{O} & \mathbf{O} \\ \mathbf{O} & \mathbf{S} \end{bmatrix}, \tilde{\mathbf{B}}_t \right\rangle$.

**Lemma 25.** *For $t \leq K$, we have*

$$\left\langle \begin{bmatrix} \mathbf{O} & \mathbf{O} \\ \mathbf{O} & \mathbf{S} \end{bmatrix}, \tilde{\mathbf{B}}_t \right\rangle \leq 4 \sum_{i=1}^{d} \lambda_i \left(\mathbf{w}_i^*\right)^2 . \tag{171}$$

*For $t > K$, we have*

$$\left\langle \begin{bmatrix} \mathbf{O} & \mathbf{O} \\ \mathbf{O} & \mathbf{S} \end{bmatrix}, \tilde{\mathbf{B}}_t \right\rangle \leq \frac{36}{n^2 (\log_2 n)^4} \sum_{i=1}^{k^*} \lambda_i \left(\mathbf{w}_i^*\right)^2 + 4 \sum_{i=k^*+1}^{d} \lambda_i \left(\mathbf{w}_i^*\right)^2 . \tag{172}$$

*Proof.* Note that $\tilde{\mathbf{B}}_t$ is block-diagonal, we have

$$\left\langle \begin{bmatrix} \mathbf{O} & \mathbf{O} \\ \mathbf{O} & \mathbf{S} \end{bmatrix}, \tilde{\mathbf{B}}_t \right\rangle = \sum_{i=1}^{d} \lambda_i \left(\mathbf{w}_i^*\right)^2 \left(\prod_{s=1}^{t} \mathbf{A}_{s,i} \begin{bmatrix} 1 \\ 1 \end{bmatrix}\right)_1^2 . \tag{173}$$

For $t \leq K$, $s \leq t$ implies $s$ belongs to the first stage. Thus, $\mathbf{A}_{s,i} = \mathbf{A}_{(\ell).i} = \mathbf{A}(\lambda_i)$. By Lemma 17,

$$\left| \left(\prod_{s=1}^{t} \mathbf{A}_{s,i} \begin{bmatrix} 1 \\ 1 \end{bmatrix}\right)_1 \right| = \left| \left(\mathbf{A}^t(\lambda_i) \begin{bmatrix} 1 \\ 1 \end{bmatrix}\right)_2 \right| \leq 2. \tag{174}$$

Therefore,

$$\begin{aligned} \left\langle \begin{bmatrix} \mathbf{O} & \mathbf{O} \\ \mathbf{O} & \mathbf{S} \end{bmatrix}, \tilde{\mathbf{B}}_t \right\rangle &= \sum_{i=1}^{d} \lambda_i \left(\mathbf{w}_i^*\right)^2 \left(\prod_{s=1}^{t} \mathbf{A}_{s,i} \begin{bmatrix} 1 \\ 1 \end{bmatrix}\right)_1^2 \\ &\leq 4 \sum_{i=1}^{d} \lambda_i \left(\mathbf{w}_i^*\right)^2 \end{aligned} \tag{175}$$

For $t > K$, suppose $t$ belongs to the $\ell$-th stage, we have $\ell \geq 2$. Since $i > k^*$ implies that $\lambda_i \in I_1$. Let $\ell_i^* = \max\left\{\ell : \frac{\lambda_i}{4^{\ell-1}} > \frac{16(1-c)\ln n}{(q-c\delta)K}\right\}$. If $\ell < \ell_i^*$, by applying Lemma 15 and Lemma 16, we have

$$
\left(\prod_{s=1}^{t} \mathbf{A}_{s,i} \begin{bmatrix} 1 \\ 1 \end{bmatrix}\right)_1^2 \leq \underbrace{\left\|\mathbf{A}_{(\ell)}^{t-K(\ell-1)}\right\|^2}_{\text{Lemma 15}} \underbrace{\left\|\mathbf{A}_{(\ell-1)}^{K}\right\|^2 \cdots \left\|\mathbf{A}_{(1)}^{K}\right\|^2}_{\text{Lemma 16}} \left\|\begin{bmatrix} 1 \\ 1 \end{bmatrix}\right\|^2
$$

$$
\leq 6K^2 \cdot \left(\frac{\sqrt{6}}{n^2 \log_2 n}\right)^{2(\ell-1)} \leq \frac{36}{n^2 (\log_2 n)^4}. \tag{176}
$$

If $\ell \geq \ell_i^*$, by applying Lemma 19 and Lemma 16, we have

$$
\left(\prod_{s=1}^{t} \mathbf{A}_{s,i} \begin{bmatrix} 1 \\ 1 \end{bmatrix}\right)_1^2 \leq \underbrace{\left\|\prod_{t=K\ell_i^*+1}^{n} \mathbf{A}_t\right\|^2}_{\text{Lemma 19}} \underbrace{\left\|\mathbf{A}_{(\ell_i^*-1)}^{K}\right\|^2 \cdots \left\|\mathbf{A}_{(1)}^{K}\right\|^2}_{\text{Lemma 16}} \left\|\begin{bmatrix} 1 \\ 1 \end{bmatrix}\right\|^2
$$

$$
\leq \frac{16}{(1-c)^2} \cdot \frac{6}{n^4 (\log_2 n)^2} \cdot 2
$$

$$
\overset{a}{\leq} \frac{1}{8n^2 (\log_2 n)^4}. \tag{177}
$$

We apply the above bound of $\sum_{i=1}^{k^*}$, and use Lemma 21 to bound $\sum_{i=k^*+1}^{d}$:

$$
\left\langle \begin{bmatrix} \mathbf{O} & \mathbf{O} \\ \mathbf{O} & \mathbf{S} \end{bmatrix}, \tilde{\mathbf{B}}_t \right\rangle
$$

$$
= \sum_{i=1}^{k^*} \lambda_i (\mathbf{w}_i^*)^2 \left(\prod_{s=1}^{t} \mathbf{A}_{s,i} \begin{bmatrix} 1 \\ 1 \end{bmatrix}\right)_1^2 + \sum_{i=k^*+1}^{d} \lambda_i (\mathbf{w}_i^*)^2 \left(\prod_{s=1}^{t} \mathbf{A}_{s,i} \begin{bmatrix} 1 \\ 1 \end{bmatrix}\right)_1^2 \tag{178}
$$

$$
\leq \frac{36}{n^2 (\log_2 n)^4} \sum_{i=1}^{k^*} \lambda_i (\mathbf{w}_i^*)^2 + 4 \sum_{i=k^*+1}^{d} \lambda_i (\mathbf{w}_i^*)^2.
$$

This completes the proof. $\qquad\square$

*Proof of Lemma 11.* From the recursive definition of $\tilde{\mathbf{B}}_t^{(1)}$ in (57) and Lemma 25, we have:

$$
\mathbf{B}_t^{(1)} = \mathcal{B}_t \circ \mathbf{B}_{t-1}^{(1)} + \mathbb{E}\left[\hat{\mathbf{G}}_t \otimes \hat{\mathbf{G}}_t\right] \circ \tilde{\mathbf{B}}_{t-1}
$$

$$
\preceq \mathcal{B}_t \circ \mathbf{B}_{t-1}^{(1)} + 4 \|\mathbf{w}^*\|_{\mathbf{S}}^2 \cdot \begin{bmatrix} \delta_t^2 \mathbf{S} & \delta_t q_t \mathbf{S} \\ \delta_t q_t \mathbf{S} & q_t^2 \mathbf{S} \end{bmatrix}. \tag{179}
$$

This form is identical to the recursion of $\tilde{\mathbf{C}}_t$ if we replace $4 \|\mathbf{w}^*\|_{\mathbf{S}}^2$ by $\sigma^2$. Therefore, we apply Lemma 4 to obtain

$$
\left\langle \tilde{\mathbf{T}}, \mathbf{B}_n^{(1)} \right\rangle \leq \|\mathbf{w}_0 - \mathbf{w}^*\|_{\mathbf{S}}^2 \cdot \left[\sum_{i=1}^{k^*} \frac{2t_{ii}}{K\lambda_i} + \frac{512}{15} K \left(\frac{q-c\delta}{1-c}\right)^2 \sum_{i=k^*+1}^{d} \lambda_i t_{ii}\right]. \tag{180}
$$

$\qquad\square$

## A.9 AUXILIARY LEMMAS

**Lemma 26.** *For $k \geq 0$ and $0 \leq x \leq 1$, we have*

$$
x^k [1 + k(1-x)] \leq 1. \tag{181}
$$

*Proof.* Let $f(x) = x^k [1 + k(1-x)]$ and its derivative $f'(x) = k(k+1)x^{k-1}(1-x) \geq 0$. Thus, $f(x) \leq f(1) = 1$. $\qquad\square$

**Lemma 27.** *For $k \in \mathbb{N}$ and $\sin\theta \neq 0$, we have*

$$\left|\frac{\sin k\theta}{\sin\theta}\right| \leq k. \tag{182}$$

*Proof.* By induction, for $k = 0$, the conclusion is trivial. Assume

$$\left|\frac{\sin(k-1)\theta}{\sin\theta}\right| \leq k - 1. \tag{183}$$

Then we have

$$\left|\frac{\sin k\theta}{\sin\theta}\right| = \left|\frac{\sin(k-1)\theta\cos\theta + \cos(k-1)\theta\sin\theta}{\sin\theta}\right|$$
$$\leq |\cos\theta|\left|\frac{\sin(k-1)\theta}{\sin\theta}\right| + |\cos(k-1)\theta| \leq k. \tag{184}$$
$$\square$$

# B  PROOFS OF OPTIMALITY ANALYSIS IN SECTION 5

This section provides the proofs of Section 5.

## B.1  PROOF OF THEOREM 8

*Proof of Theorem 8.* By the lower bound in Theorem 5, we have

$$R(k_i) \geq \sup_{\mathbf{F}\succeq\mathbf{O},\ \|\mathbf{F}\|_*\leq 1/\pi^2} \left\langle \mathbf{T}', \left(\mathbf{F}^{-1} + n\mathbf{S}'\right)^{-1}\right\rangle. \tag{185}$$

Therefore, we only have to show that

$$R(k_i) \lesssim \sup_{\mathbf{F}\succeq\mathbf{O},\ \|\mathbf{F}\|_*\leq 1/\pi^2} \left\langle \mathbf{T}', \left(\mathbf{F}^{-1} + n\mathbf{S}'\right)^{-1}\right\rangle. \tag{186}$$

Recall that

$$R(k_i) \asymp \max_{\substack{\mathsf{A}\subseteq\{1,\ldots,k_1\}:\\ \sum_{i\in\mathsf{A}}\frac{1}{n\lambda_i}\leq 1}} \left\langle \mathbf{T}', \left(\mathbf{S}'_\mathsf{A}\right)^{-1}\right\rangle + \underbrace{\sup_{\substack{\mathbf{F}\succeq\mathbf{O},\\ \|\mathbf{F}\|_*\leq 1}} \left\langle \mathbf{T}', \left(\mathbf{F}^{-1} + n\mathbf{S}'_{k_1+1:d}\right)^{-1}\right\rangle}_{(a)}.$$

Let

$$\mathbf{F}_1 = \mathrm{diag}\left\{\frac{\mathbb{1}_{i\in\mathsf{A}}}{n\lambda_i}\right\}_{i=1}^{k_1} \in \mathbb{R}^{k_1\times k_1}, \quad \mathbf{F}_2 = \operatorname*{arg\,min}_{\substack{\mathbf{F}\succeq\mathbf{O},\\ \|\mathbf{F}\|_*\leq 1}}(a), \tag{187}$$

and $\mathbf{F}_0 = \mathrm{diag}\{\mathbf{F}_1, \mathbf{F}_2\}/(2\pi^2)$. Since $\|\mathbf{F}\|_* \leq 1/\pi^2, \mathbf{F} \succeq \mathbf{O}$, we have

$$R(k_i) \asymp \left\langle \mathbf{T}', \left(\mathbf{F}_0^{-1} + n\mathbf{S}'\right)^{-1}\right\rangle \leq \sup_{\mathbf{F}\succeq\mathbf{O},\ \|\mathbf{F}\|_*\leq 1/\pi^2} \left\langle \mathbf{T}', \left(\mathbf{F}^{-1} + n\mathbf{S}'\right)^{-1}\right\rangle. \tag{188}$$

This completes the proof. $\square$

## B.2  PROOF OF COROLLARY 9

*Proof of Corollary 9.* We choose $\delta = \gamma = \frac{1}{2188\,\mathrm{tr}\,\mathbf{S}\ln n}$. From the lower bound of $n$, we have $k^* = d$ by Theorem 13. Thus,

$$\mathbb{E}_{\tilde{P}^{\otimes n}}\left\|\mathbf{w}_n^{\mathrm{SGD}} - \mathbf{w}^*\right\|_{\mathbf{T}}^2 \lesssim \sum_{i=1}^{k^*}\frac{t_{ii}}{K\lambda_i} + \max_{\mathbf{w}\in S(\mathbf{w}_0-\mathbf{w}^*)}\frac{\|\mathbf{w}\|_{\mathbf{T}}^2}{n^2(\log_2 n)^4},$$

$$\leq \frac{\ln n}{n}\mathrm{tr}\left(\mathbf{T}\mathbf{S}^{-1}\right) + \max_i\frac{\|\mathbf{w}\|_{\mathbf{U}_i\mathbf{T}\mathbf{U}_i}^2}{n^2(\log_2 n)^4},$$

$$\overset{a}{\leq} \frac{\ln n}{n}\mathrm{tr}\left(\mathbf{T}\mathbf{S}^{-1}\right) + \max_i\frac{\|\mathbf{M}^{-1/2}\mathbf{U}_i\mathbf{T}\mathbf{U}_i\mathbf{M}^{-1/2}\|}{n^2(\log_2 n)^4},$$

where $\mathbf{U}_i = \text{diag}\{\pm 1, \pm 1, \ldots, \pm 1\}$, $1 \leq i \leq 2^d$. $\overset{a}{\leq}$ follows from $\|\mathbf{w}^*\|_{\mathbf{M}} \leq 1$. Therefore, we have

$$\mathbb{E}_{\tilde{P}^{\otimes n}} \left\| \mathbf{w}_n^{\text{SGD}} - \mathbf{w}^* \right\|_{\mathbf{T}}^2 \lesssim \frac{\ln n}{n} \text{tr}\left(\mathbf{T}\mathbf{S}^{-1}\right) + \max_i \frac{\|\mathbf{M}^{-1/2}\mathbf{U}_i\mathbf{T}\mathbf{U}_i\mathbf{M}^{-1/2}\|}{n^2 (\log_2 n)^4} \tag{189}$$
$$= \tilde{\mathcal{O}}\left(\text{tr}(\mathbf{T}\mathbf{S}^{-1})/n\right)$$

$\square$

### B.3 Proof of Corollary 10

The proof of the corollary 10 is divided into two parts. We first show a different lower and upper bound (up to logarithmic factors), namely

$$\frac{\sigma^2 B d_1}{n}. \tag{190}$$

Then, we show that

$$\frac{\sigma^2 B d_1}{n} \asymp \inf_{\delta > 0} \left\{ \delta^2 + \frac{\sigma^2 B d(\delta)}{n} \right\}. \tag{191}$$

**Lemma 28** (Upper bound). *Under the conditions in Corollary 10, we have*

$$\sup_{\tilde{P}} \mathbb{E}_{\tilde{P}^{\otimes n}} \left\| \mathbf{w}_n^{\text{SGD}} - \mathbf{w}^* \right\|_{\mathbf{T}}^2 \leq \tilde{\mathcal{O}}\left( \frac{\sigma^2 B d_1}{n} \right). \tag{192}$$

*Proof.* From Theorem 6, we have

$$\sup_{\tilde{P}} \mathbb{E}_{\tilde{P}^{\otimes n}} \left\| \mathbf{w}_n^{\text{SGD}} - \mathbf{w}^* \right\|_{\mathbf{T}}^2$$

$$\lesssim \sigma^2 \left[ \frac{\ln^2 n}{n} \left\langle \mathbf{T}_{0:k_B^*}, \mathbf{S}_{0:k_B^*}^{-1} \right\rangle + n(\gamma + \delta)^2 \left\langle \mathbf{T}_{k_B^*:\infty}, \mathbf{S}_{k_B^*:\infty} \right\rangle \right] + \left\| \mathbf{T}_{k_B^*:\infty} \right\|$$

$$\leq 2\sigma^2 \left[ \frac{\ln^2 n}{n} \left\langle \mathbf{T}_{0:k_B^*}, \left(\mathbf{S}_{0:k_B^*} + \lambda_{k_B^*}\mathbf{I}\right)^{-1} \right\rangle + n(\gamma + \delta)^2 \left\langle \mathbf{T}_{k_B^*:\infty}, \lambda_{k_B^*}^2 \left(\mathbf{S}_{k_B^*:\infty} + \lambda_{k_B^*}\mathbf{I}\right)^{-1} \right\rangle \right] + \mu_{d_1+1}$$

$$\overset{a}{\leq} \tilde{\mathcal{O}}\left( \frac{\sigma^2}{n} \left\langle \mathbf{T}, \left(\mathbf{S} + \lambda_{k_B^*}\mathbf{I}\right)^{-1} \right\rangle + \mu_{d_1+1} \right)$$

$$\leq \tilde{\mathcal{O}}\left( \frac{\sigma^2}{n} \left\langle \mathbf{T}, \left(\mathbf{T}/B + \mu_{d_1}\mathbf{I}/B\right)^{-1} \right\rangle + \mu_{d_1+1} \right)$$

$$= \tilde{\mathcal{O}}\left( \frac{\sigma^2 B}{n} \sum_{i=1}^d \frac{\mu_i}{\mu_i + \mu_{d_1}} + \mu_{d_1+1} \right), \tag{193}$$

where $\overset{a}{\leq}$ uses $\lambda_{k_B^*} = \frac{32 \ln n \log_2 n}{n(\gamma + \delta)}$. From the eigenvalue regularity condition, we have

$$\sum_{i=1}^d \frac{\mu_i}{\mu_i + \mu_{d_1}} = \sum_{i=1}^{d_1} \frac{\mu_i}{\mu_i + \mu_{d_1}} + \sum_{i=d_1+1}^d \frac{\mu_i}{\mu_i + \mu_{d_1}} \tag{194}$$

$$\leq \frac{d_1}{2} + \frac{C d_1 \mu_{d_1}}{2 \mu_{d_1}} \leq \mathcal{O}(d_1).$$

Combining the above results and $\mu_{d_1+1} \leq \frac{\sigma^2 B d_1}{n}$ yields the desired result. $\square$

**Lemma 29** (Lower bound). *Under the conditions in Corollary 10, for $\mathbf{T} = B\mathbf{S}$, we have the following lower bound:*

$$\inf_{\hat{\mathbf{w}}} \sup_{\tilde{P} \in \mathcal{P}(W_{\mathbf{M}}, \mathbf{S}, \mathbf{T})} \mathbb{E}_{\tilde{P}^{\otimes n} \times P_\xi} \|\hat{\mathbf{w}} - \mathbf{w}^*\|_{\mathbf{T}}^2 \geq \Omega\left( \frac{\sigma^2 B d_1}{n} \right). \tag{195}$$

*Proof.* From Theorem 5, we have

$$\inf_{\hat{\mathbf{w}}} \sup_{\tilde{P} \in \mathcal{P}(W_{\mathbf{M}}, \mathbf{S}, \mathbf{T})} \mathbb{E}_{\tilde{P}^{\otimes n} \times P_\xi} \|\hat{\mathbf{w}} - \mathbf{w}^*\|_{\mathbf{T}}^2 \geq \sup_{\mathbf{F} \succeq \mathbf{O}, \, \|\mathbf{F}\|_* \leq 1/\pi^2} \left\langle \mathbf{T}, \left(\mathbf{F}^{-1} + n\mathbf{S}/\sigma^2\right)^{-1} \right\rangle. \quad (196)$$

Let

$$\mathbf{F}_{d_1 d_1} = \mathrm{diag}\left\{\frac{\sigma^2 B}{\pi^2 n \mu_1}, \ldots, \frac{\sigma^2 B}{\pi^2 n \mu_{d_1}}, 0, \ldots, 0\right\}, \quad (197)$$

so $\mathrm{tr}\, \mathbf{F} \leq 1/\pi^2$, and we have

$$\inf_{\hat{\mathbf{w}}} \sup_{\tilde{P} \in \mathcal{P}(W_{\mathbf{M}}, \mathbf{S}, \mathbf{T})} \mathbb{E}_{\tilde{P}^{\otimes n} \times P_\xi} \|\hat{\mathbf{w}} - \mathbf{w}^*\|_{\mathbf{T}}^2 \geq \left\langle \mathbf{T}, \left(\mathbf{F}^{-1} + n\mathbf{T}/(\sigma^2 B)\right)^{-1} \right\rangle$$

$$\geq \sum_{i=1}^{d_1} \mu_i \cdot \left(\frac{\pi^2 n \mu_i}{\sigma^2 B} + \frac{n \mu_i}{\sigma^2 B}\right)^{-1} \quad (198)$$

$$= \frac{\sigma^2 B d_1}{(1 + \pi^2) n}.$$

$\square$

By Lemma 39, we have

$$\sup_{\substack{\mathbf{F} \succeq \mathbf{O} \\ \|\mathbf{F}\|_* \leq 1/\pi^2}} \left\langle \mathbf{T}', \left(\mathbf{F}^{-1} + \frac{n\mathbf{S}'}{\sigma^2}\right)^{-1} \right\rangle = \min_{\mathbf{A} \in \mathbb{R}^{d \times d}} \frac{1}{\pi^2} \left\|(\mathbf{I} - \mathbf{A})^\top \mathbf{T}'(\mathbf{I} - \mathbf{A})\right\| + \frac{\sigma^2}{n} \left\langle \mathbf{T}', \mathbf{A}\left(\mathbf{S}'\right)^{-1} \mathbf{A}^\top \right\rangle$$

The following lemma provides an explicit form of the lower bound when $\mathbf{S}$, $\mathbf{T}$ and $\mathbf{M}$ commute.

**Lemma 30.** *Let* $\mathbf{T} = \mathrm{diag}\,\{t_i\}_{i=1}^d$ *and* $\mathbf{M} = \mathrm{diag}\,\{m_i\}_{i=1}^d$, *we have*

$$\min_{\mathbf{A} \in \mathbb{R}^{d \times d}} \frac{1}{\pi^2} \left\|(\mathbf{I} - \mathbf{A})^\top \mathbf{T}'(\mathbf{I} - \mathbf{A})\right\| + \frac{\sigma^2}{n} \left\langle \mathbf{T}', \mathbf{A}\left(\mathbf{S}'\right)^{-1} \mathbf{A}^\top \right\rangle \asymp \min_{\tau \geq 0} \frac{\tau^2}{\pi^2} + \sum_{i \in \mathbb{K}_\tau} \frac{\sigma^2 t_i}{n \lambda_i}, \quad (199)$$

*where* $\mathbb{K}_\tau = \left\{k : t_k/m_k > \tau^2\right\}$.

*Proof.* A key observation is that when $\mathbf{T}'$ is diagonal, the minimum of the LHS of (199) is attained when $\mathbf{A}$ is diagonal. Note that the LHS of (199) is a convex optimization. Let $\mathbf{A}_0$ denote a minimizer. Consider $2^d$ reflection matrices $\mathbf{U}_i = \mathrm{diag}\,\{\pm 1, \pm 1, \ldots, \pm 1\}$, then for all $i \in \left[2^d\right]$, $\mathbf{U}_i \mathbf{A}_0 \mathbf{U}_i$ is also a minimizer. From the convexity, we have that

$$\mathbf{A}^* = \frac{1}{2^d} \sum_{i=1}^{2^d} \mathbf{U}_i \mathbf{A}_0 \mathbf{U}_i \quad (200)$$

is also a minimizer, and $\mathbf{A}^*$ is diagonal. Thus, we can restrict $\mathbf{A}$ to be diagonal when minimizing the LHS of (199). Therefore, let $\mathbf{A} = \mathrm{diag}\,\{a_i\}_{i=1}^d$ and note that $\mathbf{T}' = \mathrm{diag}\,\{t_i/m_i\}_{i=1}^d$, then the LHS of (199) is equivalent to

$$\min_{a_i} \max_{k \in [d]} \frac{(1 - a_k)^2 t_k}{\pi^2 m_k} + \sum_{i=1}^d \frac{\sigma^2 a_i^2 t_i}{n \lambda_i}. \quad (201)$$

We can write out the following equivalent form:

$$\min_{a_i, \tau \geq 0} \frac{\tau^2}{\pi^2} + \sum_{i=1}^d \frac{\sigma^2 a_i^2 t_i}{n \lambda_i},$$

$$\text{s.t. } \forall i \in [d], \frac{(1 - a_i)^2 t_i}{m_i} \leq \tau^2. \quad (202)$$

We first minimize the above program with respect to $a_i$ to get

$$a_i = \begin{cases} 0, & t_i/m_i < \tau^2; \\ 1 - \tau\sqrt{m_i/t_i}, & t_i/m_i \geq \tau^2. \end{cases} \quad (203)$$

Plugging the value of $a_i$ into left hand side of (199), we obtain the first equality in (199):

$$\min_{\tau \geq 0} \frac{\tau^2}{\pi^2} + \sum_{i \in \mathbb{K}_\tau} \left(1 - \tau \sqrt{\frac{m_i}{t_i}}\right)^2 \frac{\sigma^2 t_i}{n \lambda_i}. \tag{204}$$

Let $\tau^* \geq 0$ denote the minimizer of (204), we have

$$\left(\frac{\tau^*}{\pi}\right)^2 + \sum_{i \in \mathbb{K}_{\tau^*}} \left(1 - \tau^* \sqrt{\frac{m_i}{t_i}}\right)^2 \frac{\sigma^2 t_i}{n \lambda_i} \overset{a}{\geq} \left(\frac{\tau^*}{\pi}\right)^2 + \sum_{i \in \mathbb{K}_{2\tau^*}} \left(1 - \tau^* \sqrt{\frac{m_i}{t_i}}\right)^2 \frac{\sigma^2 t_i}{n \lambda_i}$$

$$\overset{b}{\geq} \frac{1}{4} \left(\frac{2\tau^*}{\pi}\right)^2 + \sum_{i \in \mathbb{K}_{2\tau^*}} \frac{\sigma^2 t_i}{4 n \lambda_i} \overset{c}{\geq} \frac{1}{4} \min_{\tau \geq 0} \frac{\tau^2}{\pi^2} + \sum_{i \in \mathbb{K}_\tau} \frac{\sigma^2 t_i}{n \lambda_i}, \tag{205}$$

where $\overset{a}{\geq}$ is from $\mathbb{K}_{2\tau^*} \subset \mathbb{K}_{\tau^*}$, $\overset{b}{\geq}$ uses that $1 - \tau^* \sqrt{\frac{m_i}{t_i}} \geq \frac{1}{2}$ for all $i \in \mathbb{K}_{2\tau^*}$, and $\overset{c}{\geq}$ replaces $2\tau^*$ by $\tau$ and minimizes with respect to $\tau$. This completes the proof of the inequality in (199).

Let $\mathbf{A} = \mathrm{diag}\{\mathbb{1}_{i \in \mathbb{K}_\tau}\}_{i=1}^d$, we have

$$\frac{\tau^2}{\pi^2} + \sum_{i \in \mathbb{K}_\tau} \frac{\sigma^2 t_i}{n \lambda_i} = \frac{1}{\pi^2} \left\| (\mathbf{I} - \mathbf{A})^\top \mathbf{T}' (\mathbf{I} - \mathbf{A}) \right\| + \frac{\sigma^2}{n} \left\langle \mathbf{T}', \mathbf{A} (\mathbf{S}')^{-1} \mathbf{A}^\top \right\rangle. \tag{206}$$

Minimizing two sides yields

$$\min_{\tau \geq 0} \frac{\tau^2}{\pi^2} + \sum_{i \in \mathbb{K}_\tau} \frac{\sigma^2 t_i}{n \lambda_i} \geq \min_{\mathbf{A} \in \mathbb{R}^{d \times d}} \frac{1}{\pi^2} \left\| (\mathbf{I} - \mathbf{A})^\top \mathbf{T}' (\mathbf{I} - \mathbf{A}) \right\| + \frac{\sigma^2}{n} \left\langle \mathbf{T}', \mathbf{A} (\mathbf{S}')^{-1} \mathbf{A}^\top \right\rangle \tag{207}$$

$\square$

**Lemma 31.** *We have*

$$\sup_{\mathbf{F} \succeq \mathbf{O}, \; \|\mathbf{F}\|_* \leq 1/\pi^2} \left\langle \mathbf{T}, \left(\mathbf{F}^{-1} + n\mathbf{T}/(\sigma^2 B)\right)^{-1} \right\rangle \asymp \inf_{\delta > 0} \left\{ \delta^2 + \frac{\sigma^2 B d(\delta)}{n} \right\}. \tag{208}$$

*Proof.* Apply Lemma 30, we have

$$\sup_{\mathbf{F} \succeq \mathbf{O}, \; \|\mathbf{F}\|_* \leq 1/\pi^2} \left\langle \mathbf{T}, \left(\mathbf{F}^{-1} + n\mathbf{T}/(\sigma^2 B)\right)^{-1} \right\rangle \asymp \min_{\tau \geq 0} \frac{\tau^2}{\pi^2} + \frac{\sigma^2 B |\mathbb{K}_\tau|}{n}$$

$$\asymp \inf_{d'} \mu_{d'} + \frac{\sigma^2 B d'}{n} \tag{209}$$

$$= \inf_{\delta > 0} \left\{ \delta^2 + \frac{\sigma^2 B d(\delta)}{n} \right\}.$$

where $\mathbb{K}_\tau = \left\{ k : t_k/m_k > \tau^2 \right\}$. $\square$

*Proof of Corollary 10.* From Lemmas 28 and 29, we have

$$\sup_{\mathbf{F} \succeq \mathbf{O}, \; \|\mathbf{F}\|_* \leq 1/\pi^2} \left\langle \mathbf{T}, \left(\mathbf{F}^{-1} + n\mathbf{T}/(\sigma^2 B)\right)^{-1} \right\rangle \asymp \tilde{\Theta} \left( \frac{\sigma^2 B d_1}{n} \right). \tag{210}$$

Then, by Lemma 31, we know SGD achieves optimal rate $\tilde{\Theta} \left( \inf_{\delta > 0} \left\{ \delta^2 + \frac{\sigma^2 B d(\delta)}{n} \right\} \right)$. $\square$

### B.4 PROOF OF COROLLARY 14

We begin by showing that if $D_{\mathrm{KL}}(Q_\mathbf{x} \| P_\mathbf{x}^{\mathrm{KL}}) \leq C$, we have $\mathbf{T} \preceq B \cdot \mathbf{S}$, where $B$ only depends on $C$.

**Lemma 32.** *Suppose $P_\mathbf{x}^{\mathrm{KL}})$ and $Q_\mathbf{x}$ are Gaussian distributions, and $D_{\mathrm{KL}}(Q_\mathbf{x} \| P_\mathbf{x}^{\mathrm{KL}}) \leq C$, then we have $\mathbf{T} \preceq B \cdot \mathbf{S}$, where $B$ only depends on $C$.*

. Let $\mathbf{S}$ and $\mathbf{T}$ denote the source and target covariance matrix. Since

$$D_{\mathrm{KL}}(Q_{\mathbf{x}} \| P_{\mathbf{x}}^{\mathrm{KL}}) = \frac{1}{2} \left( \mathrm{tr}(\mathbf{S}^{-1/2}\mathbf{T}\mathbf{S}^{-1/2}) - d - \ln \det(\mathbf{S}^{-1/2}\mathbf{T}\mathbf{S}^{-1/2}) \right) \tag{211}$$

Let $\rho_i$ denote the eigenvalues of $\mathbf{S}^{-1/2}\mathbf{T}\mathbf{S}^{-1/2}$, we have

$$D_{\mathrm{KL}}(Q_{\mathbf{x}} \| P_{\mathbf{x}}^{\mathrm{KL}}) = \frac{1}{2} \sum_{i=1}^{d} \rho_i - 1 - \ln \rho_i < C. \tag{212}$$

Since $x - 1 - \ln x \geq 0$ for any $x > 0$, we have $\rho_i - 1 - \ln \rho_i < \epsilon$ for all $i \in [d]$. By solving the inequality, we obtain that $\rho_i$ are bounded by a constant $B$ depending on $C$. $\qquad\square$

*Proof of Corollary 14.* The proof parallels the proof of Corollary 10. Similar to Lemma 28, we have upper bound $\tilde{\mathcal{O}} \left( \inf_{\delta > 0} \left\{ \delta^2 + \frac{\sigma^2 B d(\delta)}{n} \right\} \right)$. For the lower bound, note that $Q_{\mathbf{x}} = P_{\mathbf{x}}^{\mathrm{KL}}$ implies $\mathbf{T} = \mathbf{S}$. Therefore, similar to Lemma 29, we have lower bound $\Omega \left( \inf_{\delta > 0} \left\{ \delta^2 + \frac{\sigma^2 d(\delta)}{n} \right\} \right)$. Ignore the constant $B$, we get the matching bound $\tilde{\Theta} \left( \inf_{\delta > 0} \left\{ \delta^2 + \frac{\sigma^2 d(\delta)}{n} \right\} \right)$ $\qquad\square$

### B.5 PROOF OF COROLLARY 12

We first show the upper bound in the Corollary 12.

**Lemma 33.** *Under the conditions in Corollary 12, for the region $1 > s > \frac{a}{2a-1}$, we set*

$$\tilde{\kappa} = \Theta\left(n^{\frac{(1-s)a}{(a-1)sa}}\right), \quad \delta = \Theta(1/\ln n), \quad \gamma = \Theta\left(n^{\frac{1-s}{s}} / \ln n\right). \tag{213}$$

*Then we have*

$$\mathbb{E}_{\tilde{P}^{\otimes n}} \left\| \mathbf{w}_n^{\mathrm{SGD}} - \mathbf{w}^* \right\|_{\mathbf{T}}^2 \leq \begin{cases} \tilde{\mathcal{O}}(1/n), & r \geq 1/a; \\ \tilde{\mathcal{O}}\left((1/n)^{\frac{(r+s)a-1}{sa}}\right), & r < 1/a. \end{cases} \tag{214}$$

*Proof.* Since we have

$$\frac{\tilde{\kappa}}{n \sum_{i > \tilde{\kappa}} \lambda_i} = \Theta\left(n^{\frac{(1-2a)s+a}{(a-1)s}}\right), \tag{215}$$

and $s > \frac{a}{2a-1}$, the parameter choice is feasible. From Theorem 6, we have $k^* = \tilde{\Theta}\left(n^{\frac{1}{sa}}\right)$, and

$$\begin{aligned}
\sup_{\tilde{P}} \mathbb{E}_{\tilde{P}^{\otimes n}} \left\| \mathbf{w}_n^{\mathrm{SGD}} - \mathbf{w}^* \right\|_{\mathbf{T}}^2 &\leq \tilde{\mathcal{O}} \left( \sum_{i=1}^{k^*} \frac{i^{-ra}}{n} + n(\gamma + \delta)^2 \sum_{i=k^*+1}^{d} i^{-(2+r)a} + (k^*)^{-(r+s)a+1} \right) \\
&\leq \begin{cases} \tilde{\mathcal{O}}(1/n), & r \geq 1/a; \\ \tilde{\mathcal{O}}\left((1/n)^{\frac{(r+s)a-1}{sa}}\right), & r < 1/a. \end{cases}
\end{aligned} \tag{216}$$

$\qquad\square$

**Lemma 34.** *Under the conditions in Corollary 12, for the region $s \geq 1$, we set*

$$\tilde{\kappa} = \Theta\left(1\right), \quad \delta = \gamma = \Theta\left(n^{\frac{1-s}{s}} / \ln n\right). \tag{217}$$

*Then we have*

$$\mathbb{E}_{\tilde{P}^{\otimes n}} \left\| \mathbf{w}_n^{\mathrm{SGD}} - \mathbf{w}^* \right\|_{\mathbf{T}}^2 \leq \begin{cases} \tilde{\mathcal{O}}(1/n), & r \geq 1/a; \\ \tilde{\mathcal{O}}\left((1/n)^{\frac{(r+s)a-1}{sa}}\right), & r < 1/a. \end{cases} \tag{218}$$

*Proof.* Since we have

$$\frac{\tilde{\kappa}}{n \sum_{i > \tilde{\kappa}} \lambda_i} = \Theta\left(1/n\right), \tag{219}$$

the parameter choice is feasible. From Theorem 6, we have $k^* = \tilde{\Theta}\left(n^{\frac{1}{sa}}\right)$, and

$$
\sup_{\tilde{P}} \mathbb{E}_{\tilde{P}^{\otimes n}} \left\| \mathbf{w}_n^{\mathrm{SGD}} - \mathbf{w}^* \right\|_{\mathbf{T}}^2 \leq \tilde{\mathcal{O}}\left( \sum_{i=1}^{k^*} \frac{i^{-ra}}{n} + n(\gamma+\delta)^2 \sum_{i=k^*+1}^{d} i^{-(2+r)a} + (k^*)^{-(1+r)a+1} \right)
$$
$$
\leq \begin{cases} \tilde{\mathcal{O}}(1/n), & r \geq 1/a; \\ \tilde{\mathcal{O}}\left((1/n)^{\frac{(r+s)a-1}{sa}}\right), & r < 1/a. \end{cases} \tag{220}
$$

$\square$

The lower bound follows from Lemma 39, as shown in the following lemma.

**Lemma 35.** *Under the conditions in Corollary 12, we have the following lower bound*

$$
\begin{cases} \tilde{\Omega}(1/n), & r \geq 1/a; \\ \tilde{\Omega}\left((1/n)^{\frac{(r+s)a-1}{sa}}\right), & r < 1/a. \end{cases} \tag{221}
$$

*Proof.* By Lemma 39, we have the following lower bound:

$$
\min_{\mathbf{A}\in\mathbb{R}^{d\times d}} \frac{1}{\pi^2} \left\| (\mathbf{I}-\mathbf{A})^\top \mathbf{M}^{-1/2}\mathbf{w}\mathbf{w}^\top \mathbf{M}^{-1/2}(\mathbf{I}-\mathbf{A}) \right\| + \frac{\sigma^2}{n} \left\langle \mathbf{M}^{-1/2}\mathbf{w}\mathbf{w}^\top \mathbf{M}^{-1/2}, \mathbf{A}\left(\mathbf{S}'\right)^{-1}\mathbf{A}^\top \right\rangle \tag{222}
$$

Let $\mathbf{u} = (\mathbf{I}-\mathbf{A})^\top \mathbf{M}^{-1/2}\mathbf{w}$,

$$
\min_{\mathbf{u}\in\mathbb{R}^d} \frac{1}{\pi^2}\|\mathbf{u}\|^2 + \frac{\sigma^2}{n} \left\| (\mathbf{M}^{-1/2}\mathbf{w}-\mathbf{u}) \right\|_{(\mathbf{S}')^{-1}}^2. \tag{223}
$$

Solving the optimization problem, we get the lower bound:

$$
\left\| \frac{\sigma^2}{n}(\mathbf{S}')^{-1}\mathbf{M}^{-1/2}\mathbf{w} \right\|_{\left(\frac{\sigma^2}{n}(\mathbf{S}')^{-1}+\frac{\mathbf{I}}{\pi^2}\right)^{-1}}^2 + \frac{\sigma^2}{n}\left\| \mathbf{M}^{-1/2}\mathbf{w} \right\|_{(\mathbf{S}')^{-1}}^2
$$
$$
\approx \sum_{i=1}^{d} \frac{\left(\frac{\sigma^2}{n}i^{sa}\cdot i^{(1-s)a/2}\cdot i^{-(1+r)a/2}\right)^2}{\frac{\sigma^2}{n}i^{sa}+\frac{1}{\pi^2}} + \frac{\sigma^2}{n}\sum_{i=1}^{d} i^{-ra}
$$
$$
\approx \frac{1}{n^2}\sum_{i=1}^{n^{\frac{1}{sa}}} i^{-(r-s)a} + \frac{1}{n}\sum_{i=n^{\frac{1}{sa}}+1}^{d} i^{-ra} + \frac{1}{n}\sum_{i=1}^{d} i^{-ra} \tag{224}
$$
$$
\geq \begin{cases} \tilde{\Omega}(1/n), & r \geq 1/a; \\ \tilde{\Omega}\left((1/n)^{\frac{(r+s)a-1}{sa}}\right), & r < 1/a. \end{cases}
$$

$\square$

*Proof of Corollary 12.* Combine Lemmas 33, 34 and 35 to complete the proof. $\square$

## C PROOFS OF MINIMAX OPTIMALITY IN SECTION 3.3

For completeness, we present the proofs of theorems in Section 3.3. The proofs use a different prior distribution compared to Pathak et al. (2024), which does not require explicit truncation.

### C.1 PROOF OF THEOREM 5

This section provides the proof of the lower bound. For any $\mathbf{w} \in W$, we construct the probability distribution $P_{\mathbf{w}}$ of $(\mathbf{x}, y)$ such that

$$
\mathbf{x} \sim \mathcal{N}(\mathbf{0}, \mathbf{S}), \quad y = \mathbf{x}^\top \mathbf{w} + \epsilon, \tag{225}
$$

where $\epsilon \sim \mathcal{N}(\mathbf{0}, \sigma^2)$ and $\epsilon$ and $\mathbf{x}$ are independent. $P_{\mathbf{w}}$ satisfies Assumptions 4 and 1. Let $\mathcal{G}(W, \mathbf{S}, \mathbf{T}) = \{P_{\mathbf{w}} : \mathbf{w} \in W\}$ denotes the Gaussian problem class, then we have $\mathcal{G}(W, \mathbf{S}, \mathbf{T}) \subseteq \mathcal{P}(W, \mathbf{S}, \mathbf{T})$.

The first step is to reduce the minimax risk to Bayesian risk and show that the randomness of the estimator $\hat{\mathbf{w}}$ does not help to achieve better performance. We denote an estimator which only depends on samples $\{(\mathbf{x}_i, y_i)\}_{i=1}^n$ as $\hat{\mathbf{w}}^{\det}$. We have the following lemma.

**Lemma 36.** *Suppose $\pi$ is any probability distribution supported on $W$, We have*

$$\inf_{\hat{\mathbf{w}}} \sup_{\tilde{P} \in \mathcal{P}(W, \mathbf{S}, \mathbf{T})} \mathbb{E}_{\tilde{P}^{\otimes n} \times P_\xi} \|\hat{\mathbf{w}} - \mathbf{w}^*\|_{\mathbf{T}}^2 \geq \inf_{\hat{\mathbf{w}}^{\det}} \mathbb{E}_{\mathbf{w}^* \sim \pi} \mathbb{E}_{P_{\mathbf{w}^*}^{\otimes n}} \|\hat{\mathbf{w}} - \mathbf{w}^*\|_{\mathbf{T}}^2. \tag{226}$$

*Proof.* From Yao's minimax principle (Yao, 1977), we have

$$\begin{aligned}
\inf_{\hat{\mathbf{w}}} \sup_{\tilde{P} \in \mathcal{P}(W, \mathbf{S}, \mathbf{T})} \mathbb{E}_{\tilde{P}^{\otimes n} \times P_\xi} \|\hat{\mathbf{w}} - \mathbf{w}^*\|_{\mathbf{T}}^2 &\geq \inf_{\hat{\mathbf{w}}} \sup_{P_{\mathbf{w}^*} \in \mathcal{G}(W, \mathbf{S}, \mathbf{T})} \mathbb{E}_{P_{\mathbf{w}^*}^{\otimes n} \times P_\xi} \|\hat{\mathbf{w}} - \mathbf{w}^*\|_{\mathbf{T}}^2 \\
&\geq \inf_{\hat{\mathbf{w}}} \mathbb{E}_{\mathbf{w}^* \sim \pi} \mathbb{E}_{P_{\mathbf{w}^*}^{\otimes n} \times P_\xi} \|\hat{\mathbf{w}} - \mathbf{w}^*\|_{\mathbf{T}}^2 \\
&\geq \inf_{\xi} \inf_{\hat{\mathbf{w}}} \mathbb{E}_{\mathbf{w}^* \sim \pi} \mathbb{E}_{P_{\mathbf{w}^*}^{\otimes n}} \|\hat{\mathbf{w}}(\cdot, \xi) - \mathbf{w}^*\|_{\mathbf{T}}^2 \\
&\geq \inf_{\hat{\mathbf{w}}^{\det}} \mathbb{E}_{\mathbf{w}^* \sim \pi} \mathbb{E}_{P_{\mathbf{w}^*}^{\otimes n}} \left\|\hat{\mathbf{w}}^{\det} - \mathbf{w}^*\right\|_{\mathbf{T}}^2.
\end{aligned} \tag{227}$$

$\square$

We prove a multivariate generalization of Bayesian Cramer-Rao inequality.

**Lemma 37.** *We denote the density function of $P_{\mathbf{w}}^{\otimes n}$ as $f_{\mathbf{w}}$. Given data $X = \{(\mathbf{x}_i, y_i)\}_{i=1}^n \sim P_{\mathbf{w}}^{\otimes n}$, let $\hat{\mathbf{w}}^{\det} = \hat{\mathbf{w}}^{\det}(X)$ be an estimator of $\mathbf{w}$. The Fisher information matrix of $P_{\mathbf{w}}^{\otimes n}$ be defined as*

$$\mathcal{I}(\mathbf{w}) = \int_{\mathcal{X}} (\nabla_{\mathbf{w}} \ln f_{\mathbf{w}}(x)) (\nabla_{\mathbf{w}} \ln f_{\mathbf{w}}(x))^\top f_{\mathbf{w}}(x) \mathrm{d}x. \tag{228}$$

*Consider a prior probability measure $\pi$ with density function $\pi(\mathbf{w})$ that is supported on a compact set $W \subseteq \mathbb{R}^d$ and $\pi(\mathbf{w}) = 0$ on the boundary of $W$. We define the information matrix of $\pi$ as*

$$\mathcal{I}(\pi) = \int_{\mathbb{R}^d} (\nabla \ln \pi(\mathbf{w})) (\nabla \ln \pi(\mathbf{w}))^\top \pi(\mathbf{w}) \mathrm{d}\mathbf{w}. \tag{229}$$

*Then we have*

$$\mathbb{E}_{\mathbf{w} \sim \pi} \mathbb{E}_{X \sim P_{\mathbf{w}}^{\otimes n}} (\hat{\mathbf{w}} - \mathbf{w}) (\hat{\mathbf{w}} - \mathbf{w})^\top \succeq (\mathbb{E}_{\mathbf{w} \sim \pi} \mathcal{I}(\mathbf{w}) + \mathcal{I}(\pi))^{-1}. \tag{230}$$

*Proof.* We begin by defining two random variables as

$$\boldsymbol{\xi} = \hat{\mathbf{w}}^{\det}(X) - \mathbf{w}, \quad \boldsymbol{\eta} = \nabla_{\mathbf{w}} \ln (f_{\mathbf{w}}(X) \pi(\mathbf{w})). \tag{231}$$

We denote $\mathbb{E}_{\mathbf{w} \sim \pi} \mathbb{E}_{X \sim P_{\mathbf{w}}^{\otimes n}}$ by $\mathbb{E}$ for simplicity. For any vector $\mathbf{u}, \mathbf{v} \in \mathbb{R}^d$, by Cauchy-Schwarz inequality, we have

$$\mathbb{E} \left(\mathbf{u}^\top \boldsymbol{\xi} \boldsymbol{\xi}^\top \mathbf{u}\right) \mathbb{E} \left(\mathbf{v}^\top \boldsymbol{\eta} \boldsymbol{\eta}^\top \mathbf{v}\right) \geq \left[\mathbb{E} \left(\mathbf{u}^\top \boldsymbol{\xi}\right) \left(\mathbf{v}^\top \boldsymbol{\eta}\right)\right]^2. \tag{232}$$

We will show that

$$\mathbb{E} \boldsymbol{\eta} \boldsymbol{\eta}^\top = \mathbb{E} \mathcal{I}(\mathbf{w}) + \mathcal{I}(\pi), \quad \mathbb{E} \boldsymbol{\xi} \boldsymbol{\eta}^\top = \mathbf{I}. \tag{233}$$

Note that once we have established (233), we have

$$\left[\mathbf{u}^\top \mathbb{E} \left(\hat{\mathbf{w}}^{\det} - \mathbf{w}\right) \left(\hat{\mathbf{w}}^{\det} - \mathbf{w}\right)^\top \mathbf{u}\right] \left[\mathbf{v}^\top (\mathbb{E} \mathcal{I}(\boldsymbol{\theta}) + \mathcal{I}(\lambda)) \mathbf{v}\right] \geq \left(\mathbf{u}^\top \mathbf{v}\right)^2. \tag{234}$$

Let $\mathbf{v} = (\mathbb{E} \mathcal{I}(\boldsymbol{\theta}) + \mathcal{I}(\lambda))^{-1} \mathbf{u}$, we get

$$\mathbf{u}^\top \mathbb{E} \left(\hat{\mathbf{w}}^{\det} - \mathbf{w}\right) \left(\hat{\mathbf{w}}^{\det} - \mathbf{w}\right)^\top \mathbf{u} \geq \mathbf{u}^\top (\mathbb{E} \mathcal{I}(\boldsymbol{\theta}) + \mathcal{I}(\lambda))^{-1} \mathbf{u}. \tag{235}$$

Since $\mathbf{u}$ is arbitrary, we get the desired result.

Now, we prove (233) by direct calculation. Consider the $ij$-th entry of $\mathbb{E}\boldsymbol{\eta}\boldsymbol{\eta}^\top$, which is

$$
\begin{aligned}
\mathbb{E}\boldsymbol{\eta}_i\boldsymbol{\eta}_j =& \mathbb{E}\frac{\partial \ln\left(f_{\mathbf{w}}(X)\pi(\mathbf{w})\right)}{\partial \mathbf{w}_i}\frac{\partial \ln\left(f_{\mathbf{w}}(X)\pi(\mathbf{w})\right)}{\partial \mathbf{w}_j} \\
=& \mathbb{E}\left(\frac{\partial \ln f_{\mathbf{w}}(X)}{\partial \mathbf{w}_i} + \frac{\partial \ln\pi(\mathbf{w})}{\partial \mathbf{w}_i}\right)\left(\frac{\partial \ln f_{\mathbf{w}}(X)}{\partial \mathbf{w}_j} + \frac{\partial \ln\pi(\mathbf{w})}{\partial \mathbf{w}_j}\right) \\
=& \mathbb{E}\frac{\partial \ln f_{\mathbf{w}}(X)}{\partial \mathbf{w}_i}\frac{\partial \ln f_{\mathbf{w}}(X)}{\partial \mathbf{w}_j} + \mathbb{E}\frac{\partial \ln\pi(\mathbf{w})}{\partial \mathbf{w}_i}\frac{\partial \ln\pi(\mathbf{w})}{\partial \mathbf{w}_j} \\
& + \mathbb{E}\frac{\partial \ln f_{\mathbf{w}}(X)}{\partial \mathbf{w}_i}\frac{\partial \ln\pi(\mathbf{w})}{\partial \mathbf{w}_j} + \mathbb{E}\frac{\partial \ln f_{\mathbf{w}}(X)}{\partial \mathbf{w}_j}\frac{\partial \ln\pi(\mathbf{w})}{\partial \mathbf{w}_i} \\
\overset{a}{=}& \mathbb{E}\mathcal{I}_{ij}(\mathbf{w}) + \mathcal{I}_{ij}(\pi) + \mathbb{E}\frac{\partial \ln f_{\mathbf{w}}(X)}{\partial \mathbf{w}_i}\frac{\partial \ln\pi(\mathbf{w})}{\partial \mathbf{w}_j} + \mathbb{E}\frac{\partial \ln f_{\mathbf{w}}(X)}{\partial \mathbf{w}_j}\frac{\partial \ln\pi(\mathbf{w})}{\partial \mathbf{w}_i},
\end{aligned}
\tag{236}
$$

where $\overset{a}{=}$ uses the definition of $\mathcal{I}(\mathbf{w})$ and $\mathcal{I}(\pi)$. We need to show that

$$
\mathbb{E}\frac{\partial \ln f_{\mathbf{w}}(X)}{\partial \mathbf{w}_i}\frac{\partial \ln\pi(\mathbf{w})}{\partial \mathbf{w}_j} = \mathbb{E}\frac{\partial \ln f_{\mathbf{w}}(X)}{\partial \mathbf{w}_j}\frac{\partial \ln\pi(\mathbf{w})}{\partial \mathbf{w}_i} = 0.
\tag{237}
$$

For simplicity, let $\mathcal{X} = \left(\mathbb{R}^d \times \mathbb{R}\right)^n$ be the range of $X$, then we have

$$
\begin{aligned}
\mathbb{E}\frac{\partial \ln f_{\mathbf{w}}(X)}{\partial \mathbf{w}_i}\frac{\partial \ln\pi(\mathbf{w})}{\partial \mathbf{w}_j} =& \int_{\mathcal{X}\times\mathbb{R}^d} \frac{\partial \ln f_{\mathbf{w}}(x)}{\partial \mathbf{w}_i}\frac{\partial \ln\pi(\mathbf{w})}{\partial \mathbf{w}_j}f_{\mathbf{w}}(x)\pi(\mathbf{w})\mathrm{d}x\mathrm{d}\mathbf{w} \\
=& \int_{\mathcal{X}\times\mathbb{R}^d} \frac{\partial f_{\mathbf{w}}(x)}{\partial \mathbf{w}_i}\frac{\partial\pi(\mathbf{w})}{\partial \mathbf{w}_j}\mathrm{d}x\mathrm{d}\mathbf{w} \\
\overset{a}{=}& \int_{\mathbb{R}^d}\left(\frac{\partial}{\partial \mathbf{w}_i}\int_{\mathcal{X}}f_{\mathbf{w}}(x)\mathrm{d}x\right)\frac{\partial\pi(\mathbf{w})}{\partial \mathbf{w}_j}\mathrm{d}\mathbf{w} \\
\overset{b}{=}& 0,
\end{aligned}
\tag{238}
$$

where $\overset{a}{=}$ exchanges $\int_{\mathcal{X}}$ and $\frac{\partial}{\partial \mathbf{w}_i}$, and $\overset{b}{=}$ uses $\int_{\mathcal{X}}f_{\mathbf{w}}(x)\mathrm{d}x \equiv 1$ and the derivative of a constant is 0. Thus, $\mathbb{E}\boldsymbol{\eta}\boldsymbol{\eta}^\top = \mathbb{E}\mathcal{I}(\mathbf{w}) + \mathcal{I}(\pi)$.

Consider the $ij$-th entry of $\mathbb{E}\boldsymbol{\xi}\boldsymbol{\eta}^\top$, which is

$$
\begin{aligned}
\mathbb{E}\boldsymbol{\xi}_i\boldsymbol{\eta}_j =& \mathbb{E}\left(\hat{\mathbf{w}}_i^{\mathrm{det}}(X) - \mathbf{w}_i\right)\frac{\partial \ln\left(f_{\mathbf{w}}(X)\pi(\mathbf{w})\right)}{\partial \mathbf{w}_j} \\
=& \int_{\mathcal{X}\times\mathbb{R}^d}\left(\hat{\mathbf{w}}_i^{\mathrm{det}}(x) - \mathbf{w}_i\right)\frac{\partial \ln\left(f_{\mathbf{w}}(x)\pi(\mathbf{w})\right)}{\partial \mathbf{w}_j}f_{\mathbf{w}}(x)\pi(\mathbf{w})\mathrm{d}x\mathrm{d}\mathbf{w} \\
=& \int_{\mathcal{X}\times\mathbb{R}^d}\left(\hat{\mathbf{w}}_i^{\mathrm{det}}(x) - \mathbf{w}_i\right)\frac{\partial\left(f_{\mathbf{w}}(x)\pi(\mathbf{w})\right)}{\partial \mathbf{w}_j}\mathrm{d}x\mathrm{d}\mathbf{w} \\
\overset{a}{=}& \int_{\mathcal{X}\times\mathbb{R}^d}\frac{\partial\left[\left(\hat{\mathbf{w}}_i^{\mathrm{det}}(x) - \mathbf{w}_i\right)f_{\mathbf{w}}(x)\pi(\mathbf{w})\right]}{\partial \mathbf{w}_j}\mathrm{d}x\mathrm{d}\mathbf{w} \\
& - \int_{\mathcal{X}\times\mathbb{R}^d}\frac{\partial\left(\hat{\mathbf{w}}_i^{\mathrm{det}}(x) - \mathbf{w}_i\right)}{\partial \mathbf{w}_j}f_{\mathbf{w}}(x)\pi(\mathbf{w})\mathrm{d}x\mathrm{d}\mathbf{w} \\
\overset{b}{=}& \int_{\mathcal{X}\times\mathbb{R}^{d-1}}\left[\left(\hat{\mathbf{w}}_i^{\mathrm{det}}(x) - \mathbf{w}_i\right)f_{\mathbf{w}}(x)\pi(\mathbf{w})\right]\Big|_{\mathbf{w}_j=-\infty}^{\mathbf{w}_j=+\infty}\mathrm{d}x\prod_{k\neq j}\mathrm{d}\mathbf{w}_k - \mathbb{E}\frac{\partial\left(-\mathbf{w}_i\right)}{\partial \mathbf{w}_j} \\
\overset{c}{=}& \delta_{ij},
\end{aligned}
\tag{239}
$$

where $\overset{a}{=}$ uses integration by parts, $\overset{b}{=}$ integrates with respect to $\mathbf{w}_j$, and $\overset{c}{=}$ is from the fact that $W$ is compact, so $\lambda(\mathbf{w}) = 0$ when $\mathbf{w}_j$ is sufficiently large, and $\delta_{ij}$ denotes the kronecker delta, which equals to the $ij$-th entry of identity matrix $\mathbf{I}$. Therefore, $\mathbb{E}\boldsymbol{\eta}\boldsymbol{\eta}^\top = \mathbf{I}$. This completes the proof of (233). $\qquad\square$

The above lemma provides a Bayesian Cramer-Rao inequality, which enables us to derive the lower bound in Theorem 5.

*Proof of Theorem 5.* We apply Lemma 37. In our case, let data $X = \{(\mathbf{x}_i, y_i)\}_{i=1}^n \sim P_{\mathbf{w}^*}^{\otimes n}$. By direct calculation, we have

$$\mathcal{I}(\mathbf{w}^*) = \frac{n\mathbf{S}}{\sigma^2}. \tag{240}$$

Thus, given any prior distribution $\pi$ with support included in $W = \left\{\mathbf{w}^* \in \mathbb{R}^d : \|\mathbf{w}^*\|_{\mathbf{M}}^2 \leq 1\right\}$, by Lemma 36 we have

$$\inf_{\hat{\mathbf{w}}} \sup_{\tilde{P} \in \mathcal{P}(W, \mathbf{S}, \mathbf{T})} \mathbb{E}_{\tilde{P}^{\otimes n} \times P_\xi} \|\hat{\mathbf{w}} - \mathbf{w}^*\|_{\mathbf{T}}^2 \geq \left\langle \mathbf{T}, \left(\mathcal{I}(\pi) + \frac{n\mathbf{S}}{\sigma^2}\right)^{-1}\right\rangle. \tag{241}$$

The rest of the proof is to construct the prior distribution. To build intuition, we first consider the case $\mathbf{M} = \mathbf{I}$. We construct the prior distribution $\pi$ as follows. Given any orthogonal matrix $\mathbf{U}$ and vector $\mathbf{g}$ with $\|\mathbf{g}\| \leq 1$, we define the prior density $\pi(\mathbf{w}; \mathbf{U}, \mathbf{g})$, whose support is included in unit ball, as follows:

$$\pi(\mathbf{w}; \mathbf{U}, \mathbf{g}) = \prod_{i=1}^d \cos^2\left(\frac{\pi(\mathbf{U}^\top \mathbf{w})_i}{2\mathbf{g}_i}\right) \mathbb{1}_{|(\mathbf{U}^\top \mathbf{w})_i| \leq |\mathbf{g}_i|}, \tag{242}$$

where $\mathbb{1}$ is the indicator function. Note that $\pi(\mathbf{w}; \mathbf{U}, \mathbf{g})$ has support is included in unit ball. Direct calculation shows that the information of $\pi$ is

$$\mathcal{I}(\pi(\cdot; \mathbf{U}, \mathbf{g})) = \pi^2 \mathbf{U} \operatorname{diag}\left\{\frac{1}{\mathbf{g}_1^2}, \frac{1}{\mathbf{g}_2^2}, \dots, \frac{1}{\mathbf{g}_d^2}\right\} \mathbf{U}^\top. \tag{243}$$

For a general positive definite matrix $\mathbf{M}$, we define a prior as follows:

$$\pi(\mathbf{w}; \mathbf{U}, \mathbf{g}, \mathbf{M}) = \left(\det \mathbf{M}^{1/2}\right) \pi(\mathbf{M}^{1/2}\mathbf{w}; \mathbf{U}; \mathbf{g}). \tag{244}$$

Geometrically speaking, $\pi(\mathbf{w}; \mathbf{U}, \mathbf{g}, \mathbf{M})$ is obtained by scaling $\pi(\mathbf{w}; \mathbf{U}, \mathbf{g})$ along the eigenvector of $\mathbf{M}$, such that unit circle is transformed into the ellipse $\mathbf{x}^\top \mathbf{M} \mathbf{x} = 1$, and then normalize it by the factor $\det \mathbf{M}^{1/2}$. Note that the support of $\pi(\mathbf{w}; \mathbf{U}, \mathbf{g}, \mathbf{M})$ is included in $W = \left\{\mathbf{w}^* \in \mathbb{R}^d : \|\mathbf{w}^*\|_{\mathbf{M}}^2 \leq 1\right\}$. Then, we calculate the information matrix of $\pi(\mathbf{w}; \mathbf{U}, \mathbf{g}, \mathbf{M})$. Let $\mathbf{s}(\mathbf{w}) = \nabla \ln \pi(\mathbf{w}; \mathbf{U}; \mathbf{g})$, we have $\nabla \ln \pi(\mathbf{w}; \mathbf{U}, \mathbf{g}, \mathbf{M}) = \mathbf{M}^{1/2}\mathbf{s}(\mathbf{M}^{1/2}\mathbf{w})$. Therefore,

$$\begin{aligned}
&\mathcal{I}(\pi(\cdot; \mathbf{U}, \mathbf{g}, \mathbf{M})) \\
&= \int_{\mathbb{R}^d} \left(\mathbf{M}^{1/2}\mathbf{s}(\mathbf{M}^{1/2}\mathbf{w})\right)\left(\mathbf{M}^{1/2}\mathbf{s}(\mathbf{M}^{1/2}\mathbf{w})\right)^\top \left(\det \mathbf{M}^{1/2}\right) \pi(\mathbf{M}^{1/2}\mathbf{w}; \mathbf{U}; \mathbf{g}) d\mathbf{w} \\
&= \mathbf{M}^{1/2}\left[\int_{\mathbb{R}^d} \mathbf{s}(\mathbf{v})\mathbf{s}(\mathbf{v})^\top \pi(\mathbf{v}; \mathbf{U}; \mathbf{g}) d\mathbf{v}\right] \mathbf{M}^{1/2} \quad \left(\mathbf{v} = \mathbf{M}^{1/2}\mathbf{w}\right) \\
&\overset{a}{=} \pi^2 \mathbf{M}^{1/2}\mathbf{U} \operatorname{diag}\left\{\frac{1}{\mathbf{g}_1^2}, \frac{1}{\mathbf{g}_2^2}, \dots, \frac{1}{\mathbf{g}_d^2}\right\} \mathbf{U}^\top \mathbf{M}^{1/2},
\end{aligned} \tag{245}$$

where $\overset{a}{=}$ uses the result of the information matrix of $\pi(\mathbf{w}; \mathbf{U}; \mathbf{g})$ in (243). Therefore, all the information matrices constitute the set $\left\{\mathbf{M}^{1/2}\mathbf{F}^{-1}\mathbf{M}^{1/2} : \mathbf{F} \in \mathbb{S}_{++}^{d \times d}, \|\mathbf{F}\|_* \leq 1/\pi^2\right\}$. By applying Lemma 37, we have

$$\begin{aligned}
\inf_{\hat{\mathbf{w}}} \sup_{\tilde{P} \in \mathcal{P}(W, \mathbf{S}, \mathbf{T})} \mathbb{E}_{\tilde{P}^{\otimes n} \times P_\xi} \|\hat{\mathbf{w}} - \mathbf{w}^*\|_{\mathbf{T}}^2 &\geq \sup_{\substack{\mathbf{F} \succeq \mathbf{O} \\ \|\mathbf{F}\|_* \leq 1/\pi^2}} \left\langle \mathbf{T}, \left(\mathbf{M}^{1/2}\mathbf{F}^{-1}\mathbf{M}^{1/2} + \frac{n\mathbf{S}}{\sigma^2}\right)^{-1}\right\rangle \\
&= \sup_{\substack{\mathbf{F} \succeq \mathbf{O} \\ \|\mathbf{F}\|_* \leq 1/\pi^2}} \left\langle \mathbf{T}', \left(\mathbf{F}^{-1} + \frac{n\mathbf{S}'}{\sigma^2}\right)^{-1}\right\rangle.
\end{aligned} \tag{246}$$

This completes the proof. $\qquad \square$

## C.2 PROOF OF LINEAR PRECONDITIONED ESTIMATOR IN SECTION 3.3.1

**Lemma 38** (Upper Bound). *Suppose we get samples $\{(\mathbf{x}_i, y_i)\}_{i=1}^n$ drawn from the source distribution of $\tilde{P}$. The excess risk of the optimal estimator $\hat{\mathbf{w}}_{\mathbf{A}}$ defined in (6) on the target distribution of $\tilde{P}$ can be bounded from above by:*

$$
\sup_{\tilde{P} \in \mathcal{P}(W, \mathbf{S}, \mathbf{T})} \mathbb{E}_{\tilde{P}^{\otimes n}} \|\hat{\mathbf{w}}_{\mathbf{A}} - \mathbf{w}^*\|_{\mathbf{T}}^2
$$
$$
\leq \min_{\mathbf{A} \in \mathbb{R}^{d \times d}} \left\| (\mathbf{I} - \mathbf{A})^\top \mathbf{T}'(\mathbf{I} - \mathbf{A}) \right\| + \frac{2\sigma^2 + 2\psi \|\mathbf{S}'\|}{n} \left\langle \mathbf{T}', \mathbf{A} \left(\mathbf{S}'\right)^{-1} \mathbf{A}^\top \right\rangle, \tag{247}
$$

*where $\mathbf{S}' = \mathbf{M}^{-1/2} \mathbf{S} \mathbf{M}^{-1/2}$ and $\mathbf{T}' = \mathbf{M}^{-1/2} \mathbf{T} \mathbf{M}^{-1/2}$.*

*Proof.* Let $\hat{\mathbf{w}} = \frac{1}{n} \mathbf{S}^{-1} \sum_{i=1}^n \mathbf{x}_i y_i$. Then we have $\hat{\mathbf{w}}_{\mathbf{A}} = \mathbf{A} \hat{\mathbf{w}}$. We first show that

$$
\mathbb{E}\hat{\mathbf{w}} = \mathbf{w}^*, \quad \text{cov } \hat{\mathbf{w}} \preceq \frac{2\sigma^2 + 2\psi \|\mathbf{w}^*\|_{\mathbf{S}}^2}{n} \mathbf{S}^{-1}. \tag{248}
$$

Denote $\epsilon_i = y_i - \mathbf{x}_i^\top \mathbf{w}^*$ as the response noise. Since $\mathbf{w}^*$ is an optimal parameter, we have $\mathbb{E}\epsilon_i \mathbf{x}_i = 0$. Recall that $\mathbf{S} = \mathbb{E}\mathbf{x}\mathbf{x}^\top$, and $\{(\mathbf{x}_i, y_i)\}_{i=1}^n$ are i.i.d, we have

$$
\mathbb{E}\hat{\mathbf{w}} = \frac{1}{n} \mathbf{S}^{-1} \sum_{i=1}^n \mathbb{E}\left[ \mathbf{x}_i \left( \mathbf{x}_i^\top \mathbf{w}^* + \epsilon_i \right) \right] = \mathbf{S}^{-1} \mathbb{E}_{P_{\mathbf{x}}} \left[ \mathbf{x}\mathbf{x}^\top \right] \mathbf{w}^* = \mathbf{w}^*. \tag{249}
$$

Furthermore,

$$
\begin{aligned}
\text{cov } \hat{\mathbf{w}} &\overset{a}{\preceq} \frac{1}{n} \mathbf{S}^{-1} \mathbb{E}_{P_{\mathbf{x} \times y}} \left[ y^2 \mathbf{x}\mathbf{x}^\top \right] \mathbf{S}^{-1} = \frac{1}{n} \mathbf{S}^{-1} \mathbb{E}_{P_{\mathbf{x} \times y}} \left[ \left( \mathbf{x}^\top \mathbf{w}^* + \epsilon \right)^2 \mathbf{x}\mathbf{x}^\top \right] \mathbf{S}^{-1} \\
&\overset{b}{\preceq} \frac{2}{n} \mathbf{S}^{-1} \left( \mathbb{E}_{P_{\mathbf{x}}} \left[ \left( \mathbf{x}^\top \mathbf{w}^* \right)^2 \mathbf{x}\mathbf{x}^\top \right] + \mathbb{E}_{P_{\mathbf{x} \times y}} \left[ \epsilon^2 \mathbf{x}\mathbf{x}^\top \right] \right) \mathbf{S}^{-1} \\
&\overset{c}{\preceq} \frac{2}{n} \mathbf{S}^{-1} \left( \psi \|\mathbf{w}^*\|_{\mathbf{S}}^2 \mathbf{S} + \sigma^2 \mathbf{S} \right) \mathbf{S}^{-1} \\
&= \frac{2\sigma^2 + 2\psi \|\mathbf{w}^*\|_{\mathbf{S}}^2}{n} \mathbf{S}^{-1},
\end{aligned} \tag{250}
$$

where $\overset{a}{=}$ applies cov $\hat{\mathbf{w}} \preceq \mathbb{E}\left[ \hat{\mathbf{w}}\hat{\mathbf{w}}^\top \right]$ and $\hat{\mathbf{w}}$ is the average of $n$ independent random variable, $\overset{b}{\preceq}$ uses the inequality $(a+b)^2 \leq 2a^2 + 2b^2$, and $\overset{c}{\preceq}$ uses Assumption 4 and Assumption 1.

Since $\hat{\mathbf{w}}_{\mathbf{A}} = \mathbf{M}^{-1/2} \mathbf{A} \mathbf{M}^{1/2} \hat{\mathbf{w}}$, we have

$$
\mathbb{E}\hat{\mathbf{w}}_{\mathbf{A}} = \mathbf{M}^{-1/2} \mathbf{A} \mathbf{M}^{1/2} \mathbf{w}^*, \tag{251}
$$

$$
\text{cov } \hat{\mathbf{w}}_{\mathbf{A}} \preceq \frac{2\sigma^2 + 2\psi \|\mathbf{w}^*\|_{\mathbf{S}}^2}{n} \left( \mathbf{M}^{-1/2} \mathbf{A} \mathbf{M}^{1/2} \right) \mathbf{S}^{-1} \left( \mathbf{M}^{-1/2} \mathbf{A} \mathbf{M}^{1/2} \right)^\top. \tag{252}
$$

Apply the bias-variance decomposition to $\mathbb{E}\|\hat{\mathbf{w}}_{\mathbf{A}} - \mathbf{w}^*\|_{\mathbf{T}}^2$, we obtain

$$
\mathbb{E}\|\hat{\mathbf{w}}_{\mathbf{A}} - \mathbf{w}^*\|_{\mathbf{T}}^2 = \|\mathbb{E}\hat{\mathbf{w}}_{\mathbf{A}} - \mathbf{w}^*\|_{\mathbf{T}}^2 + \langle \mathbf{T}, \text{cov } \hat{\mathbf{w}}_{\mathbf{A}} \rangle. \tag{253}
$$

Recall that $\mathbf{S}' = \mathbf{M}^{-1/2} \mathbf{S} \mathbf{M}^{-1/2}$ and $\mathbf{T}' = \mathbf{M}^{-1/2} \mathbf{T} \mathbf{M}^{-1/2}$. For the bias term, we have

$$
\begin{aligned}
\|\mathbb{E}\hat{\mathbf{w}}_{\mathbf{A}} - \mathbf{w}^*\|_{\mathbf{T}}^2 &= \left\| \left( \mathbf{I} - \mathbf{M}^{-1/2} \mathbf{A} \mathbf{M}^{1/2} \right) \mathbf{w}^* \right\|_{\mathbf{T}}^2 = \left\| \mathbf{M}^{-1/2} \left( \mathbf{I} - \mathbf{A} \right) \mathbf{M}^{1/2} \mathbf{w}^* \right\|_{\mathbf{T}}^2 \\
&= \left\| (\mathbf{I} - \mathbf{A}) \mathbf{M}^{1/2} \mathbf{w}^* \right\|_{\mathbf{T}'}^2.
\end{aligned} \tag{254}
$$

For the variance term, we have

$$
\begin{aligned}
\langle \mathbf{T}, \text{cov } \hat{\mathbf{w}}_{\mathbf{A}} \rangle &\leq \frac{2\sigma^2 + 2\psi \|\mathbf{w}^*\|_{\mathbf{S}}^2}{n} \left\langle \mathbf{M}^{-1/2} \mathbf{T} \mathbf{M}^{-1/2}, \mathbf{A} \left( \mathbf{M}^{-1/2} \mathbf{S} \mathbf{M}^{-1/2} \right)^{-1} \mathbf{A}^\top \right\rangle \\
&\leq \frac{2\sigma^2 + 2\psi \left\| \mathbf{M}^{1/2} \mathbf{w}^* \right\|_{\mathbf{S}'}^2}{n} \left\langle \mathbf{T}', \mathbf{A} \left( \mathbf{S}' \right)^{-1} \mathbf{A}^\top \right\rangle.
\end{aligned} \tag{255}
$$

Take the supremum with respect to $\mathbf{w}^* \in W = \left\{ \mathbf{w}^* \in \mathbb{R}^d : \|\mathbf{w}^*\|_{\mathbf{M}}^2 \le 1 \right\}$, and note that

$$\sup_{\mathbf{w}^* \in W} \left\| (\mathbf{I} - \mathbf{A})\mathbf{M}^{1/2}\mathbf{w}^* \right\|_{\mathbf{T}'}^2 = \left\| (\mathbf{I} - \mathbf{A})^\top \mathbf{T}'(\mathbf{I} - \mathbf{A}) \right\|, \quad \sup_{\mathbf{w}^* \in W} \left\| \mathbf{M}^{1/2}\mathbf{w}^* \right\|_{\mathbf{S}'}^2 = \|\mathbf{S}'\|. \quad (256)$$

Thus, we obtain

$$\sup_{\tilde{P} \in \mathcal{P}(W, \mathbf{S}, \mathbf{T})} \mathbb{E} \|\hat{\mathbf{w}}_{\mathbf{A}} - \mathbf{w}^*\|_{\mathbf{T}}^2 \le \left\| (\mathbf{I} - \mathbf{A})^\top \mathbf{T}'(\mathbf{I} - \mathbf{A}) \right\| + \frac{2\sigma^2 + 2\psi \|\mathbf{S}'\|}{n} \left\langle \mathbf{T}', \mathbf{A} \left( \mathbf{S}' \right)^{-1} \mathbf{A}^\top \right\rangle. \quad (257)$$

Minimizing the RHS with respect to $\mathbf{A}$ completes the proof. $\qquad\qquad\qquad\qquad \square$

### C.3 MATCHING BOUNDS

**Lemma 39** (Matching Bounds). *For any positive definite matrix $\mathbf{S}$, $\mathbf{M}$ and positive semi-definite matrix $\mathbf{T}$, the following equation holds:*

$$\sup_{\substack{\mathbf{F} \succeq \mathbf{O} \\ \|\mathbf{F}\|_* \le 1/\pi^2}} \left\langle \mathbf{T}', \left( \mathbf{F}^{-1} + \frac{n\mathbf{S}'}{\sigma^2} \right)^{-1} \right\rangle = \min_{\mathbf{A} \in \mathbb{R}^{d \times d}} \frac{1}{\pi^2} \left\| (\mathbf{I} - \mathbf{A})^\top \mathbf{T}'(\mathbf{I} - \mathbf{A}) \right\| + \frac{\sigma^2}{n} \left\langle \mathbf{T}', \mathbf{A} \left( \mathbf{S}' \right)^{-1} \mathbf{A}^\top \right\rangle,$$

*where $\mathbf{S}' = \mathbf{M}^{-1/2}\mathbf{S}\mathbf{M}^{-1/2}$ and $\mathbf{T}' = \mathbf{M}^{-1/2}\mathbf{T}\mathbf{M}^{-1/2}$.*

The proof of Lemma 39 is divided into two parts. First, we assume $\mathbf{T}'$ is invertible, and solves the optimization problem in Theorem 5 to derive the result. For the second part, we replace $\mathbf{T}'$ by $\mathbf{T}' + \epsilon \mathbf{I}$, which is invertible, and take $\epsilon \to 0$ to complete the proof.

*Proof.* For simplicity, let

$$L(\mathbf{S}', \mathbf{T}') = \sup_{\substack{\mathbf{F} \succeq \mathbf{O} \\ \|\mathbf{F}\|_* \le 1/\pi^2}} \left\langle \mathbf{T}', \left( \mathbf{F}^{-1} + \frac{n\mathbf{S}'}{\sigma^2} \right)^{-1} \right\rangle, \quad (258)$$

$$U(\mathbf{S}', \mathbf{T}') = \inf_{\mathbf{A} \in \mathbb{R}^{d \times d}} \frac{1}{\pi^2} \left\| (\mathbf{I} - \mathbf{A})^\top \mathbf{T}'(\mathbf{I} - \mathbf{A}) \right\| + \frac{\sigma^2}{n} \left\langle \mathbf{T}', \mathbf{A} \left( \mathbf{S}' \right)^{-1} \mathbf{A}^\top \right\rangle. \quad (259)$$

For the first part of the proof, we assume $\mathbf{T}'$ is invertible. We solve the optimization problem in Theorem 5. Note that the objective function $\left\langle \mathbf{T}', \left( \mathbf{F}^{-1} + \frac{n\mathbf{S}'}{\sigma^2} \right)^{-1} \right\rangle$ is concave with respect to $\mathbf{F}$ and the feasible set $\left\{ \mathbf{F} \in \mathbb{S}_{++}^{d \times d} : \|\mathbf{F}\|_* \le 1/\pi^2 \right\}$ is a convex set. Therefore, we can introduce a Lagrange multiplier $\boldsymbol{\Delta} \in \mathbb{S}^{d \times d}$ and obtain

$$L(\mathbf{S}', \mathbf{T}') = \sup_{\substack{\mathbf{F} \in \mathbb{S}^{d \times d} \\ \mathbf{B} \succeq \mathbf{O} \\ \|\mathbf{B}\|_* \le 1/\pi^2}} \inf_{\boldsymbol{\Delta} \in \mathbb{S}^{d \times d}} \left\langle \mathbf{T}', \left( \mathbf{F}^{-1} + \frac{n\mathbf{S}'}{\sigma^2} \right)^{-1} \right\rangle + \left\langle \mathbf{S}' \left( \mathbf{T}' \right)^{-1/2} \boldsymbol{\Delta} \left( \mathbf{T}' \right)^{-1/2} \mathbf{S}', \mathbf{B} - \mathbf{F} \right\rangle$$

$$\overset{a}{=} \inf_{\boldsymbol{\Delta} \in \mathbb{S}^{d \times d}} \sup_{\mathbf{F} \in \mathbb{S}^{d \times d}} \left[ \left\langle \mathbf{T}', \left( \mathbf{F}^{-1} + \frac{n\mathbf{S}'}{\sigma^2} \right)^{-1} \right\rangle - \left\langle \mathbf{S}' \left( \mathbf{T}' \right)^{-1/2} \boldsymbol{\Delta} \left( \mathbf{T}' \right)^{-1/2} \mathbf{S}', \mathbf{F} \right\rangle \right]$$

$$\qquad + \sup_{\substack{\mathbf{B} \succeq \mathbf{O} \\ \|\mathbf{B}\|_* \le 1/\pi^2}} \left\langle \mathbf{S}' \left( \mathbf{T}' \right)^{-1/2} \boldsymbol{\Delta} \left( \mathbf{T}' \right)^{-1/2} \mathbf{S}', \mathbf{B} \right\rangle$$

$$\overset{b}{=} \inf_{\boldsymbol{\Delta} \in \mathbb{S}^{d \times d}} \underbrace{\sup_{\mathbf{F} \in \mathbb{S}^{d \times d}} \left[ \left\langle \mathbf{T}', \left( \mathbf{F}^{-1} + \frac{n\mathbf{S}'}{\sigma^2} \right)^{-1} \right\rangle - \left\langle \mathbf{S}' \left( \mathbf{T}' \right)^{-1/2} \boldsymbol{\Delta} \left( \mathbf{T}' \right)^{-1/2} \mathbf{S}', \mathbf{F} \right\rangle \right]}_{(a)}$$

$$\qquad + \frac{1}{\pi^2} \left\| \mathbf{S}' \left( \mathbf{T}' \right)^{-1/2} \boldsymbol{\Delta} \left( \mathbf{T}' \right)^{-1/2} \mathbf{S}' \right\|, \quad (260)$$

where $\stackrel{a}{=}$ follows from the concavity of $\left\langle \mathbf{T}', \left(\mathbf{F}^{-1} + \frac{n\mathbf{S}'}{\sigma^2}\right)^{-1} \right\rangle$ with respect to $\mathbf{F}$ and the convexity

of feasible set $\left\{ \mathbf{F} \in \mathbb{S}_{++}^{d \times d} : \|\mathbf{F}\|_* \leq 1/\pi^2 \right\}$, and $\stackrel{b}{=}$ follows from the fact that the dual norm of nuclear norm $\|\cdot\|_*$ is 2-norm $\|\cdot\|$. To solve $(a)$, let the derivative of $(a)$ with respect to $\mathbf{F}$ be equal to $\mathbf{O}$, we get

$$\left(\mathbf{I} + \frac{n\mathbf{S}'\mathbf{F}}{\sigma^2}\right)^{-1} \mathbf{T}' \left(\mathbf{I} + \frac{n\mathbf{F}\mathbf{S}'}{\sigma^2}\right)^{-1} - \mathbf{S}' \left(\mathbf{T}'\right)^{-1/2} \boldsymbol{\Delta} \left(\mathbf{T}'\right)^{-1/2} \mathbf{S}' = \mathbf{O}. \tag{261}$$

Note that if $\boldsymbol{\Delta}$ is not a PSD matrix, then $(a) = +\infty$. Thus, $\boldsymbol{\Delta} \succeq \mathbf{O}$. Let $\left(\mathbf{I} + \frac{n\mathbf{S}'\mathbf{F}}{\sigma^2}\right)^{-1} \left(\mathbf{T}'\right)^{1/2} = \mathbf{S}' \left(\mathbf{T}'\right)^{-1/2} \boldsymbol{\Delta}^{1/2}$. Solve the equation yields

$$\mathbf{F} = \frac{\sigma^2}{n} \left[ \left(\mathbf{S}'\right)^{-1} \left(\mathbf{T}'\right)^{1/2} \boldsymbol{\Delta}^{-1/2} \left(\mathbf{T}'\right)^{1/2} \left(\mathbf{S}'\right)^{-1} - \left(\mathbf{S}'\right)^{-1} \right], \tag{262}$$

which meets the requirement that $\mathbf{F}$ is a PSD matrix. Plugging the solution into $(a)$, we have

$$(a) = \frac{\sigma^2}{n} \left[ \left\langle \mathbf{S}', \left(\mathbf{T}'\right)^{-1/2} \boldsymbol{\Delta} \left(\mathbf{T}'\right)^{-1/2} \right\rangle - 2\operatorname{tr} \boldsymbol{\Delta}^{1/2} + \operatorname{tr} \mathbf{T}' \left(\mathbf{S}'\right)^{-1} \right]$$

$$= \frac{\sigma^2}{n} \left\langle \mathbf{T}', \left[ \left(\mathbf{T}'\right)^{-1/2} \boldsymbol{\Delta}^{1/2} \left(\mathbf{T}'\right)^{-1/2} \mathbf{S}' - \mathbf{I} \right] \left(\mathbf{S}'\right)^{-1} \left[ \mathbf{S}' \left(\mathbf{T}'\right)^{-1/2} \boldsymbol{\Delta}^{1/2} \left(\mathbf{T}'\right)^{-1/2} - \mathbf{I} \right] \right\rangle.$$

Let $\mathbf{A} = \mathbf{I} - \left(\mathbf{T}'\right)^{-1/2} \boldsymbol{\Delta}^{1/2} \left(\mathbf{T}'\right)^{-1/2} \mathbf{S}'$, we obtain

$$L(\mathbf{S}', \mathbf{T}') = \inf_{\mathbf{S}'(\mathbf{I}-\mathbf{A}) \in \mathbb{S}_+^{d \times d}} \frac{1}{\pi^2} \left\| (\mathbf{I} - \mathbf{A})^\top \mathbf{T}'(\mathbf{I} - \mathbf{A}) \right\| + \frac{\sigma^2}{n} \left\langle \mathbf{T}', \mathbf{A} \left(\mathbf{S}'\right)^{-1} \mathbf{A}^\top \right\rangle. \tag{263}$$

Note that the definition of $U(\mathbf{S}', \mathbf{T}')$ in (259) imposes no constraint $\mathbf{A}$. Now we show that the constraint $\mathbf{S}'(\mathbf{I} - \mathbf{A}) \in \mathbb{S}_+^{d \times d}$ in (263) can be relaxed to $\mathbf{A} \in \mathbb{R}^{d \times d}$. For any $\mathbf{A} \in \mathbb{R}^{d \times d}$, we denote the polar decomposition of $\left(\mathbf{T}'\right)^{1/2} (\mathbf{I} - \mathbf{A}) \left(\mathbf{S}'\right)^{-1} \left(\mathbf{T}'\right)^{1/2}$ as:

$$\mathbf{U}\mathbf{Z} = \left(\mathbf{T}'\right)^{1/2} (\mathbf{I} - \mathbf{A}) \left(\mathbf{S}'\right)^{-1} \left(\mathbf{T}'\right)^{1/2}, \tag{264}$$

where $\mathbf{U}$ is an orthogonal matrix and $\mathbf{Z}$ is a PSD matrix. Substitute (264) into the objective function of $U(\mathbf{S}', \mathbf{T}')$ shown in (265), we have

$$\frac{\sigma^2}{n} \left\langle \mathbf{T}', \mathbf{A} \left(\mathbf{S}'\right)^{-1} \mathbf{A}^\top \right\rangle + \frac{1}{\pi^2} \left\| (\mathbf{I} - \mathbf{A})^\top \mathbf{T}'(\mathbf{I} - \mathbf{A}) \right\| \tag{265}$$

$$\stackrel{a}{=} \frac{\sigma^2}{n} \left\langle \mathbf{T}', \left(\mathbf{I} - \left(\mathbf{T}'\right)^{-1/2} \mathbf{U}\mathbf{Z} \left(\mathbf{T}'\right)^{-1/2} \mathbf{S}'\right) \left(\mathbf{S}'\right)^{-1} \left(\mathbf{I} - \mathbf{S}' \left(\mathbf{T}'\right)^{-1/2} \mathbf{Z}^\top \mathbf{U}^\top \left(\mathbf{T}'\right)^{-1/2}\right) \right\rangle$$

$$+ \frac{1}{\pi^2} \left\| \mathbf{S}' \left(\mathbf{T}'\right)^{-1/2} \mathbf{Z}^\top \mathbf{U}^\top \mathbf{U}\mathbf{Z} \left(\mathbf{T}'\right)^{-1/2} \mathbf{S}' \right\|$$

$$= \frac{\sigma^2}{n} \left[ \operatorname{tr} \left( \mathbf{U}\mathbf{Z} \left(\mathbf{T}'\right)^{-1/2} \mathbf{S}' \left(\mathbf{T}'\right)^{-1/2} \mathbf{Z}^\top \mathbf{U}^\top \right) - \operatorname{tr} \left( \mathbf{U}\mathbf{Z} + \mathbf{Z}^\top \mathbf{U}^\top \right) + \operatorname{tr} \left( \mathbf{T}' \left(\mathbf{S}'\right)^{-1} \right) \right]$$

$$+ \frac{1}{\pi^2} \left\| \mathbf{S}' \left(\mathbf{T}'\right)^{-1/2} \mathbf{Z}^\top \mathbf{U}^\top \mathbf{U}\mathbf{Z} \left(\mathbf{T}'\right)^{-1/2} \mathbf{S}' \right\|$$

$$\stackrel{a}{=} \frac{\sigma^2}{n} \left[ \left\langle \mathbf{S}', \left(\mathbf{T}'\right)^{-1/2} \mathbf{Z}^2 \left(\mathbf{T}'\right)^{-1/2} \right\rangle - \operatorname{tr} \left( \mathbf{U}\mathbf{Z} + \mathbf{Z}^\top \mathbf{U}^\top \right) + \operatorname{tr} \left( \mathbf{T}' \left(\mathbf{S}'\right)^{-1} \right) \right] \tag{266}$$

$$+ \frac{1}{\pi^2} \left\| \mathbf{S}' \left(\mathbf{T}'\right)^{-1/2} \mathbf{Z}^2 \left(\mathbf{T}'\right)^{-1/2} \mathbf{S}' \right\|,$$

where $\stackrel{a}{=}$ uses $\mathbf{A} = \mathbf{I} - \left(\mathbf{T}'\right)^{-1/2} \mathbf{U}\mathbf{Z} \left(\mathbf{T}'\right)^{-1/2} \mathbf{S}'$, and $\stackrel{b}{=}$ uses $\mathbf{U}\mathbf{U}^\top = \mathbf{I}$ and $\mathbf{Z}$ is a PSD matrix. We first minimize (266) with respect to $\mathbf{U}$. By Lemma 40, $-\operatorname{tr} \left( \mathbf{U}\mathbf{Z} + \mathbf{Z}^\top \mathbf{U}^\top \right)$ is minimized when $\mathbf{U} = \mathbf{I}$, which implies $\mathbf{S}'(\mathbf{I} - \mathbf{A}) = \mathbf{S}' \left(\mathbf{T}'\right)^{-1/2} \mathbf{U}\mathbf{Z} \left(\mathbf{T}'\right)^{-1/2} \mathbf{S}' \in \mathbb{S}_+^{d \times d}$. Therefore, we have

$$\inf_{\mathbf{A} \in \mathbb{R}^{d \times d}} (265) = \inf_{\mathbf{Z} \in \mathbb{S}_+^{d \times d}} \inf_{\substack{\mathbf{U} \in \mathbb{R}^{d \times d} \\ \mathbf{U}\mathbf{U}^\top = \mathbf{I}}} (266) = \inf_{\mathbf{S}'(\mathbf{I}-\mathbf{A}) \in \mathbb{S}_+^{d \times d}} (265), \tag{267}$$

We complete the first part by noting that $U(\mathbf{S}', \mathbf{T}') = \inf_{\mathbf{A} \in \mathbb{R}^{d \times d}}$ (265) by definition and $L(\mathbf{S}', \mathbf{T}') = \inf_{\mathbf{S}'(\mathbf{I}-\mathbf{A}) \in \mathbb{S}_+^{d \times d}}$ (265) which is shown in (263).

For the second part of the proof, we consider the case where $\mathbf{T}'$ is any PSD matrix, *i.e.* $\mathbf{T}'$ is possibly singular. Let $\epsilon > 0$ be arbitrary. Since $L(\mathbf{S}', \mathbf{T}')$ is linear in $\mathbf{T}'$, we have

$$L(\mathbf{S}', \mathbf{T}') \leq L(\mathbf{S}', \mathbf{T}' + \epsilon \mathbf{I}) \leq L(\mathbf{S}', \mathbf{T}') + \epsilon L(\mathbf{S}', \mathbf{I}) \tag{268}$$

Note that

$$\sup_{\substack{\mathbf{F} \succeq \mathbf{O} \\ \|\mathbf{F}\|_* \leq 1/\pi^2}} \left\langle \mathbf{I}, \left( \mathbf{F}^{-1} + \frac{n\mathbf{S}'}{\sigma^2} \right)^{-1} \right\rangle \leq \sup_{\substack{\mathbf{F} \succeq \mathbf{O} \\ \|\mathbf{F}\|_* \leq 1/\pi^2}} \operatorname{tr} \mathbf{F} \leq \frac{1}{\pi^2}. \tag{269}$$

Therefore, we have

$$L(\mathbf{S}', \mathbf{T}') = \lim_{\epsilon \to 0^+} L(\mathbf{S}', \mathbf{T}' + \epsilon \mathbf{I}). \tag{270}$$

Similarly, we have

$$\inf_{\mathbf{A} \in \mathbb{R}^{d \times d}} \frac{1}{\pi^2} \left\| (\mathbf{I} - \mathbf{A})^\top \mathbf{T}'(\mathbf{I} - \mathbf{A}) \right\| + \frac{\sigma^2}{n} \left\langle \mathbf{T}', \mathbf{A} \left( \mathbf{S}' \right)^{-1} \mathbf{A}^\top \right\rangle$$

$$\leq \inf_{\mathbf{A} \in \mathbb{R}^{d \times d}} \frac{1}{\pi^2} \left\| (\mathbf{I} - \mathbf{A})^\top (\mathbf{T}' + \epsilon \mathbf{I})(\mathbf{I} - \mathbf{A}) \right\| + \frac{\sigma^2}{n} \left\langle \mathbf{T}' + \epsilon \mathbf{I}, \mathbf{A} \left( \mathbf{S}' \right)^{-1} \mathbf{A}^\top \right\rangle$$

$$\leq \frac{1}{\pi^2} \left\| (\mathbf{I} - \mathbf{A}_0)^\top \mathbf{T}'(\mathbf{I} - \mathbf{A}_0) \right\| + \frac{\sigma^2}{n} \left\langle \mathbf{T}', \mathbf{A}_0 \left( \mathbf{S}' \right)^{-1} \mathbf{A}_0^\top \right\rangle \tag{271}$$

$$+ \epsilon \left[ \frac{1}{\pi^2} \left\| (\mathbf{I} - \mathbf{A}_0)^\top (\mathbf{I} - \mathbf{A}_0) \right\| + \frac{\sigma^2}{n} \operatorname{tr} \left( \mathbf{A}_0 \left( \mathbf{S}' \right)^{-1} \mathbf{A}_0^\top \right) \right],$$

where $\mathbf{A}_0$ is a minimizer of $\frac{1}{\pi^2} \left\| (\mathbf{I} - \mathbf{A})^\top \mathbf{T}'(\mathbf{I} - \mathbf{A}) \right\| + \frac{\sigma^2}{n} \left\langle \mathbf{T}', \mathbf{A} \left( \mathbf{S}' \right)^{-1} \mathbf{A}^\top \right\rangle$. Thus, we have

$$U(\mathbf{S}', \mathbf{T}') = \lim_{\epsilon \to 0^+} U(\mathbf{S}', \mathbf{T}' + \epsilon \mathbf{I}). \tag{272}$$

Finally, combine (270), (272) and the first part of the proof, we obtain

$$L(\mathbf{S}', \mathbf{T}') = \lim_{\epsilon \to 0^+} L(\mathbf{S}', \mathbf{T}' + \epsilon \mathbf{I}) = \lim_{\epsilon \to 0^+} U(\mathbf{S}', \mathbf{T}' + \epsilon \mathbf{I}) = U(\mathbf{S}', \mathbf{T}'). \tag{273}$$

This completes the proof for any PSD matrix $\mathbf{T}'$. $\qquad\square$

**Lemma 40.** *Let $\mathbf{Z}$ be a PSD matrix and $\mathbf{U}$ be a orthogonal matrix. Then $\operatorname{tr}(\mathbf{U}\mathbf{Z}) \leq \operatorname{tr} \mathbf{Z}$.*

*Proof.* Without loss of generality, we assume $\mathbf{Z} = \operatorname{diag}\{z_1, z_2 \ldots, z_d\}$. Let $u_{ij}$ denote the $ij$-th entry of $\mathbf{U}$. Note that $\mathbf{U}$ is orthogonal implies $|u_{ij}| \leq 1$, so

$$\operatorname{tr}(\mathbf{U}\mathbf{Z}) = \sum_{i=1}^{d} u_{ii} z_i \leq \sum_{i=1}^{d} z_i = \operatorname{tr} \mathbf{Z}, \tag{274}$$

where $=$ holds when $\mathbf{U} = \mathbf{I}$. $\qquad\square$

## D  WHEN IS EMERGENCE POSSIBLE?

When scaling up the training of large language models, models may suddenly perform much better on downstream tasks after hitting a critical sample size—an amazing phenomenon often referred to as emergence (Wei et al., 2022). Under the covariate shift setting, emergence can arise when downstream tasks demand high-quality estimation in localized source spectral regions, despite the source excess risk decreasing smoothly. Specifically, when the downstream task emphasizes directions corresponding to a certain eigensubspace $\mathbf{S}_{k_1:k_2}$, the phase transition in ASGD's bias-reduction capability indicates that the target excess risk remains flat until the effective dimension $k^*$ surpasses $k_2$. Consequently, the target excess risk of ASGD exhibits a sharp transition—from a plateau to rapid decline, when the sample size reaches $n = ((\gamma + \delta)\lambda_{k_2})^{-1}$, while the source excess risk continues its gradual decrease. The following provides an illustrative example of this mechanism.

**Example 1.** *We suppose* $\mathbf{S} = \mathrm{diag}\{i^{-a}\}_{i=1}^d$ *and* $\mathbf{M} = \mathbf{I}$. *Let* $d_0 \in [d]$, *we consider target covariance matrix* $\mathbf{T} = \mathrm{diag}\left\{d_0^{(1+r)a}\left(\max\{i, d_0\}\right)^{-(1+r)a}\right\}_{i=1}^d$, *where* $-1 < r < 1/a$. *There exists* $\mathbf{w}^* \in W_{\mathbf{I}}$, *such that the source and target excess risk of SGD with* $\delta = \gamma = \tilde{\Theta}\left(n^{-\frac{1}{a+1}}\right)$ *satisfy:*

$$\mathbb{E}_{\tilde{P}^{\otimes n}}\left\|\mathbf{w}_n^{\mathrm{SGD}} - \mathbf{w}^*\right\|_{\mathbf{T}}^2 \approx \tilde{\Theta}\left((1/n)^{\frac{a}{a+1}}\right) \tag{275}$$

$$\mathbb{E}_{\tilde{P}^{\otimes n}}\left\|\mathbf{w}_n^{\mathrm{SGD}} - \mathbf{w}^*\right\|_{\mathbf{T}}^2 \approx \begin{cases} \tilde{\Theta}\left(1\right), & n \lesssim d_0^{a+1}; \\ \tilde{\Theta}\left((d_0^{a+1}/n)^{\frac{(1+r)a}{1+a}}\right), & n \gtrsim d_0^{a+1}. \end{cases} \tag{276}$$

This example demonstrates that even when SGD achieves optimality, the emergent phenomenon can still occur. This illustrates that emergence is an inherent consequence of downstream tasks placing disproportionate emphasis on specific regions of the source representation.

# E  MORE OPTIMAL FUNCTION CLASS

The following corollary of Theorem 8 further show that SGD can achieve optimality over a Gaussian distribution class with bounded KL divergence.

**Corollary 14** (Gaussian $D_{\mathrm{KL}}$ Bounded Class). *Let constant* $C > 0$ *and* $\mathbf{M} = \mathbf{I}$. *The* $D_{\mathrm{KL}}$-*bounded class* $\mathcal{P}_{C,\mathbf{T}}^{\mathrm{KL}}$ *includes problem instances such that* $P_{\mathbf{x}}$ *and* $Q_{\mathbf{x}}$ *are Gaussian,* $\mathbb{E}_{Q_{\mathbf{x}}}\mathbf{x}\mathbf{x}^\top = \mathbf{T}$ *and* $D_{\mathrm{KL}}(Q_{\mathbf{x}}\|P_{\mathbf{x}}^{\mathrm{KL}}) \leq C$. *Under the regularity condition, and assume* $k_{\mathrm{KL}}^* = \max_k\left\{\lambda_k^{\mathrm{KL}} \geq \mu_{d_1}\right\} \leq k^{\max}$, *where* $d_1 = \max_i\left\{\mu_i \geq \sigma^2 i/n\right\}$. *SGD with* $\gamma \approx (\ln n)^2/(n\lambda_{k_{\mathrm{KL}}^*})$ *and* $\delta \approx \min\left\{\gamma, 1/(\mathrm{tr}\,\mathbf{S}\ln n)\right\}$ *can achieve the optimal rate* $\tilde{\mathcal{O}}(\inf_{\delta>0}\left\{\delta^2 + \frac{\sigma^2 d(\delta)}{n}\right\})$.

# F  EXPERIMENT DETAILS

## F.1  EXPERIMENT DETAILS OF SECTION 6

All experiments are conducted 100 times, and we calculate 95% confidence intervals. We introduce covariate shift by assigning each image of age $y$ to the source domain with probability $p(y) = 1/\left(1 + \exp\left(-\frac{y-40}{20}\right)\right)$ and to the target otherwise. We perform a grid search on the hyperparameters of both ridge and ASGD based on the validation loss.

## F.2  SIMULATIONS

This section presents the details on the simulation results. We repeat each simulation 100 times and report the average result and 95% confidence interval. Dashed lines show the theoretical rate. Unless specified, we choose dimension $d = 50000$, source covariate $\mathbf{x} \sim \mathcal{N}(\mathbf{0}, \mathbf{S})$, $\mathbf{S} = \mathrm{diag}\{i^{-a}\}_{i=1}^d$ and $\mathbf{M} = \mathbf{I}$.

- **Figure 2 (a) Comparison of SGD and Ridge in the setting of Theorem 11.** We set $d = 5000$, SGD learning rate $\gamma = 100n^{-1}$; $\mathbf{S}$ and $\mathbf{T}$ are set according to Theorem 11. Ridge only achieves sub-optimal rate $1/\sqrt{n}$, while SGD achieves minimax rate $1/n$.

- **Figure 2 (b) Asymptotic convergence rate of SGD in the setting of Corollary 9.** We set $d = 10$, SGD learning rate $\gamma = 0.1$, $\lambda_i = i^{-a}$, and target covariance matrix $\mathbf{T} = \mathbf{U}\,\mathrm{diag}\{i^{-a}\}_{i=1}^d\mathbf{U}^\top$, where $\mathbf{U}$ is a random orthogonal matrix.

- **Figure 2 (c) Simulation of Corollary 10.** Let $\lambda_i = i^{-1.5}$, we set the source distribution as follows: with probability $1/B$, the $i$-th coordinate $\mathbf{x}_i \sim \{-\sqrt{\lambda_i}, \sqrt{\lambda_i}\}$ independently; with probability $1 - 1/B$, $\mathbf{x} = \mathbf{0}$. In the target domain, the $i$-th coordinate $\mathbf{x}_i \sim \{-\sqrt{\lambda_i}, \sqrt{\lambda_i}\}$ independently. We set $B = n^c$ and SGD learning rate $\gamma = 0.01n^{-c-\frac{1-c}{a+1}}$.

- **Figure 2 (d) Simulation of Corollary 14.** We set $\mathbf{S} = \mathbf{T}$ to simulate the hard instance in the $D_{\mathrm{KL}}$ bounded class, which is constructed in the proof of Corollary 14. SGD learning rate is set to $\gamma = 0.1n^{-\frac{1}{a+1}}$.

- **Figure 2 (e) Convergence rate of Rank-$1$ case in the setting of Corollary 12.** We set SGD learning rate $\gamma = 0.1n^{\frac{1-s}{s}}$, $a = 1.5$, $\mathbf{M} = \mathbf{I}$, and $\mathbf{T} = \mathbf{w}\mathbf{w}^\top$ where $\mathbf{w} \in \mathbb{R}^d$ and $\mathbf{w}_i \asymp i^{-(1+r)a/2}$.

- **Figure 2 (f) Comparison of the convergence rate of different learning rates in the setting of Corollary 12.** For ASGD, we set $\delta = 0.1$. $\gamma = 0.1 \cdot n^{0.5}$ is the theoretical optimal learning rate, and achieves minimax rate $\Theta(n^{-0.8})$. Choosing other learning rates ($\gamma = 0.1 \cdot n^c$, $c = 0, -0.4$) leads to sub-optimal convergence rates.

- **Figure 2 (g) Emergent behavior of different target domains in the setting of Example 1 with $d_0 = 7$ fixed.** We set SGD learning rate $\gamma = 0.1n^{-\frac{1}{a+1}}$, $\mathbf{T} = \mathrm{diag}\left\{ d_0^{(1+r)a} \left( \max\{i, d_0\} \right)^{-(1+r)a} \right\}_{i=1}^d$ according to Example 1. Target excess risk exhibits different rates for different $r$, while they start to decay at nearly the same sample size $n \approx 1000$.

- **Figure 2 (h) Emergent behavior of different target domains in the setting of Example 1 with $r = 0.1$ fixed.** We set SGD learning rate $\gamma = 0.1n^{-\frac{1}{a+1}}$, $\mathbf{T} = \mathrm{diag}\left\{ d_0^{(1+r)a} \left( \max\{i, d_0\} \right)^{-(1+r)a} \right\}_{i=1}^d$ according to Example 1. Target excess risk starts to decay at different sample sizes for different $d_0$, while they exhibit almost the same convergence rate.

We conduct numerical simulations in the setting of Figure 2(f) to compare different learning rate schedulers.

- Exp decay: Algorithm 1 in this paper.
- Poly decay: $\gamma_t = \gamma_0/t$, and $\delta_t = \delta_0/t$.
- Cosine decay (Loshchilov and Hutter, 2017): $\gamma = \gamma_{\min} + (\gamma_{\max} - \gamma_{\min})[1 + \cos(\pi(t \bmod T)/T)]/2$, and $\delta = \delta_{\min} + (\delta_{\max} - \delta_{\min})[1 + \cos(\pi(t \bmod T)/T)]/2$, where we set $T = n/4$, $\gamma_{\min} = \gamma_{\max}/n^2$ and $\delta_{\min} = \delta_{\max}/n^2$
- SHB: PyTorch implementation of momentum, $\beta$ is the momentum parameter.

We repeat each simulation 100 times, and plot the average target excess risk in Figure 3. The shaded area indicates 95% confidence interval.

# G   USE OF LLM

We use LLM to polish our paper writing.

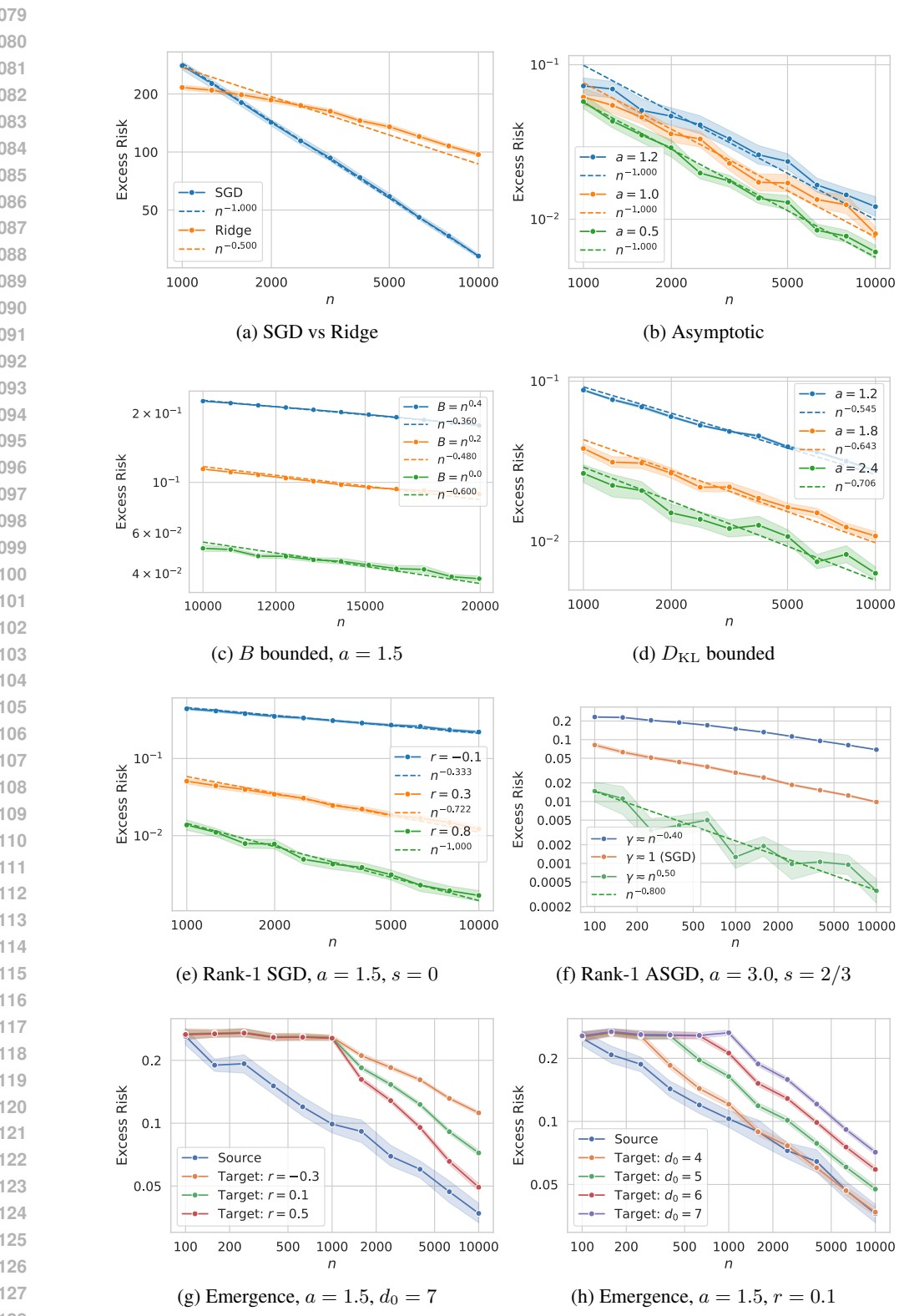

Figure 2: Simulation results. We set the source covariance matrix $\mathbf{S} = \{i^{-a}\}_{i=1}^{d}$, and other parameters are specified in the corresponding settings.

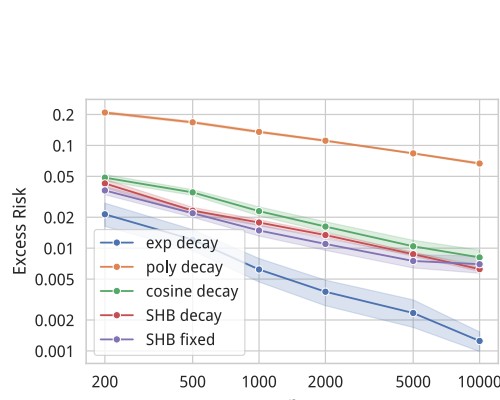

Figure 3: Comparison of the taget excess risk for different learning rate schedulers.

