# OpenReview forum: "Generalization Analysis of SGD in Linear Regression under Covariate Shift: A View from Preconditioning"
_ICLR.cc/2026/Conference — ICLR 2026 Conference Withdrawn Submission_

### Official Review · Reviewer_Uptf · 2025-10-30

**Soundness:** 3
**Presentation:** 2
**Contribution:** 2
**Rating:** 6
**Confidence:** 3

**Summary:**

The work of #22940 studies why (A)SGD can still perform well when there exists covariate shifts, i.e. a mismatch between the training data and the test data. Specifically, the authors focus on linear regression problems $\langle\mathbf{w}, \mathbf{x}\rangle$, where the input data changes between training and testing, but the way outputs depend on inputs stays the same. The paper analyzes SGD with common techniques like momentum and step-size decay and shows that, under certain conditions it can achieve near-optimal performance, meaning it generalizes well to new data. One new insight is viewing SGD as a type of *preconditioned* estimator, which helps explain when and why it works. The authors also compare SGD with other methods like ridge regression and find that SGD can actually be better in many situations (the so-called separations).

**Strengths:**

- The paper offers a novel perspective on accelerated stochastic gradient descent under covariate shift, particularly when both momentum and geometrically decaying step sizes are used. To the best of my knowledge, this fulfills a technical gap in the series of work.

- While the content is dense (as expected), the writing and overall structure is clear and easy to read. The paper not just providing upper bounds, it also includes minimax optimality conditions and separation results that position ASGD as superior to both ridge regression and vanilla SGD.

**Weaknesses:**

- The main result Theorem 6 (central risk bound) is proved under $M S=S M$. The authors call this a mild requirement, yet in practice $M$ encodes target task geometry (e.g., evaluation metric or interpolation space) and rarely commutes with the source covariance $S$, especially under real covariate shift where rotations between $S$ and $T$ are the rule, not the exception. This essentially narrows the scope of the main guarantee, while the instance-wise bound which mentioned for the general case is delayed to the appendix and not made first-class



-  Algorithm 1 cycles over $\left\lfloor\log _2 n\right\rfloor$ stages, resets step-sizes $\delta^{(\ell)}, \gamma^{(\ell)} \leftarrow\left(\delta_0, \gamma_0\right) / 4^{\ell-1}$, and assumes a fresh sample each inner iteration $\left(x_i, y_i\right)$. For a fixed dataset, this is equivalent to sampling with replacement, but the proof machinery leans on independence across the *semi-stochastic* recursion





- The UTKFace covariate shift that subsampling by age is purely synthetic and thus weaken the empirical evidence of the claims. There is no check that $P(y \mid x)$ is invariant in the feature space. Meanwhile the work omits other standard IW-ERM or kernel methods known to be competitive under covariate shift

**Questions:**

- Is your theorem 8 necessary? Based on the discussions in sec. 5, it seems that the (11) is only a sufficient condition for optimality of ASGD. Could authors consider either show necessity (or near‑necessity) under your modeling assumptions, or exhibit a broad class where failure of (11) provably prevent the ASGD optimality?

- I am wondering that whether authors can report the actual schedule that used to produce all figures. They said ‘grid search’ but didn't give ranges, seeds or selection protocol like selected $\alpha, \beta, \delta, \gamma$ across runs. Otherwise, the ASGD > ridge claim is not really convincing.



miscellaneous:

- first page, 'algorithms’s'

- L116: 'establishes upper bounds in linear for vanilla SGD', is that 'linear regression'?

- one need to unify the notations (e.g. $\otimes$ everywhere for Kronecker; reserve $\odot$ for Hadamard) in appendix, like (28)

-  consider explicitly stating the $\log _2 n$ to $\ln n$ conversion when you introduce $K=n / \log _2 n$

---

### Official Review · Reviewer_VMen · 2025-10-30

**Soundness:** 3
**Presentation:** 1
**Contribution:** 2
**Rating:** 2
**Confidence:** 4

**Summary:**

This paper studies learning under covariate shift in the setting of linear regression. The paper compares SGD and ASGD, and argues that SSGD achieves optimality for a broader class of instances.

**Strengths:**

1. The assumptions and theorems are followed with comprehensive discussions.
2. The theory is well supported by proofs.

**Weaknesses:**

1. The flow of the paper may cause difficulties in understanding. Specifically,
- The introduction of ASGD is a bit absurd. One may wonder why we consider ASGD is the main focus instead of vanilla SGD. See Question 1. I would recommend introducing properties of SGD, e.g.,  Corollary 12, before introducing ASGD.
- The definition of "optimality" does not seem clear to me in Theorem 8.
2. It looks like the abstract and the main body convey quite different messages. The abstract is mainly about why SGD performs well even in the case of covariate shift (which, according to my understanding, remains unanswered), but the main body seems to focus on the advantage of ASGD compared with SGD in the case of covariate shift.
3. The comparison between ASGD and SGD looks vague and of inferior importance to me.
- The argument of ASGD achieves optimality for a broader class of problem instances is not clearly illustrated. Specifically, Corollary 12 does not seem to provide a clear comparison between SGD and ASGD. It would be good if the authors could guide me to a similar theorem for ASGD.
- As mentioned in Weakness 2, the main question, why SGD performs well in covariate shift, does not have a clear answer to me. Compared with this main question, the role of momentum in SGD-like methods may not be as important.

**Questions:**

1. How to interpret Eq. (7)? Is this condition used anywhere else in the paper?
2. Ferbach et al. (2025) studied different ASGD variants. Will the results of this paper be any different?

Ferbach et al. Dimension-adapted Momentum Outscales SGD. 2025.

---

### Official Review · Reviewer_Ybga · 2025-10-31

**Soundness:** 2
**Presentation:** 2
**Contribution:** 3
**Rating:** 4
**Confidence:** 3

**Summary:**

The paper presents a theoretical analysis of SGD for linear regression in the presence of covariate shift. It is a common problem in ML where the distribution of input data used for training a model differs from the distribution of data the model will encounter when deployed, while the underlying relationship between inputs and outputs remains the same. The authors deal with the question why SGD-type algorithms, which are typically trained only on the source data, generalize so well to the target data without explicit knowledge of its distribution.

They derive a tight upper bound on the prediction error for ASGD, incorporating practical techniques as momentum and exponentially decaying step sizes. Introduce a novel perspective by viewing ASGD as a preconditioned estimator. The analysis demonstrates scenarios when ASGD can be superior to other methods as standard ridge regression.

**Strengths:**

Analyzing SGD as a preconditioned estimator is innovative. Addressing a practical problem in ML and providing theory. Theorem 6 connects the algorithm performance with the spectral properties of the source and target data distributions. Theory covers accelerated methods, establishes minimax optimality.

**Weaknesses:**

The focus is strictly on linear regression though the motivation comes from the success of SGD in complex domains like large language models. The theory does not directly apply to the non-linear, non-convex optimization problems that prevail in modern ML. This is why I think the practical impact on the broader ML community is questionable.

The manuscript is dense with complex notation and very technical. It is difficult to parse for anyone not already an expert in the field. Intuitive explanations could guide the reader through the technical arguments.

The central conclusion is that SGD generalizes well under covariate shift when the target task can be solved using the information in the principal directions of the source data. So, the paper shows that transfer learning works best when the source and target tasks share relevant features, which is not surprising or fundamentally new.

While this is strong theoretical work, it may be better suited for a more specialized venue in statistics or theoretical machine learning.

**Questions:**

The assumption in Theorem 6 is that constraint matrix M and the source covariance S commute. Why do you believe it is mild? Why it is representative of real-world problems? Isn't it just a mathematically convenient condition, that simplifies the spectral analysis?

---

### Official Review · Reviewer_JPLS · 2025-10-31

**Soundness:** 3
**Presentation:** 2
**Contribution:** 3
**Rating:** 4
**Confidence:** 2

**Summary:**

This paper provides an upper bound on the target excess risk for estimators obtained via (Accelerated) Stochastic Gradient Descent in the context of linear regression. Specifically, for ASGD, the authors derive the necessary conditions on the step sizes and moments under which the algorithm achieves the minimax optimal rate of $\tilde{O}(1/n)$, where the second moment of the covariates ($S$), the second moment of the target variable ($T$), and the elliptical constraint on the true parameter ($M$) are all given. The suboptimal target excess rate for the ridge regression $O(1/\sqrt{n}) is also derived. The experimental results shown are consistent with the derived rate.

**Strengths:**

- The authors provide a rigorous derivation of the theoretical upper bound on the target excess risk for the given estimators in linear regression problem.

- The presented theoretical framework is convenient in the sense that it can deduce and compare the risk of estimating methods for the linear regression problem such as ridge regression or SGD.

**Weaknesses:**

- The description on the experimental setup regarding Figure 1 seems to be insufficient. Please refer to the Questions section. Also, it is difficult to understand which experimental setting each solid line and dashed line represents.

- The presented analysis on the target excess risk is inherently limited to the linear regression model. In the covariate shift problem occurring in various fields, such as LLM, the relationship between $x$ and $y$ is typically far more complex than a linear model.

Typo in line 83 - 84, ASGD achieves the optimal $\tilde{O}(1/\sqrt{n})$ rate... : The rate for ASGD and ridge regression are swapped.

**Questions:**

- Why is the UTK-face dataset used to show the target excess risk with respect to $n$?. What is the covariate $x$ with $d=10$? How is S, T, and M chosen? How is the excess risk and the true parameter $w^*$ computed? Also, please elaborate which experimental setting each solid line and dashed line in Figure (a)-(d) represents.

- Can you deduce a similar analysis on the sparse regression problem where it is known that the number of the nonzero entries of $w^*$ is far less than $d$? Note that such problem is usually solved with LASSO or ElasticNet.

---

### Note · Authors · 2025-11-30

I have read and agree with the venue's withdrawal policy on behalf of myself and my co-authors.